# TSGym: Design Choices for Deep Multivariate Time-Series Forecasting

**Shuang Liang**[1,*], **Chaochuan Hou**[1,*], **Xu Yao**[1], **Shiping Wang**[3],
**Minqi Jiang**[1,*,†], **Songqiao Han**[1,2,†], **Hailiang Huang**[1,2,†],
[1]AI Lab, Shanghai University of Finance and Economics [2]Ant Group
[3]MoE Key Laboratory of Interdisciplinary Research of Computation and Economics
{liangs1104,houchaochuan,yaoxu}@stu.sufe.edu.cn, shiping.wsp@antgroup.com,
{jiangminqi,han.songqiao,hlhuang}@shufe.edu.cn,

## Abstract

Recently, deep learning has driven significant advancements in multivariate time series forecasting (MTSF) tasks. However, much of the current research in MTSF tends to evaluate models from a holistic perspective, which obscures the individual contributions and leaves critical issues unaddressed. Adhering to the current modeling paradigms, this work bridges these gaps by systematically decomposing deep MTSF methods into their core, fine-grained components like series-patching tokenization, channel-independent strategy, attention modules, or even Large Language Models and Time-series Foundation Models. Through extensive experiments and component-level analysis, our work offers more profound insights than previous benchmarks that typically discuss models as a whole.

Furthermore, we propose a novel automated solution called TSGym for MTSF tasks. Unlike traditional hyperparameter tuning, neural architecture searching or fixed model selection, TSGym performs fine-grained component selection and automated model construction, which enables the creation of more effective solutions tailored to diverse time series data, therefore enhancing model transferability across different data sources and robustness against distribution shifts. Extensive experiments indicate that TSGym significantly outperforms existing state-of-the-art MTSF methods. All code is publicly available on https://github.com/SUFE-AILAB/TSGym.

## 1 Introduction

Multivariate time series refer to time series data involving multiple interdependent variables, which are widely present in various fields such as finance [51], energy [5, 17], traffic [15, 69], and health [9, 29]. Among the numerous analysis tasks, multivariate time series forecasting (MTSF) attracts substantial attention from the research community due to its significant practical applications. Traditional approaches to MTSF are largely based on statistical methods [4, 72] and machine learning techniques [23, 44]. In recent years, deep learning (DL) has become the most active area of research for MTSF, driven by its ability to handle complex patterns and large-scale datasets effectively [59].

Early academic efforts of deep MTSF methods like RNN-type methods [67] are reported to struggle with capturing long-term temporal dependencies due to their inherent limitations of gradient vanishing or exploding problem [76, 78]. More recently, Transformer [56] shows significant potential, largely due to the effectiveness of its attention mechanisms in modeling temporal correlation [56, 62]. Consequently, attention mechanism has continuously been studied in MTSF, with a focus on adapting them to time series data, for instance, by exploiting sparsity inductive bias [33, 76], transforming time and frequency domains [78], and fusing multi-scale series [36]. While simpler MLP-based structures

emerged [70] offering alternatives to the established Transformer architecture in MTSF, notable modeling strategies like series-patching and channel-independent [45], significantly enhanced the performance of Transformer-based methods, thereby sustaining research interest in them. Building upon these developments, large time-series models including large language models (LLMs) [28, 79, 25] and time series foundation models (TSFMs) [27, 42] have recently been introduced, achieving promising results and fostering new research directions for MTSF. Alongside these advancements in model architectures, active research within the deep MTSF community also focuses on other critical topics, such as variable (channels) dependency modeling [45, 37, 75], series normalization methods [40, 19], and trend-seasonal decomposition [70, 38].

As the field of MTSF continues to diversify, existing studies typically address critical concerns about methodological effectiveness, either by conducting large-scale benchmarks [59, 52, 47] or performing model selection via AutoML [2, 20]. However, we identify three main challenges with these prevailing approaches: First, *the granularity of existing studies is insufficient*. Current benchmarking works evaluate or select models as a whole, which hinders a deeper understanding of the mechanisms that drive model performance. In AutoML, this lack of granularity prevents breakthroughs beyond the limits of existing models. Second, *the scope of existing studies is limited*. Current benchmarking and automated selection efforts are often confined to restricted model architectures or hyperparameters, without covering a broad range of data processing methods or feature modeling techniques. Third, *the range of existing studies is narrow*. Existing studies tend to cover only a subset of network architectures and often lack discussions on more diverse models, such as LLMs and TSFMs.

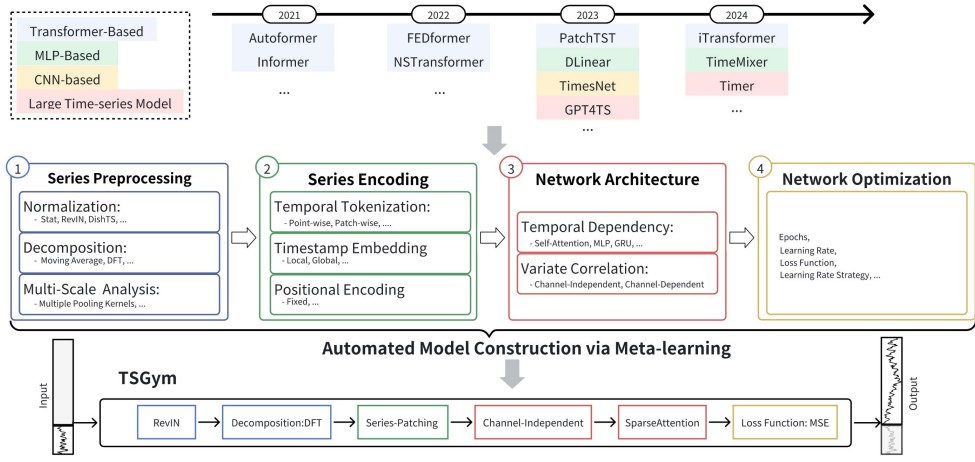

Figure 1: The Pipeline of Existing Multivariate Time Series Forecasting Methods.

To bridge these gaps, we propose TSGym—a framework designed for the **Large-scale Evaluation**, **Analysis**, and **Automated Model Construction** in deep MTSF tasks. Rather than viewing models as unified entities, TSGym systematically deconstructs popular deep MTSF methods by organizing them into distinct design dimensions that cover the entire time series modeling pipeline (see Fig. 1 and Table 1). Through extensive experiments, TSGym conducts fine-grained, isolated evaluations of core components, thereby identifying key design dimensions/choices and valuable insights from the vast MTSF methods. Moreover, TSGym proposes the first component-level model construction in MTSF tasks, which effectively overcomes limitations in the previous automation methods by enabling more flexible and customized model designs tailored to data characteristics. Extensive experimental results indicate that the proposed TSGym generally outperforms existing SOTA methods. We summarize the key contributions of TSGym as follows:

**Component-level evaluation of MTSF methods**. We propose TSGym, the first large-scale benchmark that systematically decouples deep MTSF methods. By evaluating 16 design dimensions across 10 benchmark datasets, TSGym offers key insights to inform future development for MTSF.

**Automated MTSF model construction**. Leveraging meta-learning, TSGym develops models that outperform current SOTA methods, offering the MTSF community an effective, automated, and data-adaptive solution for model design.

**Discussion on emerging large time-series models**. TSGym broadens current MTSF scope by applying systematic evaluation and automated combination not only to well-established models like MLP and Transformer, but also to novel large time-series models like LLMs and TSFMs.

## 2 Related Work

### 2.1 Deep Learning-based MTSF

MTSF evolves from traditional statistical methods like ARIMA and Gaussian processes to modern deep learning approaches. Recurrent Neural Networks (RNNs) introduce memory mechanisms for sequential data but struggle with long-term dependencies. Temporal Convolutional Networks (TCNs) improve this by capturing multi-scale patterns, though their fixed window sizes limit global context. Transformers, using self-attention, enable long-range forecasting but introduce high computational complexity, leading to efficient variants like sparse attention [65] and patch-based models [45]. Multilayer Perceptrons (MLPs) regain attention as simple yet effective models [71], with numerous variants offering competitive performance [14, 68, 16, 38]. Leveraging NLP foundation models, LLM adaptation approaches use frozen backbones and prompt engineering [26, 79] or fine-tuning [11] to transfer pretrained knowledge. Simultaneously, pure TSFMs trained on large datasets achieve zero-shot generalization [42, 21], though constrained by Transformers' complexity. Our TSGym framework modularizes six core backbones—RNNs, CNNs, Transformers, MLPs, LLMs, and TSFMs—offering flexible, hybrid integration based on temporal dependencies and resource needs.

In recent advancements in MTSF, we summarize the design paradigm through a unified pipeline (Fig. 1), consisting of four stages: *Series Preprocessing→Series Encoding→Network Architecture→Network Optimization*. Additionally, several specialized modules are proposed to enhance predictive accuracy by addressing non-stationarity, multi-scale dependencies, and inter-variable interactions. We categorize these developments into 6 specialized modules:

**(1) Normalization** methods like RevIN [30] adjust non-stationary data, improving robustness against distribution shifts. **(2) Decomposition methods**, such as Autoformer [65]'s trend-seasonality separation, isolate non-stationary components, making the data more predictable by separating trends from seasonality. **(3) Multi-scale analysis** extracts temporal patterns across granularities, as in TimeMixer [58], capturing both high-frequency fluctuations and low-frequency trends through hierarchical resolution modeling. **(4) Temporal tokenization techniques** like PatchTST[45]'s subseries-level embedding represent time series hierarchically, improving the capture of complex temporal semantics. **(5) Temporal dependency** modeling through architectures like Transformers leverages self-attention to capture long-range dependencies, effectively modeling both short- and long-term relationships. **(6) Variate correlation learning**, exemplified by DUET [48], models inter-variable dependencies using frequency-domain metric learning, improving predictions by capturing interactions across variables.

To provide a more detailed categorization and comprehensive technical specifications, please refer to Appx. B. Due to the extensive focus and continuous evolution of these modules in MTSF research, TSGym strives to decouple and modularize these key modules, exploring their real contributions and enabling more flexible model structure selection and configuration.

### 2.2 Benchmarks for Time Series Forecasting

Recent time series forecasting benchmark studies [59, 52, 47, 41] have conducted large-scale experiments across a diverse range of datasets. However, most of these works treat current models as monolithic entities. TSlib[1] [59], one of the most popular repositories for time series analysis, provides a comprehensive survey and evaluates recent time series models across various time series analysis tasks. From the perspective of time series characteristics, BasicTS [52] analyzes model architectures and the strategy of treating channels (or variables) independently. With a more extensive experimental setup, TFB [47] additionally includes machine learning and statistical forecasting methods, and covers datasets from a broader range of domains. More recently, OpenLTM[2] [41] provides a system to evaluate Time Series Foundation Models as well as Large language Models for time series methods.

Although some surveys and benchmarks analyze the fine-grained components of time series models, their scope is often limited. Wen et al. [61] discuss various time series data augmentation techniques

---

[1]https://github.com/thuml/Time-Series-Library
[2]https://github.com/thuml/OpenLTM

and evaluate their effectiveness. Another survey [62] systematically reviews fine-grained components within Transformer-based architectures, but lacks broader coverage of model structures and experimental evaluations. To the best of our knowledge, TSGym is the first benchmark that not only provides component-level fine-grained analysis, but also conducts large-scale empirical evaluations.

## 2.3 AutoML for Time Series Forecasting

Current automated approaches for DL-based MTSF can be categorized into ensemble-based [53] and meta-learning-based [2, 20] methods. The former fits and integrates various models from a predefined pool with ensemble techniques, which inevitably incurs substantial computational cost. The latter leverages meta-features to characterize datasets and selects optimal models for the given datasets. However, both approaches operate at the model level and struggle to surpass the performance ceiling of existing methods. AutoCTS++ [66] achieves automated selection by searching over model architectures and hyperparameters, but its search space is limited in scope. In contrast, TSGym is the first framework to support automated selection over a wide range of fine-grained components for MTSF, extending beyond narrow model structures, hyperparameters, and data processing strategies.

A closely related work is our previous effort, ADGym[24], which is designed for tabular anomaly detection with model decomposition. Differently, TSGym deals with multivariate time series data, which presents more complex data processing design choices, such as series sampling, series normalization, and series decomposition. Besides, TSGym considers finer-grained model structures, such as various attention variants in Transformers, and broader network types, including LLMs and TSFMs. It is worth mentioning that the success of TSGym validates the universality of the model decomposition framework, marking an innovation and progression distinct from ADGym. Further details on the differences between two works can be found in Appx.F.

# 3 TSGym: *Benchmarking* and *Automating* Design Choices in Deep MTSF

## 3.1 Problem Definition for MTSF

In this paper, we focus on the common MTSF settings for time series data containing $C$ variates. Given historical data $\chi = \{\boldsymbol{x}_1^t, \ldots, \boldsymbol{x}_C^t\}_{t=1}^{L}$, where $L$ is the look-back sequence length and $\boldsymbol{x}_i^t$ is the $i$-th variate, the forecasting task is to predict $T$-step future sequence $\hat{\chi} = \{\hat{\boldsymbol{x}}_1^t, \ldots, \boldsymbol{x}_C^t\}_{t=L+1}^{L+T}$. To avoid error accumulation ($T > 1$), we directly predict all future steps, following [76].

## 3.2 Large Benchmarking towards Design Choices of Deep MTSF

Considering the above numerous methods proposing for MTSF tasks, the foremost priority involves decoupling the current state-of-the-art (SOTA) methods and further conducting large-scale benchmark to identify the core components that really drive the improvements in time-series forecasting.

Following the taxonomy of the previous study [62, 71], we decouple existing SOTA methods according to the standard process of MTSF modeling, while significantly expanding the diversity of the modeling pipeline. Based on the flow direction from the input to the output sequence, the **Pipeline** of TSGym includes: *Series Prepropcessing→Series Encoding→Network Architecture→Network Optimization*, as is demonstrated in Fig. 1. Moreover, we structure each pipeline step according to distinct **Design Dimensions**, where a DL-based time-series forecasting model can be instantiated by specified **Design Choices**, as is shown in Table 1.

Through the proposed design dimensions and choices, TSGym provides detailed description of time-series modeling pipeline, disentangling key elements within mainstream time-series forecasting methods and facilitating component-level comparison/automated construction. For example, TS-Gym includes multi-scale mixing module proposed in TimeMixer [58], Inverted Encoding method proposed in iTransformer [37], Channel-independent strategy and Series-Patching encoding used in PatchTST [45], various attention mechanism discussed in [62], and also LLM and TSFM network type choices that are often integrated without fully considering their interactions with other design dimensions. Detailed descriptions of all design choices are provided in Appx. G.1.

## 3.3 Automated construction MTSF models via TSGym

**Overview**. Differing from traditional methods that focus on selecting an off-the-shelf model, TSGym aims to customize models given the downstream MTSF tasks and data descriptions. Given a pre-defined conflict-free model set $\mathcal{M} = \{M_1, ..., M_m\}$, each model $M_i$ is instantiated by the

Table 1: TSGym supports comprehensive design choices for deep time-series forecasting methods.

| Pipeline | Design Dimensions | Design Choices |
|---|---|---|
| ↓Series Preprocessing | Series Normalization | [None, Stat, RevIN, DishTS] |
| | Series Decomposition | [None, MA, MoEMA, DFT] |
| | Series Sampling/Mixing | [False, True] |
| ↓Series Encoding | Channel Independent | [False, True] |
| | Sequence Length | [48, 96, 192, 512] |
| | Series Embedding | [Inverted Encoding, Positional Encoding, Series Patching] |
| ↓Network Architecture | Network Type | [MLP, RNN, Transformer, LLM, TSFM] |
| | Series Attention | [Null, SelfAttn, AutoCorr, SparseAttn, FrequencyAttn, DestationaryAttn] |
| | Feature Attention | [Null, SelfAttn, SparseAttn, FrequencyAttn] |
| | $d\_model$ | [64, 256] |
| | $d\_ff$ | [256, 1024] |
| | Encoder Layers | [2, 3] |
| ↓Network Training | Epochs | [10, 20, 50] |
| | Loss Function | [MSE, MAE, HUBER] |
| | Learning Rate | [1e-3, 1e-4] |
| | Learning Rate Strategy | [Null, Type1] |

design choice combinations illustrated in Table 1. TSGym learns the mapping function from these automatically combined models to their associated forecasting performance on the training datasets, and generalize to the test dataset(s) to select the best model based on predicted results.

**Meta-learning for automated MTSF model construction**. Formally speaking, TSGym propose $k$ design dimensions $\mathcal{DD} = \{DD_1, ..., DD_k\}$ for comprehensively describing each step of aforementioned pipeline in deep learning time-series modeling. Each design dimension $DD_i$ represents a set containing elements of different design choices $DC$. By taking the Cartesian product of the sets $\mathcal{DD}$ corresponding to different design dimensions, we obtain the pool of all valid model combinations $\mathcal{M} = DD_1 \times DD_2 \times \cdots \times DD_k = \{(DC_1, DC_2, \ldots, DC_k) \mid DC_i \in DD_i, i = 1, 2, \ldots, k\}$. Considering the potentially large number of combinations and the computational cost, we randomly sampled $\mathcal{M}$ to $\mathcal{M}_s$, where $M_i = (DC_1 = RevIN, DC_1 = DFT, ..., DC_k = Type1) \in \mathcal{M}_s$, for example, which means $M_i$ instantiates RevIN method to normalize input series, then decompose it to the seasonal and trend term. Subsequently, following the Series Encoding and Network Architecture constructing pipeline (as illustrated in Table 1), finally the Type1, i.e., a step decay learning rate strategy is employed to adjust the learning rate for updating the model parameters.

Suppose we have $n$ training datasets $\mathcal{D}_{\text{train}} = \{\mathcal{D}_1, \ldots, \mathcal{D}_n\}$ and the number of sampled model combinations (i.e., the size of the set $\mathcal{M}_s$) is $m$, TSGym conducts extensive experiments on $n$ historical training datasets to evaluate and further collect the forecasting performance of $m$ model combinations. TSGym then acquire the MSE performance matrix $\boldsymbol{P} \in \mathbb{R}^{n \times m}$, where $\boldsymbol{P}_{i,j}$ corresponds to the $j$-th auto-constructed MTSF model's performance on the $i$-th training dataset. Since the difficulty of prediction tasks varies across training datasets, leading to significant differences in the numerical range of performance metrics. Directly using these metrics (e.g., MSE) as training targets of a meta-learner may result in overfitting on more difficult dataset(s). Therefore, we convert the performance metrics of $\mathcal{M}_s$ into their corresponding normalized ranking, where $\boldsymbol{R}_{i,j} = rank(P_{i,j})/m \in [0,1]$ and smaller values indicate better performance on the corresponding dataset.

Distinguished from previous model selection approaches [2, 3], TSGym decouples more recently MTSF methods (including MLP-Mixer-type, Transformer-based, LLM and TSFM models), and supports fine-grained model construction at the component level, rather than being constrained to a fixed, limited set of existing models, which enables significantly greater flexibility and effectiveness. Specifically, TSGym follows the idea of meta-learning to construct a meta-learner that learns the mapping function $f(\cdot)$ from training dataset $\mathcal{D}_i$ and model combination $M_j$, to the performance rankings $\boldsymbol{R}_{i,j}$, as is shown in Eq. 1. We leverage meta-features $\mathbf{E}_i^{meta}$ that capture multiple aspects such as statistical, temporal, spectral, and fractal features, and distribution shift metrics to fully describe the complex data characteristics of time series datasets. Learnable continuous embeddings $\mathbf{E}_j^{comp}$ are used to represent different model combinations and are updated through the gradient backpropagation of the meta-learner.

$$f(\mathcal{D}_i, M_j) = \boldsymbol{R}_{i,j}, f \; : \; \underbrace{\mathbf{E}_i^{meta}}_{\text{meta features}}, \; \underbrace{\mathbf{E}_j^{comp}}_{\text{component embed.}} \; \mapsto \boldsymbol{R}_{i,j} \;, \;\; i \in \{1, \ldots, n\}, j \in \{1, \ldots, m\} \quad (1)$$

We used a simple two-layer MLP as the meta-learner and trained it through a regression problem, thereby transferring the learned mapping to new test datasets. For a newcoming dataset (i.e., test dataset $\mathbf{X}_{\text{test}}$), we acquire the predicted relative ranking of different components using the trained $f(\cdot)$, and select top-1 ($k$) to construct MTSF model(s). Note this procedure is zero-shot without needing any neural network training on $\mathbf{X}_{\text{test}}$ but only extracting meta-features and pipeline embeddings. We show the effectiveness of the meta-predictor in §4.3.

# 4 Experiments

## 4.1 Experiment Settings

**Datasets**. Following most prior works [65, 64, 26], we adopt 9 datasets as experimental data for MTSF tasks, ETT (4 subsets), Traffic, Electricity, Weather, Exchange, ILI. And we utilize the M4 dataset for short-term forecasting tasks. The forecast horizon $L$ for long-term forecasting is $\{96, 192, 336, 720\}$, while for the ILI dataset, it is $\{24, 36, 48, 60\}$. For short-term forecasting, the forecast horizons are $\{6, 8, 13, 14, 18, 48\}$. More details can be seen in Appx. A.

**Baseline**. We present a comprehensive set of baseline comparison experiments to demonstrate the superior performance of the pipelines automatically constructed by TSGym. Due to space limitations, the baseline methods presented in this section include the latest clustering-based approach DUET [48], time series mixing methods (TimeMixer [58]), MLP-based methods (MICN [57]), RNN-based methods (SegRNN [34]), CNN-based methods (TimesNet [64]), and Transformer-based methods (PatchTST [45], Crossformer [75], Autoformer [65]). We present experiments based on the complete baseline in the Appx. H.

**Evaluation Metrics**. We follow the experimental setup of most prior works, using Mean Squared Error (MSE) and Mean Absolute Error (MAE) as evaluation metrics for long-term forecasting tasks, and using Symmetric Mean Absolute Percentage Error (SMAPE), Mean Absolute Scaled Error (MASE), and Overall Weighted Average (OWA) as metrics for short-term forecasting tasks. The mathematical formulas for these evaluation metrics are provided in the Appx. D.

**Meta-predictor in TSGym**. The meta-predictor is instantiated as a two-layer MLP and trained for 100 epochs with early stopping. The training process utilizes the Adam optimizer [31] with a learning rate of 0.001 and batch size of 512. See details in Appx. G.2.

## 4.2 Large Evaluation on AD Design Choices

In this work, we perform large evaluations on the decoupled pipelines according to the standard procedure of MTSF methods. Such analysis is often overlooked in previous studies, and we investigate each design dimension of decoupled pipelines by fixing its corresponding design choice (e.g., Self Attention), and randomly sampling other dimensional design choices to construct MTSF pipelines.

In the following sections, we provide systematic conclusions based on long-term MTSF experimental results, addressing several gaps in the current MTSF research community. Specifically, different design choices are compared and demonstrated using a spider chart, where each vertex represents a dataset. Design choices that are closer to the vertices exhibit superior performance in their respective design dimensions. We analyze these components based on the different design dimensions, namely *Series Preprocessing*, *Series Encoding* and *Network Construction*. Finally, we evaluated the performance on four datasets under the fixed model architectures of LLMs or TSFMs, and discussed the effects of various design choices in the subsection on *Large Time Series Models*. More detailed information, including the complete design choices and their performance on short-term time series forecasting tasks, is provided in the Appx. H.

*Series Preprocessing*. Based on the systematic evaluation of Series Preprocessing strategies in Fig. 2a and Fig. 2b, several key insights emerge. Series Normalization (Fig. 2a) proves universally effective, with RevIN and Stationary achieving the lowest MSE (75th percentile) across diverse datasets, establishing them as essential baselines for stabilizing non-stationary dynamics. In contrast, Series Decomposition (Fig. 2b) demonstrates varying effectiveness depending on the dataset. For example, decomposition methods improve performance on datasets like ETTm1 when using MA, while others, such as ETTh1, ECL, ETTh2, and traffic datasets, perform better without decomposition.

*Series Encoding*. Several clear observations emerge from the Series Encoding stage, as illustrated in Fig.2c and Fig.2d. First, as shown in panel Fig. 2c, channel-independent methods consistently

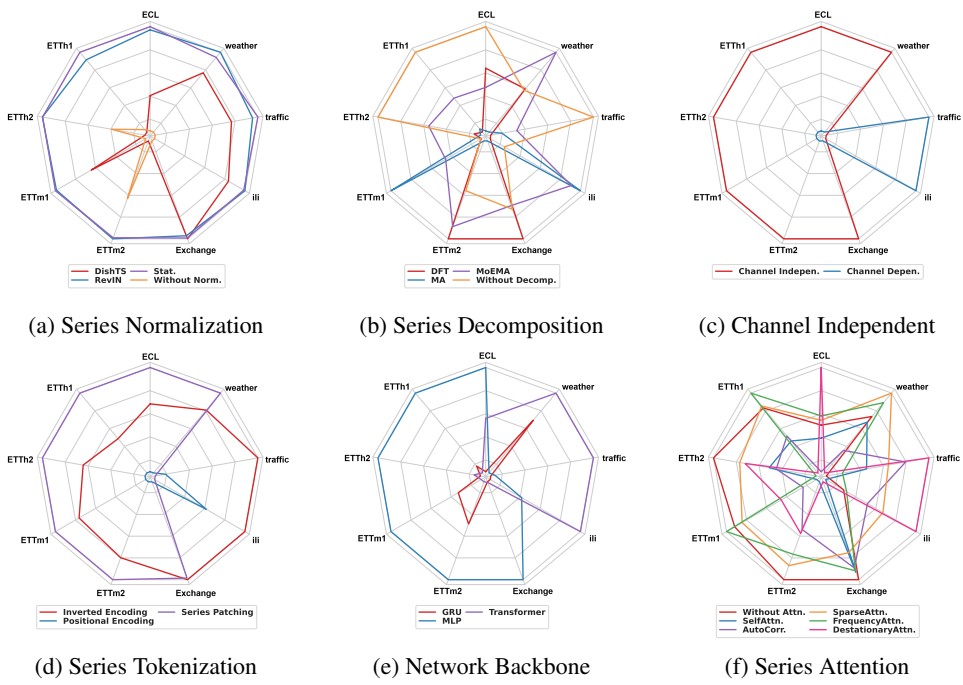



(a) Series Normalization     (b) Series Decomposition     (c) Channel Independent

(d) Series Tokenization     (e) Network Backbone     (f) Series Attention



Figure 2: Overall performance across key design dimensions. The results (MSE) are based on the 75th percentile across all forecasting horizons.

outperform channel-dependent ones, with the exception of traffic and ILI, highlighting the advantage of modeling each variable separately. Regarding tokenization strategies Fig. 2d, both patch-wise (sequence patching) and series-wise (inverted encoding) approaches outperform the traditional point-wise encoding. Among them, patch-wise encoding shows strong performance across most datasets, whereas inverted encoding proves particularly effective on the traffic dataset.

***Network Construction***. As is shown in Fig. 2e, we find that complex network architectures like the Transformer are not always necessary, which only performs better than MLP on weather, traffic and ILI datasets. This result is reasonable since more sophisticated DL backbone like Transformer and its variants often require specific hyperparameter combinations and tailored architectural designs to achieve satisfactory performance. We indicate that these observed results further emphasize the importance of automated model construction tailored to specific data characteristics. Indeed, we show in § 4.3 that including the Transformer architecture in the set of design choices would enhance the performance of constructed model via TSGym. Fig. 2f suggests that employing Attention mechanisms for modeling dependencies of input sequence does not offer significant advantages, which aligns with recent findings [70]. Additionally, the spider chart shows no significant performance differences were observed among different variants of attention mechanisms.

***Large Time Series Models***. We choose three design dimensions crucial to large time-series models for discussion, as is shown in Fig.3. The most surprising finding pertains to the model backbones. We observe that GPT4TS demonstrates stable and competitive performance across most datasets. In contrast, Time-LLM, which is also based on LLMs, exhibits opposite results. A highly plausible explanation is that time series embeddings processed through different methods struggle to align consistently with the embedding space of word representations, which is an important part in Time-LLM, resulting in suboptimal performance of Time-LLM under diverse experimental configurations.

## 4.3 Automatic Component Construction via TSGym

Extensive experimental results discussed above indicate that in deep time series modeling, most design choices are determined by data characteristics, meaning one-size-fits-all approaches are seldom effective. This, in turn, emphasizes the necessity of automated model construction.

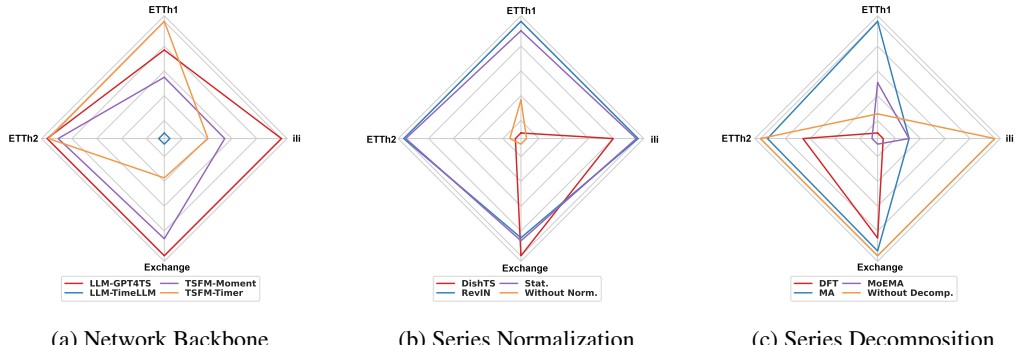

(a) Network Backbone   (b) Series Normalization   (c) Series Decomposition

Figure 3: Overall performance of different design choices across 3 design dimensions (Network Backbone 3a, Series Normalization 2a, Series Decomposition 2b) when using LLMs or TSFMs. The results (MSE) are based on the 75th percentile across all forecasting horizons.

In this subsection, we compare the MTSF pipeline selected by TSGym with existing SOTA methods. Through large-scale experiments, we found that TSGym outperforms existing SOTA models in both long- and short-term MTSF tasks. Regarding algorithm efficiency, our experiments demonstrate that even when limited to a search pool of lightweight model structures, such as MLP and RNNs, TSGym can still achieve competitive results. We analyze the effectiveness of the pipelines automatically constructed by TSGym through five key questions as follows. Additional details, such as the results based on more metrics and more complex meta-features, can be found in the Appx.H.

**Question 1: Is the model constructed by meta-predictor better than existing SOTA methods?**
Comprehensive forecasting results are presented in Table 2 and 3, where the best performances are highlighted in **red** and the second-best results are underlined. Compared with other state-of-the-art forecasters, TSGym demonstrates superior capability, especially in handling high-dimensional time series data. Its consistent top-ranking performance across multiple datasets underlines its robustness and effectiveness for complex multivariate forecasting tasks.

Table 2: Short-term forecasting task on M4. The results are averaged from several datasets under different sample intervals. See Table in Appendix for the full results.

| Models | TSGym (ours) | TimeMixer | MICN | TimesNet | PatchTST | DLinear | Crossformer | Autoformer | SegRNN |
|---|---|---|---|---|---|---|---|---|---|
| OWA | **0.872** | 0.884 | 0.984 | 0.907 | 0.965 | 0.922 | 8.856 | 1.273 | 1.007 |
| SMAPE | 12.013 | **11.985** | 13.025 | 12.199 | 12.848 | 12.511 | >30 | 16.392 | 13.509 |
| MASE | **1.575** | 1.615 | 1.839 | 1.662 | 1.738 | 1.693 | >10 | 2.317 | 1.823 |

Table 3: Long-term forecasting task. The past sequence length is set as 36 for ILI and 96 for the others. All the results are averaged from 4 different prediction lengths, that is $\{24, 36, 48, 60\}$ for ILI and $\{96, 192, 336, 720\}$ for the others. See Table in Appendix for the full results.

| Models | TSGym (Ours) | | DUET [48] | | TimeMixer [58] | | MICN [57] | | TimesNet [64] | | PatchTST [45] | | DLinear [71] | | Crossformer [75] | | Autoformer [34] | | SegRNN [65] | |
|---|---|---|---|---|---|---|---|---|---|---|---|---|---|---|---|---|---|---|---|---|
| Metric | MSE | MAE | MSE | MAE | MSE | MAE | MSE | MAE | MSE | MAE | MSE | MAE | MSE | MAE | MSE | MAE | MSE | MAE | MSE | MAE |
| **ETTm1** | **0.357** | **0.383** | 0.407 | 0.409 | 0.384 | 0.399 | 0.402 | 0.429 | 0.432 | 0.430 | 0.390 | 0.404 | 0.404 | 0.407 | 0.501 | 0.501 | 0.532 | 0.496 | 0.388 | 0.404 |
| **ETTm2** | **0.261** | **0.319** | 0.296 | 0.338 | 0.277 | 0.325 | 0.342 | 0.391 | 0.296 | 0.334 | 0.288 | 0.334 | 0.349 | 0.399 | 1.487 | 0.789 | 0.330 | 0.368 | 0.273 | 0.322 |
| **ETTh1** | 0.426 | 0.440 | 0.433 | 0.437 | 0.448 | 0.438 | 0.589 | 0.537 | 0.474 | 0.464 | 0.454 | 0.449 | 0.465 | 0.461 | 0.544 | 0.520 | 0.492 | 0.485 | **0.422** | **0.429** |
| **ETTh2** | **0.358** | **0.400** | 0.380 | 0.403 | 0.383 | 0.406 | 0.585 | 0.530 | 0.415 | 0.424 | 0.385 | 0.409 | 0.566 | 0.520 | 1.552 | 0.908 | 0.446 | 0.460 | 0.374 | 0.405 |
| **ECL** | 0.170 | **0.265** | 0.179 | 0.262 | 0.185 | 0.273 | 0.186 | 0.297 | 0.219 | 0.314 | 0.209 | 0.298 | 0.225 | 0.319 | 0.193 | 0.289 | 0.234 | 0.340 | 0.216 | 0.302 |
| **Traffic** | **0.435** | **0.313** | 0.797 | 0.427 | 0.496 | 0.313 | 0.544 | 0.320 | 0.645 | 0.348 | 0.497 | 0.321 | 0.673 | 0.419 | 1.458 | 0.782 | 0.637 | 0.397 | 0.807 | 0.411 |
| **Weather** | **0.229** | **0.268** | 0.252 | 0.277 | 0.244 | 0.274 | 0.264 | 0.316 | 0.261 | 0.287 | 0.256 | 0.279 | 0.265 | 0.317 | 0.253 | 0.312 | 0.339 | 0.379 | 0.251 | 0.298 |
| **Exchange** | 0.410 | 0.431 | **0.322** | **0.384** | 0.359 | 0.402 | 0.359 | 0.422 | 0.405 | 0.437 | 0.381 | 0.412 | 0.346 | 0.414 | 0.904 | 0.695 | 0.506 | 0.500 | 0.408 | 0.423 |
| **ILI** | 2.233 | 1.015 | 2.640 | 1.018 | 4.502 | 1.557 | 2.938 | 1.178 | **2.140** | 0.907 | 2.160 | **0.901** | 4.367 | 1.540 | 4.311 | 1.396 | 3.156 | 1.207 | 4.305 | 1.397 |
| 1st Count | 11 | | 3 | | 0 | | 0 | | 1 | | 1 | | 0 | | 0 | | 0 | | 2 | |

Specifically, TSGym achieves the lowest MSE and MAE on a total of 11 occasions, reflecting its strong generalization ability over both medium and long forecasting horizons. While some baseline models, like DUET, PatchTST and SegRNN, occasionally show competitive results on certain datasets. As for short-term forecasting tasks, both TSGym and TSMixer demonstrate competitive performance, with TSGym outperforming on most evaluation metrics.

**Question 2: Is TSGym with lightweight architecture better than existing SOTA methods?**

In the previous section, we compared TSGym using the full component pool with SOTA and found that TSGym outperforms SOTA on 66.7% of the datasets evaluated. In this ablation experiment Table

4, we specifically compare the -Transformer configuration of TSGym with DUET. Remarkably, even after removing Transformer-related components from the TSGym component pool and retaining only the more computationally efficient MLP- and RNN-based models, TSGym still outperforms DUET on the majority of datasets. This demonstrates the robustness and efficiency of TSGym's architecture and highlights the strong predictive power of the simplified MLP-based design.

**Question 3: Does the training strategies bring significant improvement for TSGym?**

Following Table 4, we find that the introduction of the resampling method enhances TSGym's meta-predictor performance across 2 datasets, improving both robustness and accuracy. The +AllPL configuration, which trains on datasets with varying prediction lengths and transfers this knowledge to a test set with a single prediction length, further improves generalization, with the best performance observed on the ETTh1 dataset. Additionally, removing the Transformer component (-Transformer) leads to performance gains on certain datasets, suggesting that a simplified MLP- or RNN-based architecture can be more effective in specific scenarios. These results highlight the flexibility of TSGym's design and the potential benefits of customizing the component pool to suit dataset characteristics.

Table 4: Ablation study evaluates the removal of Transformer-based components and different training strategies, with results averaged over all prediction lengths, and the final row shows how often TSGym variants outperform DUET.

| Models | TSGym | | -Transformer | | +Resample | | +AllPL | | DUET | |
|---|---|---|---|---|---|---|---|---|---|---|
| Metric | MSE | MAE | MSE | MAE | MSE | MAE | MSE | MAE | MSE | MAE |
| ETTm1 | **0.357** | **0.383** | 0.363 | 0.388 | 0.361 | 0.384 | 0.367 | 0.391 | 0.407 | 0.409 |
| ETTm2 | 0.261 | **0.319** | 0.274 | 0.330 | **0.260** | 0.320 | 0.265 | 0.320 | 0.296 | 0.338 |
| ETTh1 | 0.426 | 0.440 | 0.441 | 0.450 | 0.432 | 0.440 | **0.424** | **0.434** | 0.433 | 0.437 |
| ETTh2 | 0.358 | 0.400 | 0.358 | 0.401 | **0.357** | **0.398** | 0.361 | 0.404 | 0.380 | 0.403 |
| ECL | **0.170** | 0.265 | 0.174 | 0.273 | 0.170 | 0.265 | 0.185 | 0.277 | 0.179 | **0.262** |
| Traffic | 0.435 | 0.313 | **0.415** | **0.293** | 0.429 | 0.307 | 0.429 | 0.306 | 0.797 | 0.427 |
| Weather | 0.229 | 0.268 | **0.226** | **0.265** | 0.229 | 0.267 | 0.234 | 0.271 | 0.252 | 0.277 |
| Exchange | 0.410 | 0.431 | 0.409 | 0.435 | 0.399 | 0.430 | 0.377 | 0.411 | **0.322** | **0.384** |
| ILI | 2.233 | **1.015** | 2.437 | 1.053 | 2.698 | 1.114 | **2.173** | 1.020 | 2.640 | 1.018 |
| **Better Count** | 14/18 | | 12/18 | | 12/18 | | 12/18 | | | |

**Question 4: Does large time-series models bring significant improvement for TSGym?**
Table 5 evaluates the impact of incorporating LLM and TSFM into the base TSGym framework. It is evident that both the introduction of LLM and TSFM consistently improve forecasting accuracy compared to the baseline TSGym configuration. This demonstrates the effectiveness of leveraging advanced model architectures and fusion strategies to further enhance the predictive performance of TSGym, particularly on complex multivariate time series datasets.

Table 5: Ablation study of TSGym incorporating LLM and TSFM in 4 datasets. The average results of all prediction lengths are listed here.

| Models | TSGym | | +LLM | | +TSFM | |
|---|---|---|---|---|---|---|
| Metric | MSE | MAE | MSE | MAE | MSE | MAE |
| ETTh1 | 0.442 | 0.440 | 0.442 | 0.438 | **0.432** | **0.437** |
| ETTh2 | 0.411 | 0.429 | **0.364** | **0.402** | 0.376 | 0.409 |
| Exchange | 0.673 | 0.522 | 0.503 | 0.466 | **0.374** | **0.411** |
| ILI | 2.682 | 1.112 | **2.470** | **1.063** | 2.860 | 1.163 |

# 5  Conclusions, Limitations, and Future Directions

To advance beyond holistic evaluations in multivariate time-series forecasting (MTSF), this paper introduced TSGym, a novel framework centered on fine-grained component analysis and the automated construction of specialized forecasting models. By systematically decomposing MTSF pipelines into design dimensions and choices informed by recent studies, TSGym uncovers crucial insights into component-level forecasting performance and leverages meta-learning method for the automated construction of customized models. Extensive experimental results indicate that the MTSF models constructed by the proposed TSGym significantly outperform current MTSF SOTA solutions—demonstrating the advantage of adaptively customizing models according to distinct data characteristics. Our results show that TSGym is highly effective, even without exhaustively covering all SOTA components, and TSGym is made publicly available to benefit the MTSF community.

Future efforts will focus on expanding TSGym's range of forecasting techniques with emerging techniques and refining its meta-learning capabilities by incorporating multi-objective optimization to balance predictive performance against computational costs, especially for large time-series models, while also broadening its applicability across diverse time series analysis tasks.

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
