# Appendix

For further details, we provide more information in the Appendix, including the evaluated 10 datasets (§A), key modules (§B), compared baselines (§C), metrics mathematical formula (§D), system configuration (§E), ADGym comparison analysis (§F), the details of proposed TSGym (§G), and additional experimental results (§H).

# A  Dataset List

We conduct extensive evaluations on nine standard long-term forecasting benchmarks - four ETT variants (ETTh1, ETTh2, ETTm1, ETTm2), Electricity (abbreviated as ECL), Traffic, Weather, Exchange, and ILI, complemented by the M4 dataset for short-term forecasting tasks, with complete dataset specifications provided in Table A1.

Table A1: Data description of the 10 datasets included in TSGym.

| Task | Dataset | Domain | Frequency | Lengths | Dim | Description |
|---|---|---|---|---|---|---|
| | ETTh1 | Electricity | 1 hour | 14,400 | 7 | Power transformer 1, comprising seven indicators such as oil temperature and useful load |
| | ETTh2 | Electricity | 1 hour | 14,400 | 7 | Power transformer 2, comprising seven indicators such as oil temperature and useful load |
| | ETTm1 | Electricity | 15 mins | 57,600 | 7 | Power transformer 1, comprising seven indicators such as oil temperature and useful load |
| | ETTm2 | Electricity | 15 mins | 57,600 | 7 | Power transformer 2, comprising seven indicators such as oil temperature and useful load |
| LTF | ECL | Electricity | 1 hour | 26,304 | 321 | Electricity records the electricity consumption in kWh every 1 hour from 2012 to 2014 |
| | Traffic | Traffic | 1 hour | 17,544 | 862 | Road occupancy rates measured by 862 sensors on San Francisco Bay area freeways |
| | Weather | Environment | 10 mins | 52,696 | 21 | Recorded every for the whole year 2020, which contains 21 meteorological indicators |
| | Exchange | Economic | 1 day | 7,588 | 8 | ExchangeRate collects the daily exchange rates of eight countries |
| | ILI | Health | 1 week | 966 | 7 | Recorded indicators of patients data from Centers for Disease Control and Prevention |
| STF | M4 | Demographic, Finance, Industry, Macro, Micro and Other | Yearly Quarterly Monthly Weekly Daily Hourly | 19-9933 | 100000 | M4 competition dataset containing 100,000 unaligned time series with varying lengths and time periods |

# B  Key Modules

Modern deep learning for MTSF utilizes several specialized modules to tackle non-stationarity, multi-scale dependencies, and inter-variable interactions. In this section, we analyze the design and efficacy of prevalent specialized modules adopted in state-of-the-art models (Fig. 1).

**Normalization modules** address temporal distribution shifts through adaptive statistical alignment. While z-score normalization employs fixed moments, modern techniques enhance adaptability: RevIN [26] introduces learnable affine transforms with reversible normalization/denormalization; Dish-TS [15] decouples inter-/intra-series distribution coefficients; Non-Stationary Transformer [35] integrates statistical moments into attention via de-stationary mechanisms. These methods balance stationarized modeling with inherent non-stationary dynamics.

**Decomposition methods**, standard in time series analysis, break down series into components like trend and seasonality to improve predictability and handle distribution shifts. **(1) Time-domain decomposition** utilizes moving average operations to isolate slowly-varying trends from high-frequency fluctuations that represent seasonality (e.g., DLinear [63], Autoformer, FEDformer). **(2) Frequency-domain decomposition** partitions series via Discrete Fourier Transform (DFT), assigning low-frequency spectra to trends and high-frequency bands to seasonality, which is applied in the Koopa [34] model.

**Multi-Scale modeling** addresses the inherent temporal hierarchy in time series data, where patterns manifest differently across various granularities (e.g., minute-level fluctuations vs. daily trends). Pyraformer [32] integrates multi-convolution kernels via pyramidal attention to establish hierarchical temporal dependencies. FEDformer [69] employs mixed experts to combine trend components from multiple pooling kernels with varying receptive fields, where larger kernels capture macro patterns while smaller ones preserve local details. TimeMixer [50] extends this paradigm through bidirectional mixing operations - upward propagation refines fine-scale seasonal features while downward aggregation consolidates coarse-scale trends. FiLM [68] dynamically adjusts temporal resolutions through learnable lookback windows, enabling adaptive focus on relevant historical contexts across scales. Crossformer [66] implements flexible patchsize configurations, where multi-granular patches independently model short-term fluctuations and long-term cycles through dimension-aware processing.

**Temporal Tokenization strategies**, originating from Transformers [51, 33] and now extended to RNNs [30], vary by temporal representation granularity: **(1) Point-wise** methods (e.g., Informer [67], Pyraformer [32]) process individual timestamps as tokens. They offer temporal precision but face quadratic complexity, requiring attention sparsification that may hinder long-range dependency capture. **(2) Patch-wise** strategies (e.g., PatchTST

[40]) aggregate local temporal segments into patches. Pathformer [10] similarly employs patch-based processing via adaptive multi-scale pathways. **(3) Series-wise** approaches (e.g., iTransformer [33]) construct global variate representations, enabling cross-variate modeling but risking temporal misalignment. TimeXer [52] uses hybrid tokenization: patch-level for endogenous variables and series-level for exogenous, bridged by a learnable global token.

**Temporal Dependency Modeling** captures dynamic inter-step dependencies through diverse architectural mechanisms, balancing local interactions and global patterns. Recurrent state transitions (e.g., LSTM) model sequential memory via gated memory cells; temporal convolutions (e.g., TCN [6]) construct multi-scale receptive fields using dilated kernels; attention mechanisms (e.g., Transformers) enable direct pairwise interactions across arbitrary time steps. Efficiency-driven innovations include sparse attention (Informer [67]), periodicity-based aggregation (Autoformer [57]), and state-space hybrids (Mamba [18]), achieving tractable long-range dependency modeling while preserving temporal fidelity.

**Variate Correlation**, fundamental to modeling critical correlations in multivariate time series forecasting (MTSF), operates through two primary paradigms [43]: **(1) Channel-Independent (CI) Strategy**: Processes channels independently with shared parameters (e.g., PatchTST [40]), ensuring robustness and efficiency but ignoring multivariate dependencies, limiting use with strong inter-channel interactions [41].

**(2) Channel-Dependent (CD) Strategy**: Integrates channel information via methods like channel-wise self-attention (iTransformer [33]) or MLP-based mixing (TSMixer [11]). This allows explicit dependency modeling but risks overfitting and struggles with noise in high dimensions.

# C    Compared Baselines

We systematically compare state-of-the-art forecasting models using the 6 architectural modules introduced in Section B. Table C2 presents the configuration of each baseline in terms of these modules. The "Notes" column provides concise annotations of each model's key methodological features, allowing for quick identification of the technical differentiators among the baselines.

# D    Metrics Mathematical Formula

The metrics used in this paper can be calculated as follows[56]:

$$\text{MSE} = \frac{1}{H}\sum_{i=1}^{H}(\mathbf{X}_i - \widehat{\mathbf{X}}_i)^2, \qquad\qquad \text{MAE} = \frac{1}{H}\sum_{i=1}^{H}|\mathbf{X}_i - \widehat{\mathbf{X}}_i|,$$

$$\text{SMAPE} = \frac{200}{H}\sum_{i=1}^{H}\frac{|\mathbf{X}_i - \widehat{\mathbf{X}}_i|}{|\mathbf{X}_i| + |\widehat{\mathbf{X}}_i|}, \qquad\qquad \text{MAPE} = \frac{100}{H}\sum_{i=1}^{H}\frac{|\mathbf{X}_i - \widehat{\mathbf{X}}_i|}{|\mathbf{X}_i|},$$

$$\text{MASE} = \frac{1}{H}\sum_{i=1}^{H}\frac{|\mathbf{X}_i - \widehat{\mathbf{X}}_i|}{\frac{1}{H-m}\sum_{j=m+1}^{H}|\mathbf{X}_j - \mathbf{X}_{j-m}|}, \qquad \text{OWA} = \frac{1}{2}\left[\frac{\text{SMAPE}}{\text{SMAPE}_{\text{Naïve2}}} + \frac{\text{MASE}}{\text{MASE}_{\text{Naïve2}}}\right],$$

where $m$ is the periodicity of the data. $\mathbf{X}, \widehat{\mathbf{X}} \in \mathbb{R}^{H \times C}$ are the ground truth and prediction results of the future with $H$ time points and $C$ dimensions. $\mathbf{X}_i$ means the $i$-th future time point.

# E    System Configuration

We conducted all experiments in the same experimental environment, which includes four NVIDIA A100 GPUs with 80GB and eight 40GB of memory. We saved overall experimental time by running experiments in parallel.

# F    Compared with ADGym

Compared with ADGym [20], TSGym exhibits the following differences and advantages:

**(1) Broader model structure design choices**. ADGym includes only MLP, autoencoder (AE), ResNet, and Transformer architectures, while TSGym provides an in-depth decoupling of different attention mechanisms within Transformers and incorporates two pre-trained large models: LLMs and TSFM. **(2) More diverse data processing design choices**. ADGym focuses solely on data augmentation and two normalization methods, whereas TSGym encompasses series sampling, series normalization, series decomposition, as well as various series encoding options. **(3) More complex meta-features**. The meta-features in ADGym include statistical metrics for tabular datasets, while TSGym considers multiple sequence characteristics across different channels in multivariate time series, such as distribution drift, sequence autocorrelation, and more. **(4) More standardized**

Table C2: Component Configurations of 27 Baseline Models

| Backbone | Method | Normali-zation | Decom-position | Multi-Scale | Token-izations | Temporal Dependency | Variate Corre-lation | Notes |
|---|---|---|---|---|---|---|---|---|
| **RNN** | SegRNN[30] | SubLast | | | Patch-wise | GRU | CI | Reduces iterations via patch-wise processing and parallel multi-step forecasting. |
| | Mamba[18] | Stat | | | Point-wise | Selective State Space Model | CD | Efficient model selectively propagating information without attention or MLP blocks. |
| **CNN** | SCINet[31] | Stat | | TRUE | Point-wise | Conv1d | CD | Recursively downsamples, convolves, and interacts with data to capture complex temporal dynamics. |
| | MICN[49] | | MA | TRUE | Point-wise | Conv1d | CD | Combines local features and global correlations using multi-scale convolutions with linear complexity. |
| | TimesNet[56] | Stat | | TRUE | Point-wise | Conv2d | CD | Transforms 1D time series into 2D tensors to capture multi-periodicity and temporal variations. |
| **MLP** | FiLM[68] | RevIN | | TRUE | Point-wise | Legendre Projection Unit | CD | Preserves historical info and reduces noise with Legendre and Fourier projections. |
| | LightTS[65] | | | | Patch-wise | MLP | CD | Lightweight MLP model for multivariate forecasting, using continuous and interval sampling for efficiency. |
| | DLinear[63] | | MA | | Point-wise | MLP | CI/CD | Decomposes series into trend and seasonal components, then applies linear layers for improved forecasting. |
| | Koopa[34] | Stat | DFT | | Point-wise, Patch-wise | MLP | CD | Uses Koopman theory to model non-stationary dynamics, handling time-variant and time-invariant components. |
| | TSMixer[11] | | | | Point-wise | MLP | CD | Simple MLP model efficiently captures both time and feature dependencies for forecasting. |
| | FreTS[60] | | | | Point-wise | Frequency-domain MLP | CI/CD | Uses frequency-domain MLPs to capture global dependencies and focus on key frequency components. |
| | TiDE[13] | Stat | | | Point-wise | MLP | CI | Fast MLP-based model for long-term forecasting, handling covariates and non-linear dependencies. |
| | TimeMixer[50] | RevIN | MA | TRUE | Point-wise | MLP | CI/CD | Fully MLP-based model, disentangles and mixes multi-scale temporal patterns. |
| **Transformer** | Reformer[1] | | | | Point-wise | LSHSelf-Attention | CD | Memory-efficient Transformer with locality-sensitive hashing for faster training on long sequences. |
| | Informer[67] | | | | Point-wise | ProbSparse-Attention | CD | Efficient Transformer with ProbSparse-Attention and a generative decoder for faster long-sequence forecasting. |
| | TFT[29] | Stat | | | Point-wise | Self-Attention | CD | High-performance, interpretable multi-horizon forecasting model combining recurrent layers for local processing and attention layers for long-term dependencies. |
| | Autoformer[57] | | MA | | Point-wise | Auto-Correlation | CD | Uses Auto-Correlation and decomposition for accurate long-term predictions. |
| | PyraFormer[32] | | | TRUE | Point-wise | Pyramid-Attention | CD | Captures temporal dependencies at multiple resolutions with constant signal path length. |
| | NSTransformer[35] | Stat | | | Point-wise | De-stationary Attention | CD | Restores non-stationary information through de-stationary attention for improved forecasting. |
| | ETSformer[55] | | DFT | | Point-wise | Exponential-Smoothing-Attention | CD | Integrates exponential smoothing and frequency attention for accuracy, efficiency, and interpretability. |
| | FEDformer[69] | | MA | TRUE | Point-wise | AutoCorrelation | CD | Combines seasonal-trend decomposition with frequency-enhanced Transformer for efficient forecasting. |
| | Crossformer[66] | | | TRUE | Patch-wise | TwoStage-Attention | CD | Captures both temporal and cross-variable dependencies with two-stage attention. |
| | PatchTST[40] | Stat | | | Patch-wise | FullAttention | CI | Segments time series into patches and uses channel-independent embeddings. |
| | iTransformer[33] | Stat | | | Series-wise | FullAttention | CD | Redefines token embedding to treat time points as series-wise tokens for better multivariate modeling. |
| | TimeXer[52] | Stat | | | Series-wise | FullAttention | CD | Enhances forecasting by incorporating exogenous variables via patch-wise and variate-wise attention. |
| | PAttn[47] | Stat | | | Patch-wise | FullAttention | CI | Similar to PatchTST, uses attention-based patching for efficient forecasting without large language models. |
| | DUET[43] | RevIN | MA | | Point-wise | FullAttention | CI/CD | Enhances multivariate forecasting by using Mixture of Experts (MOE) for temporal clustering and a frequency-domain similarity mask matrix for channel clustering. |

Table F3: Compared with ADGym, TSGym covers a broader and more in-depth design space, as well as a more structured and extensive automated selection experiment.

| | ADGym | TSGym |
|---|---|---|
| Design Dimensions | 13 | 16 |
| Design Space Size | 195,9552 | 796,2624 |
| Model Architectures | MLP,AE,ResNet,FTTransformer | MLP,RNN, Transformers, LLM, TSFM |
| Max of Data Samples | 3000 | 57,600 |
| Meta Feature Dimensions | 200 | 1404 |
| Baseline Methods | 7 | 27 |

**automated selection experiments**. Due to time constraints, ADGym limits the sample size to fewer than 3000 samples, whereas TSGym imposes no such restriction, providing a larger-scale experimental design that leads to more solid experimental conclusions.

In summary, compared with ADGym, TSGym makes **significant progress and development in both components benchmarking and automated selection.** More details can be seen in table F3.

# G  Details of TSGym

In this section, we introduce detailed descriptions of the design choices, extracted meta-features and the trained meta-predictors.

## G.1  More Details of Design Choices in TSGym.

We selected competitive components from the key modules of existing state-of-the-art (SOTA) works as our design choices. In Appx. B, we introduced the underlying principles of these components according to different design dimensions. While most of the individual components have demonstrated their effectiveness through ablation studies in their respective original papers, the interactions and synergies among them when combined have never been systematically explored. Notably, when assembling complete pipelines from different design choices, we automatically exclude incompatible combinations, such as pairing MLP-based architectures with diverse series attention modules.

## G.2  Meta-features and Meta-predictors

**Details and the selected list of meta-features**. All meta-features in this paper integrate two complementary perspectives: (1) static characteristics extracted via TSFEL [42] spanning temporal, statistical, spectral, and fractal domains, and (2) dynamic behavioral metrics from TFB [7] to quantify temporal distribution shifts. In Section 4.2, we present the results of the meta-predictor trained on meta-features derived from static characteristics, which corresponds to the default setting in TSGym. Furthermore, in Fig. G1, we visualize the dimension-reduced meta-features across different datasets. In Table H10, we report the performance of the meta-predictor under various meta-feature configurations. The following categorizes these features with their analytical purposes (see Tables G4–G7 for implementation details):

- **Temporal features** (Table G4): Characterize sequential dynamics through trend detection, entropy analysis, and change-point statistics, preserving sensitivity to temporal ordering.

- **Statistical features** (Table G5): Capture distribution properties via central tendency (mean/median), dispersion (variance/IQR), and shape descriptors (skewness/kurtosis), invariant to observation order.

- **Spectral features** (Table G6): Decompose signals into frequency components using Fourier/wavelet transforms, identifying dominant periodicities and hidden oscillations.

- **Fractal features** (Table G7): Quantify multiscale complexity through fractal dimensions and Hurst exponents, reflecting self-similarity patterns across temporal resolutions.

- **Shifting Metric**: To complement static features, this TFB-derived metric measures temporal distribution drift via KL-divergence between adjacent windows. Values approaching 1 indicate severe shifts caused by external perturbations or systemic transitions, providing a diagnostic tool for non-stationary dynamics.

**Details of the trained meta-predictors**. For each design choice, we first use the LabelEncoder class from scikit-learn to convert it into a numerical class index. This index is then fed into an $nn.Embedding$ layer within our model to obtain a dense vector representation. These learned embeddings, along with other meta-features, subsequently form the input to the meta-predictor. The meta-predictor is optimized using Pearson loss to learn the relative performance ranks of different design choices, thereby emphasizing the linear correlation between predicted and actual rankings.

Moreover, we experimented with different training strategies to guide the meta-predictor in selecting the top-1 design pipelines.

(1) **+Resample**: Constraining the number of combinations from different datasets to be equal when training the meta-predictor.

(2) **+AllPL**: Training on datasets with varying prediction lengths and transfers this knowledge to a test set with a single prediction length.

(3) We train the meta-predictor using diverse meta features, including those generated by segmenting the datasets based on timestamps (**Sub**), those combining information from different time periods (**Whole**), and those designed to capture distributional shifts (**Delta**). The symbol **"+"** denotes the concatenation of multiple meta features.

We report the results of **+Resample** and **+AllPL** in Table 4, and the results of diverse meta-features in Table H10.

## Table G4: **Temporal** Meta-feature Specifications

| Feature | Description | Functionality |
|---|---|---|
| Absolute Energy | Computes the absolute energy of the signal. | Measures the total energy of the signal, often used to understand signal power and activity levels. |
| Area Under the Curve | Computes the area under the curve of the signal computed with the trapezoid rule. | Provides a measure of the overall signal amplitude or ""energy"" over time. |
| Autocorrelation | Calculates the first 1/e crossing of the autocorrelation function (ACF). | Measures the correlation of the signal with its own past values, useful for identifying repeating patterns. |
| Average Power | Computes the average power of the signal. | Averages the squared values of the signal, capturing its power over time. |
| Centroid | Computes the centroid along the time axis. | Indicates the ""center"" or ""balance point"" of the signal in time, providing insight into its distribution. |
| Signal Distance | Computes signal traveled distance. | Measures the total path length covered by the signal over time, capturing the extent of signal fluctuations. |
| Negative Turning | Computes number of negative turning points of the signal. | Counts the number of times the signal changes direction from positive to negative. |
| Neighbourhood Peaks | Computes the number of peaks from a defined neighbourhood of the signal. | Identifies the number of peak points within a specified window, useful for pattern detection. |
| Peak-to-Peak Distance | Computes the peak to peak distance. | Measures the time interval between successive peaks, indicating the period of oscillations. |
| Positive Turning | Computes number of positive turning points of the signal. | Counts the number of times the signal changes direction from negative to positive. |
| Root Mean Square | Computes root mean square of the signal. | Calculates the square root of the average squared values of the signal, often used as a measure of signal strength. |
| Slope | Computes the slope of the signal. | Measures the rate of change in the signal's amplitude over time, indicating trends or shifts. |
| Sum of Absolute Differences | Computes sum of absolute differences of the signal. | Measures the total variation in the signal by summing the absolute differences between consecutive values. |
| Zero-Crossing Rate | Computes Zero-crossing rate of the signal. | Counts how many times the signal crosses the zero axis, indicating its frequency and periodicity. |

## Table G5: **Statistical** Meta-feature Specifications

| Feature | Description | Functionality |
|---|---|---|
| Maximum Value | Computes the maximum value of the signal. | Identifies the highest amplitude or peak value in the signal, useful for determining extreme values. |
| Mean Value | Computes mean value of the signal. | Calculates the average value of the signal, providing insight into its central tendency. |
| Median | Computes the median of the signal. | Finds the middle value of the signal when sorted, offering robustness to outliers. |
| Minimum Value | Computes the minimum value of the signal. | Identifies the lowest amplitude or trough value in the signal, useful for detecting minima. |
| Standard Deviation | Computes standard deviation (std) of the signal. | Measures the variation or spread of the signal values, indicating how much the signal deviates from the mean. |
| Variance | Computes variance of the signal. | Quantifies the spread of signal values, related to the square of the standard deviation. |
| Empirical Cumulative Distribution Function | Computes the values of ECDF along the time axis. | Provides a cumulative distribution function, representing the probability distribution of the signal values. |
| ECDF Percentile | Computes the percentile value of the ECDF. | Extracts specific percentiles from the cumulative distribution, useful for understanding the signal's quantiles. |
| ECDF Percentile Count | Computes the cumulative sum of samples that are less than the percentile. | Measures the number of samples falling below a given percentile, providing distribution insights. |
| ECDF Slope | Computes the slope of the ECDF between two percentiles. | Measures the steepness or rate of change in the cumulative distribution, indicating distribution sharpness. |
| Histogram Mode | Compute the mode of a histogram using a given number of bins. | Finds the most frequent value in the signal's histogram, representing the peak of the signal's distribution. |
| Interquartile Range | Computes interquartile range of the signal. | Measures the range between the 25th and 75th percentiles, indicating the spread of the central 50% of the signal values. |
| Kurtosis | Computes kurtosis of the signal. | Measures the ""tailedness"" of the signal distribution, indicating the presence of outliers or extreme values. |
| Mean Absolute Deviation | Computes mean absolute deviation of the signal. | Measures the average deviation of the signal values from the mean, providing an indication of signal variability. |
| Mean Absolute Difference | Computes mean absolute differences of the signal. | Calculates the average of absolute differences between successive signal values, reflecting the signal's smoothness. |
| Mean Difference | Computes mean of differences of the signal. | Computes the average of the first-order differences, used to measure overall signal change. |
| Median Absolute Deviation | Computes median absolute deviation of the signal. | Measures the spread of the signal values around the median, offering a robust measure of variability. |
| Median Absolute Difference | Computes median absolute differences of the signal. | Similar to mean absolute difference but based on the median, used to assess signal smoothness. |
| Median Difference | Computes median of differences of the signal. | Calculates the median of first-order differences, providing insights into signal trend stability. |
| Skewness | Computes skewness of the signal. | Measures the asymmetry of the signal's distribution, indicating whether it is skewed towards higher or lower values. |

## Table G6: **Spectral** Meta-feature Specifications

| Feature | Description | Functionality |
| --- | --- | --- |
| Entropy | Computes the entropy of the signal using Shannon Entropy. | Quantifies the uncertainty or randomness in the signal, offering insights into its complexity. |
| Fundamental Frequency | Computes the fundamental frequency of the signal. | Identifies the primary frequency at which the signal oscillates, crucial for detecting periodic behaviors. |
| Human Range Energy | Computes the human range energy ratio. | Measures the energy in the human audible range, useful for identifying signals relevant to human hearing. |
| Linear Prediction Cepstral Coefficients | Computes the linear prediction cepstral coefficients. | Extracts features related to the signal's frequency components, commonly used in speech and audio processing. |
| Maximum Frequency | Computes maximum frequency of the signal. | Identifies the highest frequency component of the signal, providing insight into its frequency range. |
| Maximum Power Spectrum | Computes maximum power spectrum density of the signal. | Measures the peak value in the power spectral density, identifying dominant frequencies in the signal. |
| Median Frequency | Computes median frequency of the signal. | Identifies the frequency that divides the signal's power spectrum into two equal halves. |
| Mel-Frequency Cepstral Coefficients | Computes the MEL cepstral coefficients. | Used to extract features representing the spectral characteristics of the signal, primarily used in speech analysis. |
| Multiscale Entropy | Computes the Multiscale entropy (MSE) of the signal, that performs entropy analysis over multiple scales. | Quantifies the signal's complexity at different scales, useful for detecting non-linear temporal behaviors. |
| Power Bandwidth | Computes power spectrum density bandwidth of the signal. | Measures the width of the frequency band where the majority of the signal's power is concentrated. |
| Spectral Centroid | Barycenter of the spectrum. | Identifies the ""center"" of the signal's frequency spectrum, used in sound and audio analysis. |
| Spectral Decrease | Represents the amount of decreasing of the spectra amplitude. | Measures how rapidly the spectral amplitude decreases across frequency, useful for identifying spectral roll-off. |
| Spectral Distance | Computes the signal spectral distance. | Quantifies the difference between the signal's spectrum and a reference, helpful in pattern recognition. |
| Spectral Entropy | Computes the spectral entropy of the signal based on Fourier transform. | Measures the randomness or complexity in the frequency domain of the signal. |
| Spectral Kurtosis | Measures the flatness of a distribution around its mean value. | Quantifies the tail heaviness of the signal's frequency distribution, identifying outliers or abnormal distributions. |
| Spectral Positive Turning | Computes number of positive turning points of the fft magnitude signal. | Counts the points where the signal's Fourier transform changes direction from negative to positive. |
| Spectral Roll-Off | Computes the spectral roll-off of the signal. | Measures the frequency below which a specified percentage of the total spectral energy is contained. |
| Spectral Roll-On | Computes the spectral roll-on of the signal. | Similar to roll-off but identifies the frequency above which a specified amount of energy is concentrated. |
| Spectral Skewness | Measures the asymmetry of a distribution around its mean value. | Measures the skew in the signal's frequency distribution, highlighting the presence of spectral biases. |
| Spectral Slope | Computes the spectral slope. | Quantifies the slope of the power spectral density, often used to distinguish between harmonic and non-harmonic signals. |
| Spectral Spread | Measures the spread of the spectrum around its mean value. | Measures the dispersion or spread of the signal's spectral energy. |
| Spectral Variation | Computes the amount of variation of the spectrum along time. | Quantifies how much the frequency content of the signal changes over time. |
| Spectrogram Mean Coefficients | Calculates the average power spectral density (PSD) for each frequency throughout the entire signal. | Averages the power spectral density across all time intervals, capturing the signal's overall spectral energy distribution. |
| Wavelet Absolute Mean | Computes CWT absolute mean value of each wavelet scale. | Measures the average wavelet transform magnitude across scales, useful for detecting changes in signal frequency. |
| Wavelet Energy | Computes CWT energy of each wavelet scale. | Quantifies the energy at each wavelet scale, reflecting the signal's energy distribution across frequencies. |
| Wavelet Entropy | Computes CWT entropy of the signal. | Measures the complexity or unpredictability of the signal at different wavelet scales. |
| Wavelet Standard Deviation | Computes CWT std value of each wavelet scale. | Measures the variation or spread of the wavelet transform across different scales. |
| Wavelet Variance | Computes CWT variance value of each wavelet scale. | Quantifies the dispersion of the signal at different wavelet scales. |

## Table G7: **Fractal** Meta-feature Specifications

| Feature | Description | Functionality |
| --- | --- | --- |
| Detrended Fluctuation Analysis | Computes the Detrended Fluctuation Analysis (DFA) of the signal. | Measures long-range correlations and self-similarity in the signal, used for identifying fractal behavior. |
| Higuchi Fractal Dimension | Computes the fractal dimension of a signal using Higuchi's method (HFD). | Measures the complexity of the signal's pattern by calculating its fractal dimension. |
| Hurst Exponent | Computes the Hurst exponent of the signal through the Rescaled range (R/S) analysis. | Measures the long-term memory or persistence in the signal, useful for identifying trends and randomness. |
| Lempel-Ziv Complexity | Computes the Lempel-Ziv's (LZ) complexity index, normalized by the signal's length. | Quantifies the randomness or predictability of the signal based on its compressibility. |
| Maximum Fractal Length | Computes the Maximum Fractal Length (MFL) of the signal. | Measures the fractal dimension at the smallest scale of the signal, reflecting its intricate pattern complexity. |
| Petrosian Fractal Dimension | Computes the Petrosian Fractal Dimension of a signal. | Measures the signal's fractal dimension based on its variation across different scales. |

Table H8: Full results for the long-term forecasting task. All the results are averaged from 4 different prediction lengths, that is $\{24, 36, 48, 60\}$ for ILI and $\{96, 192, 336, 720\}$ for the others.

| Models | TSGym (Ours) | | DUET [43] | | TimeMixer [50] | | MICN [49] | | TimesNet [56] | | PatchTST [40] | | DLinear [63] | | Crossformer [66] | | Autoformer [57] | | SegRNN [30] | | Mamba [18] | | iTransformer [33] | | TimeXer [52] | |
|---|---|---|---|---|---|---|---|---|---|---|---|---|---|---|---|---|---|---|---|---|---|---|---|---|---|---|
| Metric | MSE | MAE | MSE | MAE | MSE | MAE | MSE | MAE | MSE | MAE | MSE | MAE | MSE | MAE | MSE | MAE | MSE | MAE | MSE | MAE | MSE | MAE | MSE | MAE | MSE | MAE |
| ETTm1 | **0.360** | **0.384** | 0.407 | 0.409 | 0.384 | 0.399 | 0.402 | 0.429 | 0.432 | 0.430 | 0.390 | 0.404 | 0.404 | 0.407 | 0.501 | 0.501 | 0.532 | 0.496 | 0.388 | 0.404 | 0.501 | 0.466 | 0.414 | 0.415 | 0.386 | 0.400 |
| ETTm2 | 0.265 | **0.322** | 0.296 | 0.338 | 0.277 | 0.325 | 0.342 | 0.391 | 0.296 | 0.334 | 0.288 | 0.334 | 0.349 | 0.399 | 1.487 | 0.789 | 0.356 | 0.370 | 0.273 | 0.322 | 0.356 | 0.370 | 0.290 | 0.332 | 0.279 | 0.325 |
| ETTh1 | 0.425 | 0.434 | 0.433 | 0.437 | 0.448 | 0.438 | 0.589 | 0.537 | 0.474 | 0.464 | 0.454 | 0.449 | 0.465 | 0.461 | 0.544 | 0.520 | 0.492 | 0.485 | **0.422** | **0.429** | 0.544 | 0.504 | 0.462 | 0.452 | 0.446 | 0.443 |
| ETTh2 | **0.371** | 0.406 | 0.380 | 0.403 | 0.383 | 0.406 | 0.585 | 0.530 | 0.415 | 0.424 | 0.385 | 0.409 | 0.566 | 0.520 | 1.552 | 0.908 | 0.446 | 0.460 | 0.374 | 0.405 | 0.465 | 0.448 | 0.382 | 0.406 | 0.372 | **0.399** |
| ECL | **0.179** | 0.275 | 0.179 | **0.262** | 0.185 | 0.273 | 0.186 | 0.297 | 0.219 | 0.314 | 0.209 | 0.298 | 0.225 | 0.319 | 0.193 | 0.289 | 0.234 | 0.340 | 0.216 | 0.302 | 0.209 | 0.312 | 0.190 | 0.277 | 0.191 | 0.286 |
| Traffic | **0.434** | **0.310** | 0.797 | 0.427 | 0.496 | 0.313 | 0.544 | 0.320 | 0.645 | 0.348 | 0.497 | 0.321 | 0.673 | 0.419 | 1.458 | 0.782 | 0.637 | 0.397 | 0.807 | 0.411 | 0.679 | 0.380 | 0.474 | 0.318 | 0.509 | 0.333 |
| Weather | **0.229** | **0.267** | 0.252 | 0.277 | 0.244 | 0.274 | 0.264 | 0.316 | 0.261 | 0.287 | 0.256 | 0.279 | 0.265 | 0.317 | 0.253 | 0.312 | 0.339 | 0.379 | 0.251 | 0.298 | 0.291 | 0.315 | 0.259 | 0.280 | 0.243 | 0.273 |
| Exchange | 0.392 | 0.418 | **0.322** | **0.384** | 0.359 | 0.402 | 0.346 | 0.422 | 0.405 | 0.437 | 0.381 | 0.412 | 0.346 | 0.414 | 0.904 | 0.695 | 0.506 | 0.500 | 0.408 | 0.423 | 0.714 | 0.562 | 0.369 | 0.410 | 0.410 | 0.424 |
| ILI | 2.345 | 1.053 | 2.640 | 1.018 | 4.502 | 1.557 | 2.938 | 1.178 | 2.140 | 0.907 | 2.160 | **0.901** | 4.367 | 1.540 | 4.311 | 1.396 | 3.156 | 1.207 | 4.305 | 1.397 | 3.729 | 1.335 | 2.305 | 0.974 | 2.633 | 1.034 |
| 1st Count | 9 | | 3 | | 0 | | 0 | | 0 | | 1 | | 0 | | 0 | | 0 | | 2 | | 0 | | 0 | | 1 | |

| Models | PAttn [47] | | Koopa [34] | | TSMixer [11] | | FreTS [60] | | Pyraformer [32] | | Nonstationary [36] | | ETSformer [55] | | FEDformer [69] | | SCINet [31] | | LightTS [65] | | Informer [67] | | Transformer [48] | | Reformer [1] | |
|---|---|---|---|---|---|---|---|---|---|---|---|---|---|---|---|---|---|---|---|---|---|---|---|---|---|---|
| Metric | MSE | MAE | MSE | MAE | MSE | MAE | MSE | MAE | MSE | MAE | MSE | MAE | MSE | MAE | MSE | MAE | MSE | MAE | MSE | MAE | MSE | MAE | MSE | MAE | MSE | MAE |
| ETTm1 | 0.384 | 0.399 | 0.367 | 0.396 | 0.527 | 0.512 | 0.409 | 0.417 | 0.695 | 0.593 | 0.509 | 0.467 | 0.636 | 0.592 | 0.438 | 0.450 | 0.409 | 0.412 | 0.438 | 0.445 | 0.969 | 0.736 | 0.836 | 0.678 | 0.998 | 0.723 |
| ETTm2 | 0.291 | 0.336 | **0.264** | 0.327 | 1.030 | 0.750 | 0.336 | 0.378 | 1.565 | 0.876 | 0.412 | 0.398 | 1.381 | 0.807 | 0.301 | 0.348 | 0.294 | 0.335 | 0.432 | 0.448 | 1.454 | 0.851 | 1.856 | 0.996 | | |
| ETTh1 | 0.468 | 0.454 | 0.472 | 0.471 | 0.615 | 0.579 | 0.476 | 0.464 | 0.814 | 0.692 | 0.610 | 0.543 | 0.750 | 0.651 | 0.448 | 0.461 | 0.520 | 0.488 | 0.530 | 0.505 | 1.057 | 0.798 | 0.930 | 0.768 | 0.973 | 0.739 |
| ETTh2 | 0.386 | 0.412 | 0.388 | 0.423 | 2.160 | 1.220 | 0.548 | 0.514 | 3.776 | 1.557 | 0.552 | 0.505 | 0.572 | 0.534 | 0.427 | 0.446 | 0.428 | 0.440 | 0.633 | 0.551 | 4.535 | 1.745 | 2.976 | 1.369 | 2.487 | 1.238 |
| ECL | 0.205 | 0.286 | 0.219 | 0.319 | 0.229 | 0.337 | 0.209 | 0.296 | 0.295 | 0.387 | 0.194 | 0.296 | 0.275 | 0.370 | 0.225 | 0.336 | 0.220 | 0.323 | 0.243 | 0.344 | 0.369 | 0.444 | 0.273 | 0.367 | 0.324 | 0.404 |
| Traffic | 0.513 | 0.328 | 0.595 | 0.413 | 0.599 | 0.403 | 0.597 | 0.377 | 0.697 | 0.391 | 0.642 | 0.351 | 1.035 | 0.584 | 0.615 | 0.379 | 0.654 | 0.419 | 0.656 | 0.428 | 0.830 | 0.464 | 0.708 | 0.384 | 0.694 | 0.380 |
| Weather | 0.257 | 0.280 | 0.230 | 0.271 | 0.242 | 0.301 | 0.255 | 0.299 | 0.284 | 0.349 | 0.289 | 0.312 | 0.365 | 0.424 | 0.315 | 0.369 | 0.256 | 0.283 | 0.245 | 0.295 | 0.572 | 0.523 | 0.599 | 0.531 | 0.475 | 0.472 |
| Exchange | 0.365 | 0.407 | 0.610 | 0.516 | 0.487 | 0.546 | 0.442 | 0.453 | 1.183 | 0.855 | 0.557 | 0.490 | 0.361 | 0.416 | 0.520 | 0.502 | 0.374 | 0.418 | 0.486 | 0.493 | 1.548 | 0.997 | 1.379 | 0.921 | 1.612 | 1.044 |
| ILI | 2.359 | 0.975 | **2.064** | 0.912 | 5.617 | 1.680 | 3.447 | 1.279 | 4.691 | 1.442 | 2.592 | 1.012 | 4.046 | 1.419 | 3.088 | 1.214 | 6.505 | 1.853 | 7.078 | 1.975 | 5.035 | 1.539 | 4.682 | 1.448 | 4.211 | 1.350 |
| 1st Count | 0 | | 2 | | 0 | | 0 | | 0 | | 0 | | 0 | | 0 | | 0 | | 0 | | 0 | | 0 | | 0 | |

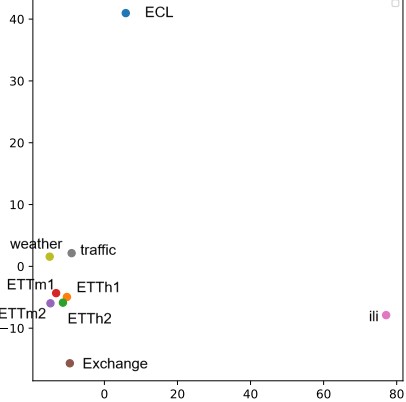

(a) PCA projection of meta-features for 9 long-term forecasting datasets

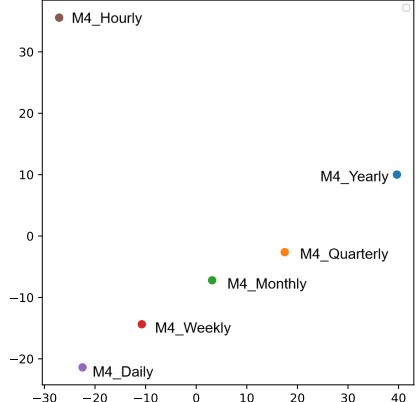

(b) PCA projection of meta-features for 6 short-term forecasting datasets

Figure G1: Distributions of meta-features after PCA dimensionality reduction, comparing datasets for long-term and short-term time series forecasting tasks.

## H  Additional Experimental Results

### H.1  Comprehensive Benchmarking of TSGym Against State-of-the-Art Methods

Due to space limitations in the main text, here we provide complete experimental comparisons for both long-term and short-term forecasting tasks. Table H.1 details the full long-term forecasting performance across all prediction horizons, while Table H.1 presents the comprehensive short-term forecasting results. Following standard benchmarking conventions, we highlight top-performing methods in **red** and second-best results with underlined formatting. These extensive evaluations consistently validate TSGym's competitive performance across diverse temporal prediction scenarios. In addition, we investigate the impact of different meta-feature configurations through controlled ablation studies. As demonstrated in Table H10, no individual meta-feature configuration exhibits consistent superiority across all datasets.

### H.2  Additional Results of Large Evaluations on Design Choices

To systematically evaluate our architectural decisions, we conduct detailed ablation studies focusing on 17 component-level analyses, presented separately in Tables H11–H12 for clarity and due to space constraints. These comparative experiments assess the performance impact of different design choices for each component across nine datasets in the long-term forecasting task. **Bolded** values indicate the best-performing configuration for each dataset, while the summary row highlights the most frequently superior design choices, with **red-bolded**

Table H9: Full results for the short-term forecasting task in the M4 dataset. *. in the Transformers indicates the name of ∗former.

| | Models | TSGym (Ours) | TimeMixer [50] | MICN [49] | TimesNet [56] | PatchTST [40] | DLinear [63] | Cross. [66] | Auto. [57] | SegRNN [30] | Mamba [18] | iTrans. [33] | TimeXer [52] | PAttn [47] | TSMixer [11] | FreTS [60] | Pyra. [32] | ETS. [55] | FED. [69] | SCINet [31] | LightTS [65] | In. [67] | Trans. [48] | Re. [1] | TiDE [13] | FiLM [68] |
|---|---|---|---|---|---|---|---|---|---|---|---|---|---|---|---|---|---|---|---|---|---|---|---|---|---|---|
| Yearly | OWA | 0.779 | 0.789 | 0.873 | 0.784 | 0.798 | 0.843 | 4.407 | 1.019 | 0.858 | 0.790 | 0.807 | 0.797 | 0.823 | 0.798 | 0.800 | 0.883 | 0.982 | 0.806 | 0.801 | 0.795 | 0.887 | 4.406 | 0.829 | 0.807 | 0.806 |
| | sMAPE | 13.286 | 13.467 | 14.586 | 13.344 | 13.606 | 14.402 | 71.464 | 17.294 | 14.323 | 13.388 | 13.681 | 13.551 | 13.893 | 13.539 | 13.579 | 14.967 | 16.105 | 13.631 | 13.573 | 13.516 | 15.086 | 69.405 | 14.064 | 14.006 | 13.988 |
| | MASE | 2.960 | 3.000 | 3.392 | 2.983 | 3.033 | 3.198 | 17.649 | 3.897 | 3.335 | 3.021 | 3.090 | 3.043 | 3.162 | 3.051 | 3.056 | 3.377 | 3.888 | 3.088 | 3.065 | 3.030 | 3.382 | 18.144 | 3.166 | 3.011 | 3.007 |
| Quarterly | OWA | 0.891 | 0.911 | 1.025 | 0.886 | 0.975 | 0.928 | 8.208 | 1.290 | 1.009 | 0.912 | 1.050 | 0.942 | 0.897 | 0.903 | 0.908 | 1.002 | 1.312 | 0.940 | 0.922 | 0.886 | 1.248 | 8.190 | 1.007 | 0.959 | 0.960 |
| | sMAPE | 10.232 | 10.313 | 11.427 | 10.063 | 10.975 | 10.500 | 74.297 | 14.085 | 11.193 | 10.320 | 11.752 | 10.536 | 10.203 | 10.218 | 10.329 | 11.259 | 13.600 | 10.655 | 10.426 | 10.166 | 13.644 | 73.944 | 11.324 | 10.719 | 10.742 |
| | MASE | 1.169 | 1.215 | 1.388 | 1.176 | 1.306 | 1.238 | 13.260 | 1.784 | 1.374 | 1.217 | 1.416 | 1.272 | 1.189 | 1.204 | 1.205 | 1.345 | 1.906 | 1.252 | 1.229 | 1.163 | 1.723 | 13.256 | 1.352 | 1.295 | 1.296 |
| Monthly | OWA | 0.863 | 0.895 | 0.986 | 0.939 | 1.020 | 0.936 | 7.637 | 1.369 | 1.074 | 0.931 | 0.982 | 0.944 | 0.951 | 0.898 | 0.915 | 1.043 | 1.281 | 1.001 | 0.924 | 0.884 | 1.151 | 7.668 | 1.448 | 0.942 | 0.942 |
| | sMAPE | 12.570 | 12.823 | 13.798 | 13.314 | 14.156 | 13.384 | 68.873 | 18.132 | 15.052 | 13.152 | 13.737 | 13.254 | 13.421 | 12.865 | 13.059 | 14.666 | 15.449 | 14.112 | 13.146 | 12.717 | 15.806 | 69.992 | 18.782 | 13.381 | 13.352 |
| | MASE | 0.909 | 0.959 | 1.080 | 1.015 | 1.126 | 1.004 | 11.165 | 1.576 | 1.175 | 1.010 | 1.076 | 1.030 | 1.034 | 0.962 | 0.984 | 1.138 | 1.586 | 1.090 | 0.996 | 0.943 | 1.283 | 11.149 | 1.694 | 1.017 | 1.019 |
| Weekly | OWA | 0.983 | 1.266 | 1.500 | 1.310 | 1.035 | 1.461 | 28.636 | 1.575 | 0.998 | 1.449 | 1.424 | 1.179 | 1.124 | 1.525 | 1.286 | 1.313 | 1.024 | 1.094 | 1.438 | 1.340 | 1.537 | 28.093 | 1.473 | 1.451 | 1.280 |
| | sMAPE | 9.467 | 11.555 | 11.790 | 11.569 | 9.546 | 11.805 | 198.371 | 12.727 | 9.149 | 12.495 | 12.050 | 10.455 | 10.326 | 12.364 | 11.394 | 11.742 | 9.363 | 9.635 | 12.757 | 12.060 | 11.967 | 191.424 | 11.522 | 12.425 | 11.539 |
| | MASE | 2.593 | 3.529 | 4.760 | 3.770 | 2.857 | 4.534 | 98.925 | 4.892 | 2.771 | 4.262 | 4.254 | 3.379 | 3.111 | 4.723 | 3.688 | 3.732 | 2.848 | 3.155 | 4.122 | 3.789 | 4.911 | 98.015 | 4.690 | 4.295 | 3.609 |
| Daily | OWA | 0.982 | 1.040 | 1.245 | 1.079 | 1.018 | 1.090 | 48.627 | 1.418 | 0.988 | 1.130 | 1.191 | 1.047 | 1.011 | 1.230 | 1.088 | 1.111 | 1.061 | 0.998 | 1.082 | 1.086 | 1.235 | 29.620 | 1.496 | 1.106 | 1.082 |
| | sMAPE | 3.005 | 3.162 | 3.786 | 3.294 | 3.103 | 3.319 | 179.226 | 4.248 | 3.009 | 3.417 | 3.607 | 3.198 | 3.089 | 3.677 | 3.304 | 3.398 | 3.216 | 3.060 | 3.288 | 3.313 | 3.727 | 99.709 | 4.521 | 3.342 | 3.267 |
| | MASE | 3.205 | 3.418 | 4.089 | 3.530 | 3.332 | 3.577 | 125.892 | 4.722 | 3.237 | 3.732 | 3.928 | 3.423 | 3.303 | 4.109 | 3.580 | 3.626 | 3.497 | 3.247 | 3.552 | 3.557 | 4.086 | 86.873 | 4.941 | 3.653 | 3.580 |
| Hourly | OWA | 0.902 | 1.372 | 1.526 | 3.171 | 2.625 | 1.040 | 11.691 | 1.623 | 1.704 | 0.730 | 1.214 | 1.636 | 1.683 | 1.243 | 1.393 | 2.315 | 1.201 | 1.126 | 1.372 | 1.183 | 3.166 | 6.498 | 3.204 | 1.231 | 1.445 |
| | sMAPE | 18.203 | 19.994 | 24.631 | 34.626 | 29.980 | 17.260 | 128.419 | 25.407 | 34.523 | 14.944 | 19.573 | 21.244 | 23.431 | 19.809 | 20.805 | 26.112 | 19.751 | 18.858 | 24.196 | 21.382 | 34.038 | 99.324 | 34.755 | 20.828 | 21.088 |
| | MASE | 1.947 | 3.966 | 4.100 | 10.680 | 8.667 | 2.732 | 39.269 | 4.465 | 3.663 | 1.549 | 3.266 | 5.067 | 5.009 | 3.372 | 3.964 | 7.685 | 3.178 | 2.937 | 3.420 | 2.881 | 10.730 | 18.188 | 10.821 | 3.183 | 4.172 |
| Average | OWA | 0.856 | 0.884 | 0.984 | 0.907 | 0.965 | 0.922 | 8.856 | 1.273 | 1.007 | 0.903 | 0.969 | 0.918 | 0.915 | 0.894 | 0.897 | 1.005 | 1.209 | 0.942 | 0.906 | 0.875 | 1.127 | 8.039 | 1.209 | 0.925 | 0.924 |
| | sMAPE | 11.781 | 11.985 | 13.025 | 12.199 | 12.848 | 12.511 | 76.147 | 16.392 | 13.509 | 12.120 | 12.838 | 12.268 | 12.351 | 12.023 | 12.137 | 13.478 | 14.635 | 12.708 | 12.219 | 11.925 | 14.673 | 72.619 | 15.344 | 12.489 | 12.471 |
| | MASE | 1.551 | 1.615 | 1.839 | 1.662 | 1.738 | 1.693 | 18.440 | 2.317 | 1.823 | 1.651 | 1.762 | 1.677 | 1.680 | 1.657 | 1.645 | 1.844 | 2.284 | 1.695 | 1.657 | 1.605 | 2.042 | 16.805 | 2.136 | 1.674 | 1.673 |
| | 1st Count | 14 | 0 | 0 | 2 | 0 | 0 | 0 | 0 | 1 | 3 | 0 | 0 | 0 | 0 | 0 | 0 | 0 | 0 | 0 | 1 | 0 | 0 | 0 | 0 | 0 |

Table H10: Effects of different meta feature settings on the long-term forecasting task. All metric values are averaged across different prediction lengths. For more details about meta features, refer to section G.2.

| Models | Whole+Sub+Delta | | Sub | | Sub+Delta | | Delta | | Whole(default) | | DUET | |
|---|---|---|---|---|---|---|---|---|---|---|---|---|
| Metric | mse | mae | mse | mae | mse | mae | mse | mae | mse | mae | mse | mae |
| ETTm1 | 0.357 | 0.383 | 0.363 | 0.388 | **0.354** | **0.380** | 0.377 | 0.405 | 0.357 | 0.383 | 0.407 | 0.409 |
| ETTm2 | 0.273 | 0.329 | 0.269 | 0.329 | 0.266 | 0.324 | 0.380 | 0.389 | **0.261** | **0.319** | 0.296 | 0.338 |
| ETTh1 | **0.417** | **0.429** | 0.433 | 0.435 | 0.418 | 0.433 | 0.558 | 0.496 | 0.426 | 0.440 | 0.433 | 0.437 |
| ETTh2 | 0.375 | 0.407 | 0.362 | 0.402 | 0.360 | **0.398** | 1.256 | 0.744 | **0.358** | 0.400 | 0.380 | 0.403 |
| ECL | 0.172 | 0.266 | 0.171 | 0.269 | 0.176 | 0.270 | 0.182 | 0.278 | **0.170** | 0.265 | 0.179 | **0.262** |
| Traffic | 0.433 | 0.309 | 0.437 | 0.313 | **0.432** | **0.308** | 0.587 | 0.368 | 0.435 | 0.313 | 0.797 | 0.427 |
| Weather | 0.239 | 0.274 | **0.228** | **0.266** | 0.233 | 0.270 | 0.263 | 0.310 | 0.229 | 0.268 | 0.252 | 0.277 |
| Exchange | 0.406 | 0.429 | 0.408 | 0.429 | 0.404 | 0.428 | 0.761 | 0.622 | 0.410 | 0.431 | **0.322** | **0.384** |
| ILI | 2.401 | 1.030 | 3.099 | 1.195 | 2.855 | 1.141 | 2.814 | 1.125 | **2.233** | **1.015** | 2.640 | 1.018 |

Table H11: Long-term Forecasting Performance of Different Design Choices – Part I (6 Components). Performance of various configurations for 6 Components across multiple datasets, evaluated using best MSE, median, and IQR. **Bolded** entries indicate the best-performing hyperparameter for each dataset. The last row shows the number of times each configuration achieved the best result, with **red-bolded** values highlighting the most frequently superior design.

| dataset | stat | x_mark False | x_mark True | multi-granularity False | multi-granularity True | Normalization DishTS | Normalization None | Normalization RevIN | Normalization Stat | Decomposition DFT | Decomposition MA | Decomposition MoEMA | Decomposition None | Channel-independent False | Channel-independent True | Tokenization inverted-encoding | Tokenization series-encoding | Tokenization series-patching |
|---|---|---|---|---|---|---|---|---|---|---|---|---|---|---|---|---|---|---|
| ETTm1 | Best | 0.352 | 0.35 | 0.349 | 0.352 | 0.36 | 0.362 | 0.351 | 0.353 | 0.351 | 0.352 | 0.354 | 0.354 | 0.354 | 0.35 | 0.354 | 0.352 | 0.351 |
| | Median | 0.423 | 0.452 | 0.428 | 0.455 | 0.528 | 0.583 | 0.406 | 0.405 | 0.454 | 0.406 | 0.45 | 0.476 | 0.469 | 0.389 | 0.404 | 0.485 | 0.384 |
| | IQR | 0.145 | 0.179 | 0.155 | 0.186 | 0.212 | 0.235 | 0.097 | 0.097 | 0.159 | 0.131 | 0.169 | 0.179 | 0.195 | 0.107 | 0.132 | 0.207 | 0.092 |
| ETTm2 | Best | 0.255 | 0.255 | 0.254 | 0.255 | 0.272 | 0.277 | 0.253 | 0.255 | 0.257 | 0.256 | 0.258 | 0.256 | 0.259 | 0.253 | 0.259 | 0.257 | 0.254 |
| | Median | 0.367 | 0.367 | 0.356 | 0.406 | 0.76 | 1.07 | 0.297 | 0.3 | 0.353 | 0.381 | 0.384 | 0.38 | 0.408 | 0.307 | 0.305 | 0.452 | 0.307 |
| | IQR | 0.579 | 0.782 | 0.428 | 0.947 | 0.746 | 1.267 | 0.045 | 0.036 | 0.552 | 0.769 | 0.681 | 0.51 | 0.871 | 0.154 | 0.226 | 0.963 | 0.177 |
| ETTh1 | Best | 0.401 | 0.408 | 0.403 | 0.405 | 0.44 | 0.433 | 0.404 | 0.402 | 0.407 | 0.41 | 0.405 | 0.411 | 0.412 | 0.401 | 0.412 | 0.414 | 0.401 |
| | Median | 0.491 | 0.491 | 0.484 | 0.51 | 0.545 | 0.632 | 0.468 | 0.462 | 0.494 | 0.489 | 0.495 | 0.486 | 0.511 | 0.462 | 0.478 | 0.52 | 0.456 |
| | IQR | 0.125 | 0.126 | 0.098 | 0.183 | 0.207 | 0.371 | 0.05 | 0.05 | 0.141 | 0.098 | 0.157 | 0.118 | 0.202 | 0.042 | 0.062 | 0.241 | 0.037 |
| ETTh2 | Best | 0.326 | 0.334 | 0.339 | 0.326 | 0.41 | 0.402 | 0.325 | 0.337 | 0.338 | 0.327 | 0.344 | 0.339 | 0.327 | 0.341 | 0.348 | 0.325 | 0.347 |
| | Median | 0.444 | 0.467 | 0.449 | 0.455 | 1.188 | 1.764 | 0.392 | 0.397 | 0.43 | 0.547 | 0.547 | 0.532 | 0.394 | 0.495 | 0.468 | 0.388 | |
| | IQR | 0.939 | 0.732 | 0.65 | 1.74 | 1.657 | 2.855 | 0.044 | 0.048 | 0.494 | 1.035 | 1.105 | 0.726 | 1.513 | 0.189 | 0.418 | 1.924 | 0.217 |
| ECL | Best | 0.159 | 0.157 | 0.157 | 0.159 | 0.159 | 0.16 | 0.159 | 0.157 | 0.158 | 0.163 | 0.161 | 0.157 | 0.157 | 0.163 | 0.157 | 0.158 | 0.164 |
| | Median | 0.208 | 0.204 | 0.204 | 0.208 | 0.219 | 0.229 | 0.191 | 0.191 | 0.206 | 0.209 | 0.206 | 0.202 | 0.207 | 0.202 | 0.194 | 0.213 | 0.19 |
| | IQR | 0.056 | 0.057 | 0.058 | 0.055 | 0.053 | 0.059 | 0.035 | 0.052 | 0.065 | 0.053 | 0.054 | 0.056 | 0.057 | 0.055 | 0.051 | 0.061 | 0.051 |
| traffic | Best | 0.394 | 0.396 | 0.398 | 0.394 | 0.411 | 0.441 | 0.398 | 0.394 | 0.398 | 0.4 | 0.394 | 0.4 | 0.394 | 0.409 | 0.399 | 0.394 | 0.409 |
| | Median | 0.555 | 0.599 | 0.576 | 0.581 | 0.546 | 0.657 | 0.542 | 0.496 | 0.607 | 0.575 | 0.56 | 0.561 | 0.568 | 0.627 | 0.531 | 0.6 | 0.607 |
| | IQR | 0.19 | 0.199 | 0.208 | 0.183 | 0.164 | 0.12 | 0.207 | 0.195 | 0.185 | 0.19 | 0.203 | 0.198 | 0.196 | 0.195 | 0.19 | 0.191 | 0.183 |
| weather | Best | 0.223 | 0.22 | 0.22 | 0.222 | 0.225 | 0.225 | 0.22 | 0.224 | 0.226 | 0.223 | 0.22 | 0.221 | 0.22 | 0.22 | 0.22 | 0.22 | 0.223 |
| | Median | 0.261 | 0.272 | 0.259 | 0.274 | 0.267 | 0.301 | 0.255 | 0.256 | 0.265 | 0.27 | 0.261 | 0.263 | 0.273 | 0.246 | 0.247 | 0.281 | 0.246 |
| | IQR | 0.041 | 0.079 | 0.047 | 0.065 | 0.059 | 0.21 | 0.033 | 0.04 | 0.047 | 0.053 | 0.052 | 0.055 | 0.062 | 0.035 | 0.034 | 0.073 | 0.033 |
| Exchange | Best | 0.245 | 0.237 | 0.239 | 0.242 | 0.24 | 0.25 | 0.351 | 0.337 | 0.243 | 0.239 | 0.246 | 0.242 | 0.24 | 0.238 | 0.24 | 0.245 | 0.238 |
| | Median | 0.493 | 0.502 | 0.462 | 0.548 | 0.674 | 0.93 | 0.432 | 0.415 | 0.472 | 0.519 | 0.495 | 0.507 | 0.569 | 0.394 | 0.415 | 0.582 | 0.395 |
| | IQR | 0.43 | 0.471 | 0.434 | 0.491 | 0.869 | 0.855 | 0.164 | 0.168 | 0.386 | 0.546 | 0.43 | 0.466 | 0.595 | 0.144 | 0.253 | 0.613 | 0.149 |
| ili | Best | 1.596 | 1.546 | 1.562 | 1.576 | 1.763 | 2.351 | 1.584 | 1.555 | 1.673 | 1.599 | 1.581 | 1.573 | 1.545 | 1.745 | 1.583 | 1.548 | 1.745 |
| | Median | 2.813 | 2.883 | 2.881 | 2.797 | 2.785 | 4.416 | 2.486 | 2.493 | 2.878 | 2.784 | 2.859 | 2.883 | 2.796 | 3.043 | 2.865 | 2.844 | 2.819 |
| | IQR | 1.621 | 1.698 | 1.656 | 1.681 | 1.152 | 0.975 | 0.742 | 0.764 | 1.612 | 1.596 | 1.701 | 1.739 | 1.665 | 1.693 | 1.757 | 1.68 | 1.442 |
| 1st Count | | **20** | 7 | **22** | 5 | 1 | 1 | **16** | 9 | 8 | **9** | 5 | 5 | 8 | **19** | 5 | 3 | **19** |

entries denoting the dominant configurations. This fine-grained analysis offers empirical insights to guide component selection in time-series forecasting systems.

Table H12: Long-term Forecasting Performance of Different Design Choices – Part II (4 Components) and Part II (7 Components). Same structure and evaluation metrics as Table H11.

(a) Part II – 4 Components (Backbone, Attention, etc.)

| dataset | stat | Backbone | | | Attention | | | | | | Feature-Attention | | | | Sequence Length | | | |
|---|---|---|---|---|---|---|---|---|---|---|---|---|---|---|---|---|---|---|
| | | GRU | MLP | Trans-former | auto-cor-relation | de-stationary-attention | frequency-enhanced-attention | null | self-attention | sparse-attention | frequency-enhanced-attention | null | self-attention | sparse-attention | 192 | 48 | 512 | 96 |
| ETTm1 | Best | 0.352 | 0.352 | **0.351** | 0.359 | 0.382 | 0.354 | **0.35** | 0.359 | 0.354 | 0.356 | **0.35** | 0.355 | 0.36 | 0.352 | 0.476 | **0.349** | 0.38 |
| | Median | 0.457 | **0.411** | 0.449 | 0.499 | 0.455 | **0.409** | 0.437 | 0.486 | 0.441 | 0.453 | **0.408** | 0.472 | 0.462 | **0.386** | 0.545 | 0.395 | 0.423 |
| | IQR | 0.157 | 0.179 | **0.151** | 0.242 | **0.087** | 0.106 | 0.164 | 0.189 | 0.143 | 0.182 | **0.146** | 0.171 | 0.202 | 0.106 | 0.089 | 0.121 | **0.082** |
| ETTm2 | Best | 0.264 | **0.255** | 0.256 | 0.258 | 0.289 | 0.265 | **0.255** | 0.26 | 0.267 | 0.26 | **0.253** | 0.26 | 0.262 | 0.263 | 0.293 | **0.253** | 0.274 |
| | Median | 0.407 | **0.34** | 0.416 | 0.663 | **0.32** | 0.335 | 0.356 | 0.437 | 0.766 | 0.534 | **0.326** | 0.398 | 0.389 | **0.34** | 0.403 | 0.341 | 0.415 |
| | IQR | 0.565 | **0.382** | 0.854 | 0.9 | **0.033** | 0.911 | 0.441 | 0.711 | 1.086 | 0.906 | **0.281** | 0.836 | 0.86 | 0.836 | **0.429** | 0.922 | 0.753 |
| ETTh1 | Best | 0.411 | **0.402** | 0.406 | 0.418 | 0.471 | 0.413 | **0.401** | 0.432 | 0.406 | 0.42 | **0.401** | 0.417 | 0.412 | 0.422 | 0.445 | **0.401** | 0.435 |
| | Median | 0.496 | **0.48** | 0.502 | 0.51 | 0.526 | **0.475** | 0.487 | 0.527 | 0.504 | 0.495 | **0.481** | 0.505 | 0.513 | 0.487 | 0.498 | **0.482** | 0.49 |
| | IQR | 0.115 | **0.09** | 0.167 | 0.19 | **0.06** | 0.07 | 0.105 | 0.207 | 0.214 | 0.149 | **0.078** | 0.137 | 0.244 | 0.128 | **0.114** | 0.156 | 0.114 |
| ETTh2 | Best | **0.327** | 0.339 | 0.344 | 0.356 | 0.383 | 0.35 | **0.325** | 0.36 | 0.355 | 0.347 | 0.334 | **0.329** | 0.346 | 0.351 | 0.384 | **0.325** | 0.361 |
| | Median | 0.516 | **0.427** | 0.453 | 0.462 | **0.41** | 0.546 | 0.445 | 0.589 | 0.504 | 0.544 | **0.421** | 0.449 | 0.587 | **0.422** | 0.473 | 0.438 | 0.531 |
| | IQR | 0.642 | **0.617** | 1.453 | 2.021 | **0.03** | 0.754 | 0.65 | 1.345 | 1.393 | 2.209 | **0.305** | 1.086 | 1.483 | 0.64 | 0.842 | 1.394 | 0.721 |
| ECL | Best | 0.163 | 0.163 | **0.157** | 0.163 | 0.165 | 0.16 | 0.162 | 0.158 | **0.157** | **0.158** | 0.158 | 0.159 | 0.158 | 0.162 | 0.181 | **0.157** | 0.169 |
| | Median | 0.214 | 0.205 | **0.201** | 0.205 | **0.181** | 0.207 | 0.209 | 0.199 | 0.195 | 0.194 | 0.213 | 0.199 | 0.209 | 0.183 | 0.241 | **0.182** | 0.209 |
| | IQR | 0.056 | 0.06 | **0.054** | 0.054 | **0.048** | 0.055 | 0.06 | 0.048 | 0.052 | 0.052 | 0.06 | **0.051** | 0.054 | **0.025** | 0.044 | 0.047 | 0.041 |
| traffic | Best | 0.409 | 0.408 | **0.394** | 0.407 | 0.417 | 0.401 | 0.407 | **0.394** | 0.399 | 0.407 | 0.399 | **0.394** | 0.402 | 0.409 | 0.515 | **0.394** | 0.446 |
| | Median | 0.585 | 0.608 | **0.558** | 0.576 | **0.475** | 0.583 | 0.596 | 0.596 | 0.523 | 0.539 | 0.654 | **0.507** | 0.537 | 0.476 | 0.685 | **0.453** | 0.578 |
| | IQR | **0.179** | 0.215 | 0.195 | 0.208 | **0.102** | 0.181 | 0.198 | 0.199 | 0.19 | 0.149 | **0.138** | 0.195 | 0.163 | 0.135 | **0.126** | 0.181 | 0.144 |
| weather | Best | 0.222 | **0.221** | 0.221 | 0.227 | **0.21** | 0.229 | 0.22 | 0.226 | 0.221 | 0.227 | **0.22** | 0.222 | 0.223 | 0.227 | 0.253 | **0.22** | 0.239 |
| | Median | **0.264** | 0.268 | 0.266 | 0.279 | **0.233** | 0.264 | 0.265 | 0.274 | 0.25 | 0.272 | **0.254** | 0.27 | 0.282 | 0.248 | 0.286 | **0.242** | 0.258 |
| | IQR | 0.049 | **0.047** | 0.054 | 0.065 | **0.018** | 0.047 | 0.049 | 0.043 | 0.06 | 0.048 | 0.048 | **0.044** | 0.095 | 0.04 | 0.035 | 0.063 | **0.031** |
| Exchange | Best | 0.24 | **0.238** | 0.256 | 0.269 | 0.406 | 0.278 | **0.237** | 0.263 | 0.282 | 0.251 | **0.238** | 0.248 | 0.247 | 0.274 | 0.24 | 0.294 | **0.238** |
| | Median | 0.537 | **0.435** | 0.574 | 0.602 | 0.615 | 0.545 | **0.473** | 0.56 | 0.59 | 0.536 | **0.443** | 0.574 | 0.557 | 0.503 | **0.401** | 0.81 | 0.42 |
| | IQR | 0.445 | **0.366** | 0.517 | 0.499 | **0.164** | 0.6 | 0.433 | 0.492 | 0.492 | 0.604 | **0.302** | 0.56 | 0.506 | 0.349 | **0.226** | 0.822 | 0.245 |
| ili | Best | 1.619 | 1.561 | **1.551** | 1.597 | 1.665 | 1.672 | **1.561** | 1.637 | 1.642 | 1.603 | 1.629 | 1.63 | **1.552** | 1.878 | 1.715 | 2.269 | **1.546** |
| | Median | 2.953 | 2.851 | **2.761** | 2.731 | **2.451** | 2.949 | 2.889 | 2.791 | 2.728 | **2.646** | 3.043 | 2.815 | 2.755 | 2.622 | 2.705 | 3.799 | **2.487** |
| | IQR | 1.816 | **1.516** | 1.661 | 1.623 | **0.656** | 1.652 | 1.676 | 1.746 | 1.642 | **1.55** | 1.747 | 1.605 | 1.672 | **1.069** | 1.616 | 1.691 | 1.694 |
| 1st Count | | 3 | **15** | 9 | 0 | **16** | 2 | 7 | 1 | 1 | 4 | **17** | 5 | 1 | 6 | 5 | **11** | 5 |

(b) Part III – 7 Components (d_model, d_ff, etc.)

| dataset | stat | d_model | | d_ff | | Encoder layers | | Training Epochs | | | Loss Function | | | Learning Rate | | Learning Rate Strategy | |
|---|---|---|---|---|---|---|---|---|---|---|---|---|---|---|---|---|---|
| | | 256 | 64 | 1024 | 256 | 2 | 3 | 10 | 20 | 50 | HUBER | MAE | MSE | 0.0001 | 0.001 | null | type |
| ETTm1 | Best | 0.352 | **0.35** | 0.352 | **0.35** | 0.352 | **0.35** | 0.352 | **0.351** | 0.353 | 0.359 | 0.356 | **0.35** | **0.35** | 0.351 | 0.355 | **0.35** |
| | Median | 0.462 | **0.423** | 0.462 | **0.423** | **0.424** | 0.451 | 0.46 | 0.433 | **0.428** | 0.437 | **0.433** | 0.442 | **0.425** | 0.448 | 0.431 | 0.446 |
| | IQR | 0.166 | **0.153** | 0.166 | **0.153** | **0.154** | 0.156 | 0.173 | 0.163 | **0.15** | **0.113** | 0.139 | 0.166 | **0.151** | 0.175 | **0.141** | 0.176 |
| ETTm2 | Best | 0.256 | **0.253** | 0.256 | **0.253** | **0.254** | 0.255 | 0.256 | 0.255 | **0.254** | 0.266 | 0.261 | **0.253** | 0.255 | **0.253** | 0.256 | **0.253** |
| | Median | 0.395 | **0.35** | 0.395 | **0.35** | 0.367 | **0.363** | **0.352** | 0.381 | 0.376 | **0.353** | 0.45 | 0.37 | 0.359 | 0.373 | 0.379 | **0.355** |
| | IQR | 0.792 | **0.435** | 0.792 | **0.435** | 0.785 | **0.535** | **0.585** | 0.767 | 0.607 | **0.66** | 0.681 | 0.665 | 0.575 | 0.729 | 0.62 | 0.783 |
| ETTh1 | Best | **0.401** | 0.408 | **0.401** | 0.408 | 0.407 | **0.401** | **0.402** | 0.406 | 0.406 | 0.408 | **0.401** | 0.417 | **0.401** | 0.408 | **0.402** | 0.404 |
| | Median | **0.491** | 0.491 | **0.491** | 0.491 | **0.49** | 0.492 | 0.493 | 0.49 | **0.489** | 0.489 | **0.487** | 0.498 | **0.477** | 0.503 | **0.49** | 0.493 |
| | IQR | **0.117** | 0.134 | **0.117** | 0.134 | **0.114** | 0.132 | 0.119 | 0.156 | **0.113** | 0.137 | 0.125 | **0.104** | **0.107** | 0.134 | **0.111** | 0.141 |
| ETTh2 | Best | 0.338 | **0.326** | 0.338 | **0.326** | **0.325** | 0.341 | 0.336 | 0.342 | **0.325** | 0.341 | **0.326** | 0.336 | 0.337 | **0.325** | 0.337 | **0.325** |
| | Median | 0.474 | **0.441** | 0.474 | **0.441** | **0.442** | 0.467 | 0.462 | **0.442** | 0.447 | **0.415** | 0.425 | 0.471 | **0.446** | 0.451 | 0.461 | **0.445** |
| | IQR | 0.77 | 0.871 | 0.77 | 0.871 | 0.902 | **0.752** | 1.286 | 0.806 | **0.577** | **0.59** | 0.928 | 0.901 | **0.756** | 0.956 | 0.969 | **0.739** |
| ECL | Best | **0.157** | 0.16 | **0.157** | 0.16 | 0.158 | **0.157** | 0.159 | 0.159 | **0.157** | 0.158 | 0.159 | **0.157** | **0.157** | 0.158 | **0.157** | 0.158 |
| | Median | **0.204** | 0.207 | **0.204** | 0.207 | 0.209 | **0.202** | **0.205** | 0.205 | 0.207 | 0.208 | **0.193** | 0.206 | 0.216 | **0.199** | **0.198** | 0.213 |
| | IQR | 0.057 | **0.056** | 0.057 | **0.056** | 0.058 | **0.054** | 0.057 | **0.056** | 0.057 | 0.055 | **0.048** | 0.057 | 0.062 | **0.051** | **0.05** | 0.06 |
| traffic | Best | **0.394** | 0.4 | **0.394** | 0.4 | 0.4 | **0.394** | 0.401 | 0.401 | **0.394** | 0.418 | 0.423 | **0.394** | 0.405 | **0.394** | 0.398 | **0.394** |
| | Median | **0.548** | 0.604 | **0.548** | 0.604 | **0.566** | 0.587 | 0.59 | 0.585 | **0.562** | 0.627 | 0.62 | **0.57** | 0.594 | **0.56** | **0.551** | 0.604 |
| | IQR | **0.195** | 0.206 | **0.195** | 0.206 | **0.194** | 0.197 | 0.209 | 0.194 | **0.19** | 0.144 | 0.164 | **0.195** | 0.216 | **0.189** | **0.186** | 0.212 |
| weather | Best | **0.22** | 0.22 | **0.22** | 0.22 | **0.22** | 0.22 | 0.224 | **0.22** | 0.22 | 0.223 | 0.225 | **0.22** | 0.222 | **0.22** | 0.221 | **0.22** |
| | Median | 0.27 | **0.26** | 0.27 | **0.26** | 0.266 | **0.265** | 0.268 | **0.261** | 0.267 | 0.277 | **0.255** | 0.266 | **0.261** | 0.27 | **0.263** | 0.268 |
| | IQR | 0.052 | **0.049** | 0.052 | **0.049** | 0.053 | 0.05 | 0.053 | **0.048** | 0.051 | 0.129 | **0.039** | 0.05 | **0.044** | 0.055 | **0.046** | 0.056 |
| Exchange | Best | **0.237** | 0.243 | **0.237** | 0.243 | 0.244 | **0.239** | 0.244 | **0.239** | 0.246 | **0.241** | 0.249 | 0.241 | 0.24 | **0.238** | 0.244 | **0.238** |
| | Median | 0.528 | **0.465** | 0.528 | **0.465** | 0.5 | **0.494** | 0.503 | 0.494 | **0.489** | 0.494 | **0.442** | 0.509 | **0.444** | 0.551 | 0.513 | **0.486** |
| | IQR | 0.518 | **0.407** | 0.518 | **0.407** | 0.495 | **0.424** | 0.488 | **0.408** | 0.451 | 0.449 | **0.412** | 0.468 | **0.38** | 0.545 | 0.495 | **0.433** |
| ili | Best | **1.546** | 1.632 | **1.546** | 1.632 | 1.564 | **1.553** | 1.586 | **1.553** | 1.613 | **1.582** | 1.59 | 1.585 | 1.662 | **1.545** | 1.591 | **1.563** |
| | Median | **2.737** | 2.971 | **2.737** | 2.971 | **2.846** | 2.854 | 2.9 | **2.805** | 2.85 | 2.9 | 2.933 | **2.801** | 3.161 | **2.636** | **2.679** | 3.203 |
| | IQR | **1.599** | 1.732 | **1.599** | 1.732 | **1.635** | 1.683 | 1.713 | **1.611** | 1.694 | 1.677 | 1.657 | **1.653** | 1.78 | **1.47** | **1.438** | 1.819 |
| 1st Count | | **14** | 13 | **14** | 13 | 12 | **15** | 4 | 11 | **12** | 8 | **10** | 9 | **15** | 12 | **15** | 12 |

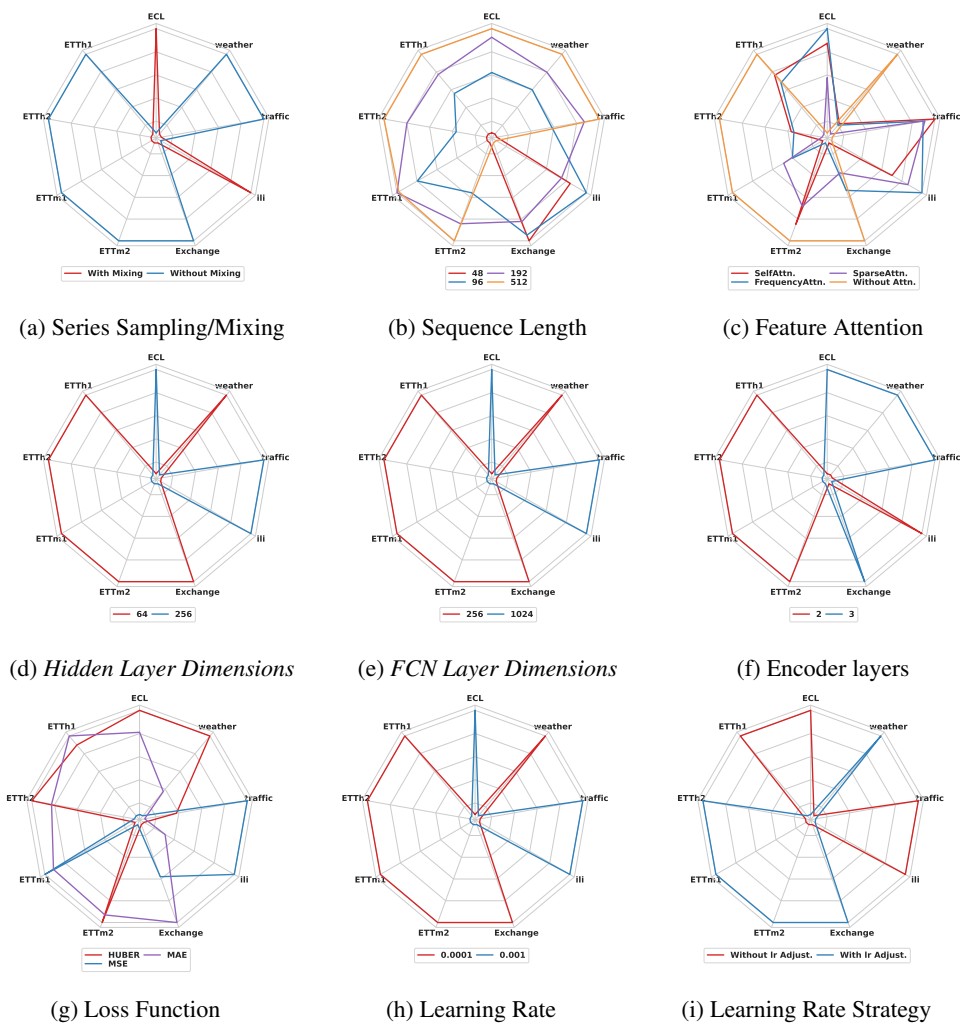

(a) Series Sampling/Mixing     (b) Sequence Length     (c) Feature Attention

(d) *Hidden Layer Dimensions*     (e) *FCN Layer Dimensions*     (f) Encoder layers

(g) Loss Function     (h) Learning Rate     (i) Learning Rate Strategy

Figure H2: Overall performance across additional design dimensions in long-term forecasting. The results (MSE) are based on the 75th percentile across all forecasting horizons.

### H.2.1 Design Choices Evaluation Results for Long-term Forecasting Using MSE as the Metric

**Spider Chart Analysis**. Fig. H2 extends the baseline comparisons presented in Fig. 2 by employing multi-dimensional spider charts, where each vertex corresponds to a benchmark dataset. Closer proximity to the outer edge of a vertex indicates better performance of the associated design choice on that particular dataset. These visual representations offer an intuitive understanding of how different architectural decisions influence model effectiveness across diverse forecasting domains. Notably, configurations for components including Series Sampling/Mixing (Fig. H2a), Hidden Layer Dimensions (Fig. H2d), FCN Layer Dimensions (Fig. H2e), Learning Rate (Fig. H2h), and Learning Rate Strategy (Fig. H2i) demonstrate similar spatial patterns in the radar charts. Specifically, ECL, ILI, and Traffic datasets exhibit consistent parameter preferences across these components, suggesting intrinsic alignment between their temporal patterns and specific architectural configurations.

In addition, Fig. H3 provides a broader evaluation of large-scale time series models, revealing that conventional architectures still maintain a competitive advantage over LLM-based models, especially in domain-specific forecasting tasks where structural inductive biases play a crucial role.

**Box Plots Analysis**. The impact of various design choices for each architectural component is further illustrated through box plots in Fig. H4 and Fig. H5. These visualizations complement the spider charts by providing a statistical perspective on performance variability and robustness across multiple benchmark datasets. Together, the two forms of analysis offer a comprehensive view of how different configurations affect forecasting accuracy.

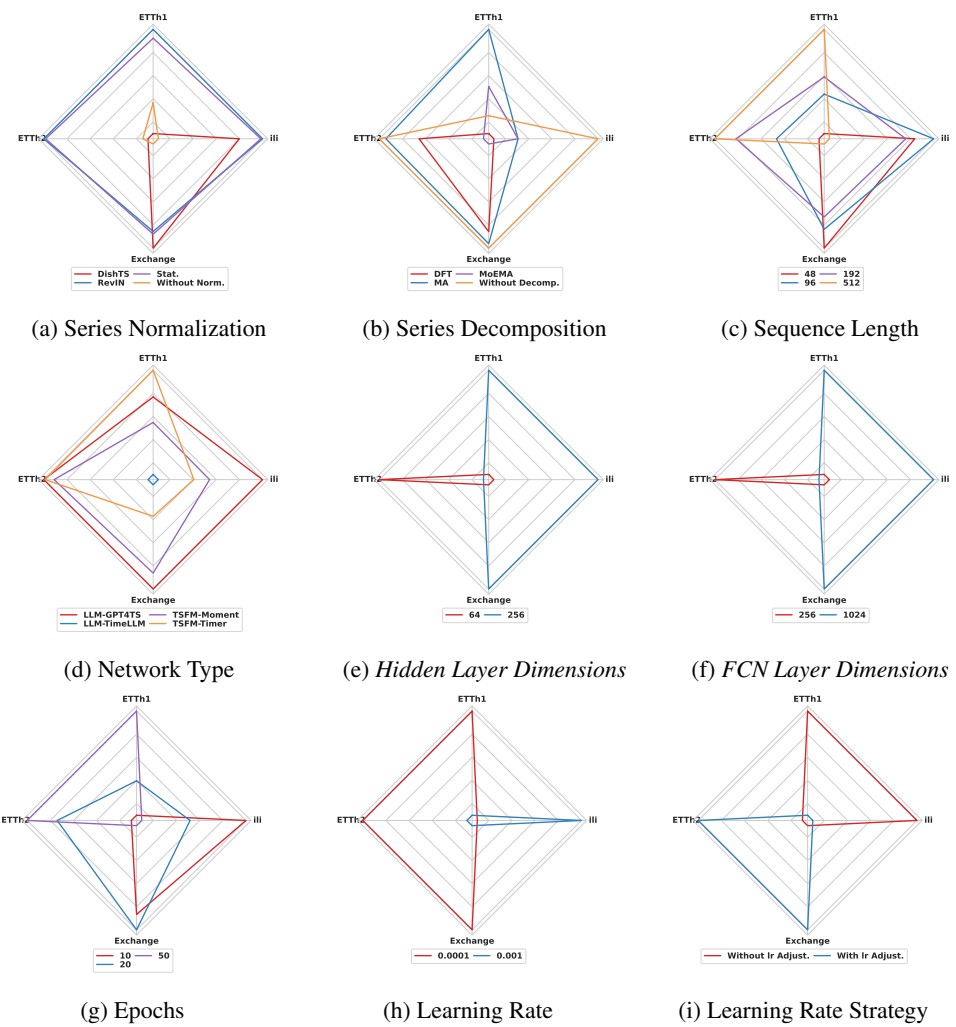

(a) Series Normalization     (b) Series Decomposition     (c) Sequence Length

(d) Network Type     (e) *Hidden Layer Dimensions*     (f) *FCN Layer Dimensions*

(g) Epochs     (h) Learning Rate     (i) Learning Rate Strategy

Figure H3: Overall performance across all design dimensions when using LLMs or TSFMs in long-term forecasting. The results (MSE) are based on the 75th percentile across all forecasting horizons.

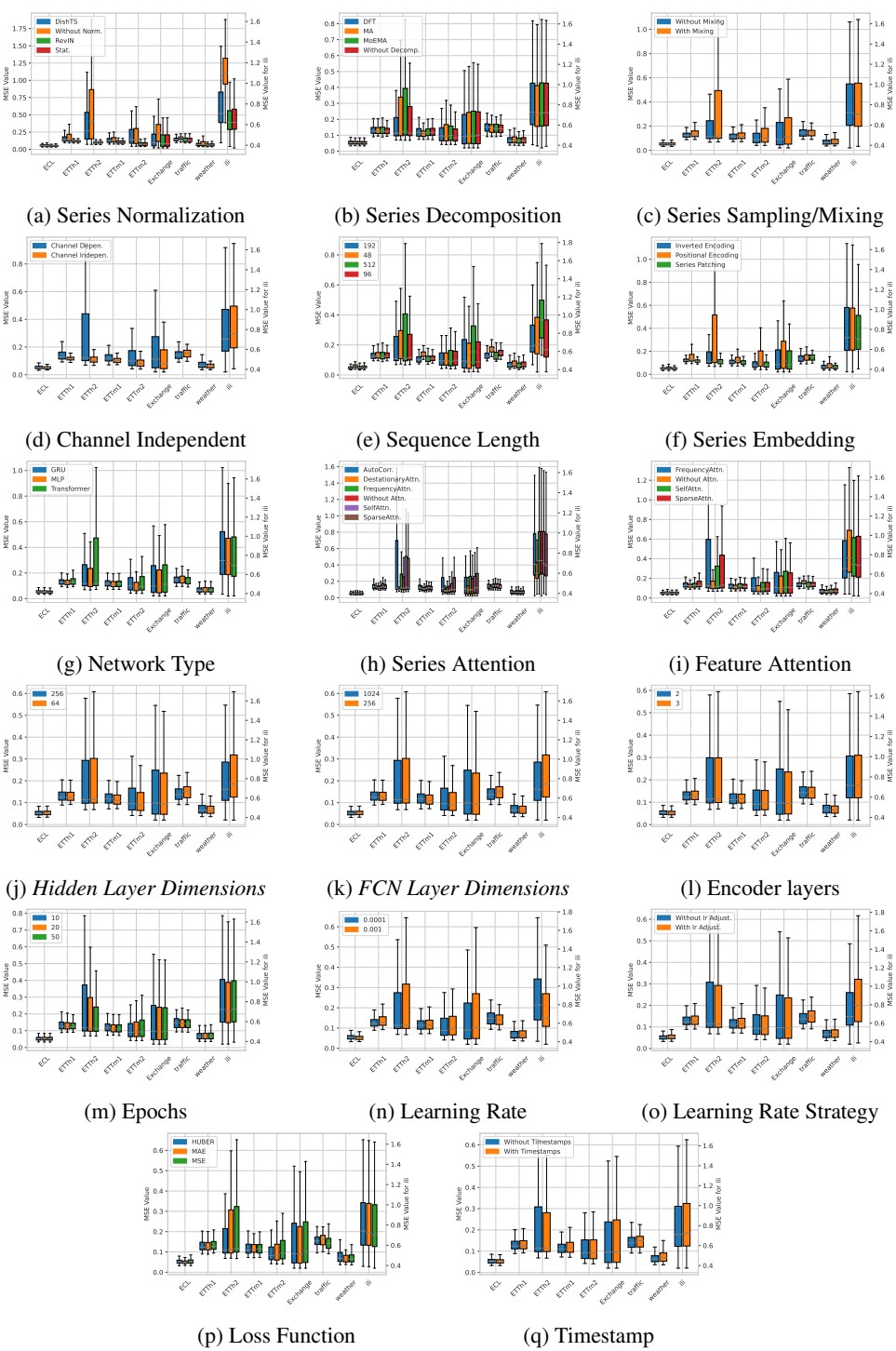

Figure H4: Overall performance across all design dimensions in long-term forecasting. The results (**MSE**) are averaged across all forecasting horizons. Due to the significantly different value range and variability of the ILI dataset compared to other datasets, its box plot is plotted using the right-hand *y*-axis, while all other datasets share the left-hand *y*-axis.

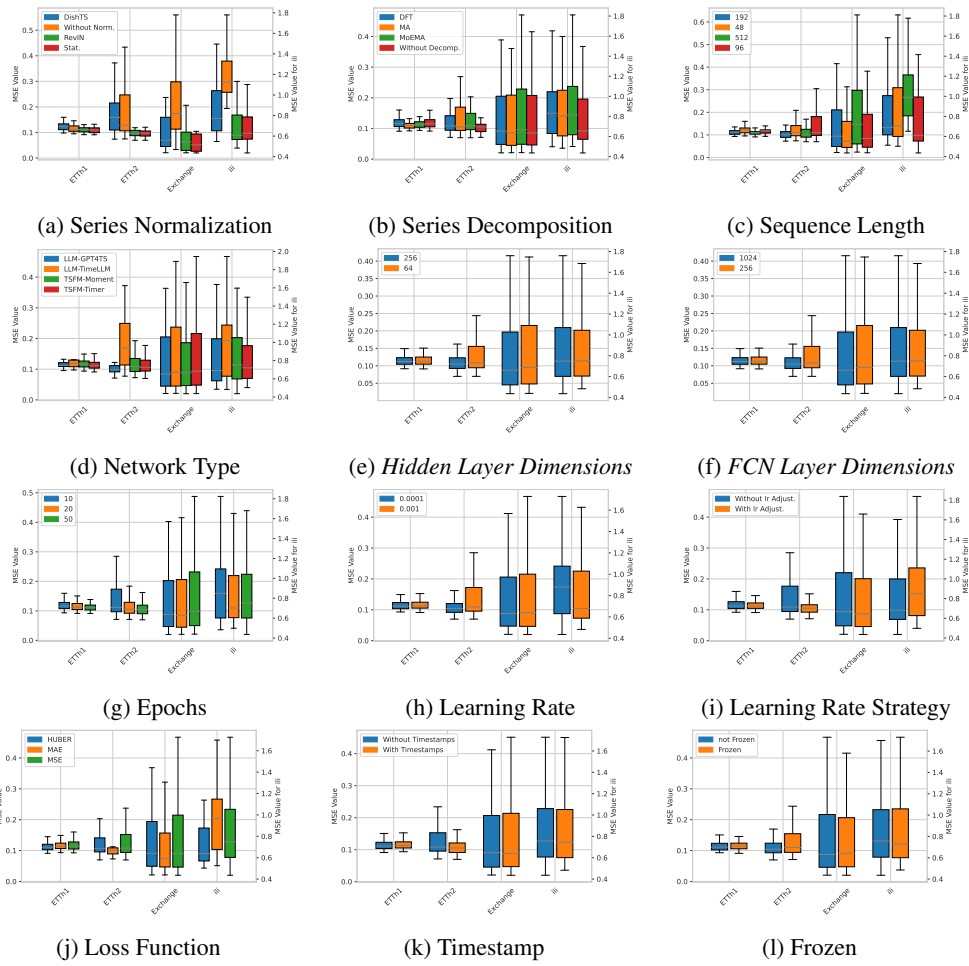

(a) Series Normalization      (b) Series Decomposition      (c) Sequence Length

(d) Network Type      (e) *Hidden Layer Dimensions*      (f) *FCN Layer Dimensions*

(g) Epochs      (h) Learning Rate      (i) Learning Rate Strategy

(j) Loss Function      (k) Timestamp      (l) Frozen

Figure H5: Overall performance across all design dimensions when using LLMs or TSFMs in long-term forecasting. The results (**MSE**) are averaged across all forecasting horizons. Due to the significantly different value range and variability of the ILI dataset compared to other datasets, its box plot is plotted using the right-hand *y*-axis, while all other datasets share the left-hand *y*-axis.

### H.2.2 Design Choices Evaluation Results for Long-term Forecasting Using MAE as the Metric

For the MAE-based performance evaluation, we analyze the effects of different design choices using both spider charts and box plots (Fig. H6 and Fig. H7). These visualizations complement the MSE-based analysis and confirm the generalizability of our findings across error metrics. In particular, normalization methods such as RevIN and Stationary consistently achieve the lowest MAE values, underscoring their effectiveness in mitigating non-stationarity. Similarly, decomposition strategies exhibit selective benefits: MA-based methods improve predictions on datasets like ETTh1 and ETTm2, while raw-series modeling remains more effective on ECL and Traffic, where decomposition tends to degrade performance.

Beyond preprocessing, MAE evaluations further validate the consistency of our architectural insights. Channel-independent designs retain strong performance across most datasets, except on Traffic and ILI, where localized dependencies dominate. Tokenization methods show stable ranking across both metrics, with patch-wise encoding consistently outperforming point-wise approaches. Notably, complex architectures such as Transformers provide only marginal gains over MLPs in certain cases (e.g., Traffic), suggesting that their benefits may not justify the added complexity. Overall, the alignment between MAE and MSE results reinforces the robustness of our design principles, demonstrating that the observed patterns are not metric-specific but instead reflect core relationships between architecture and forecasting performance.

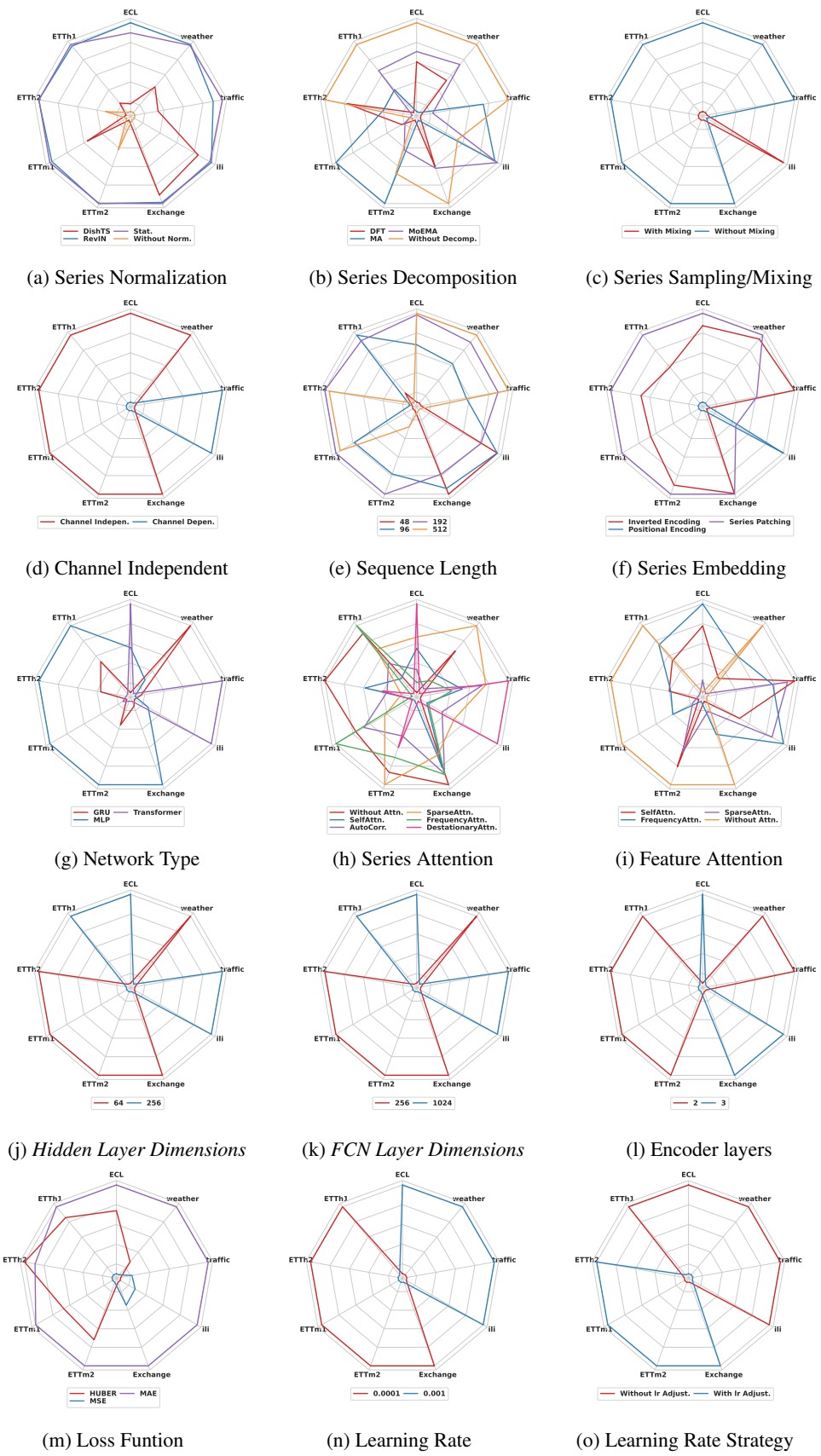

Figure H6: Overall performance across key design dimensions in long-term forecasting. The results (**MAE**) are based on the 75th percentile across all forecasting horizons.

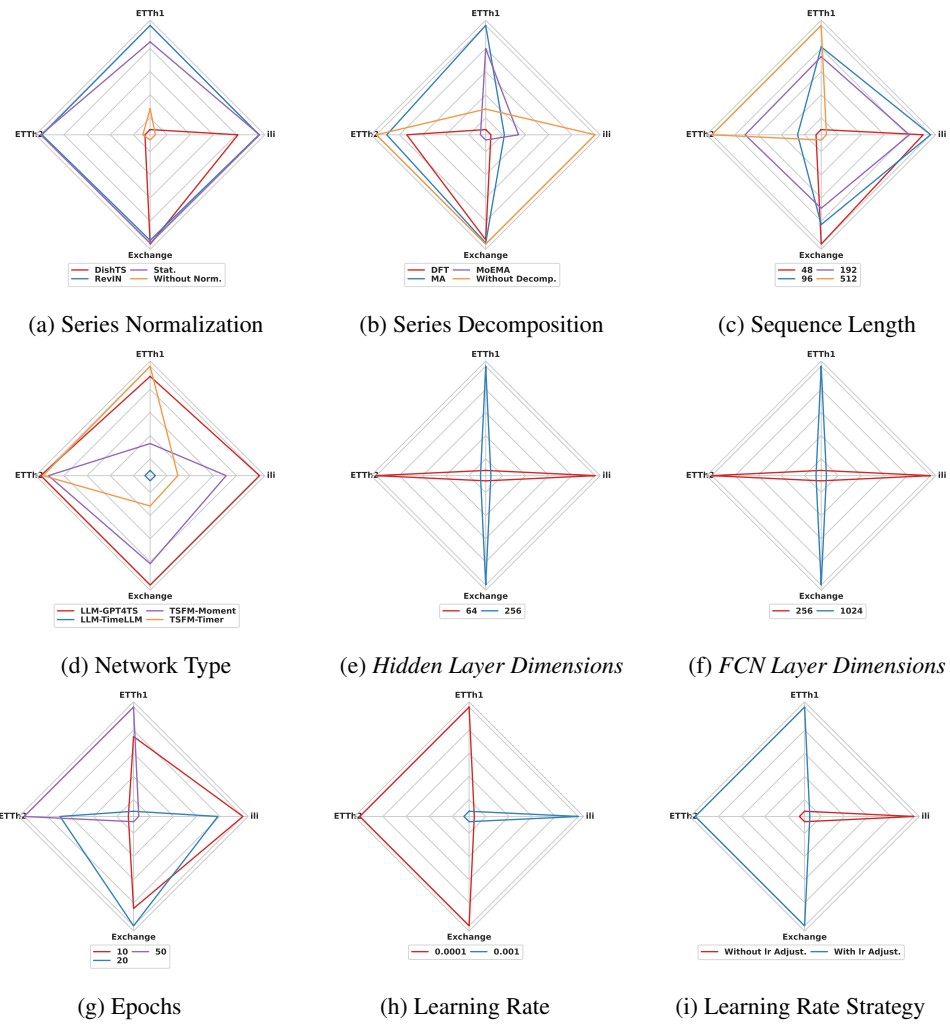

Figure H7: Overall performance across all design dimensions when using LLMs or TSFMs in long-term forecasting. The results (**MAE**) are based on the 75th percentile across all forecasting horizons.

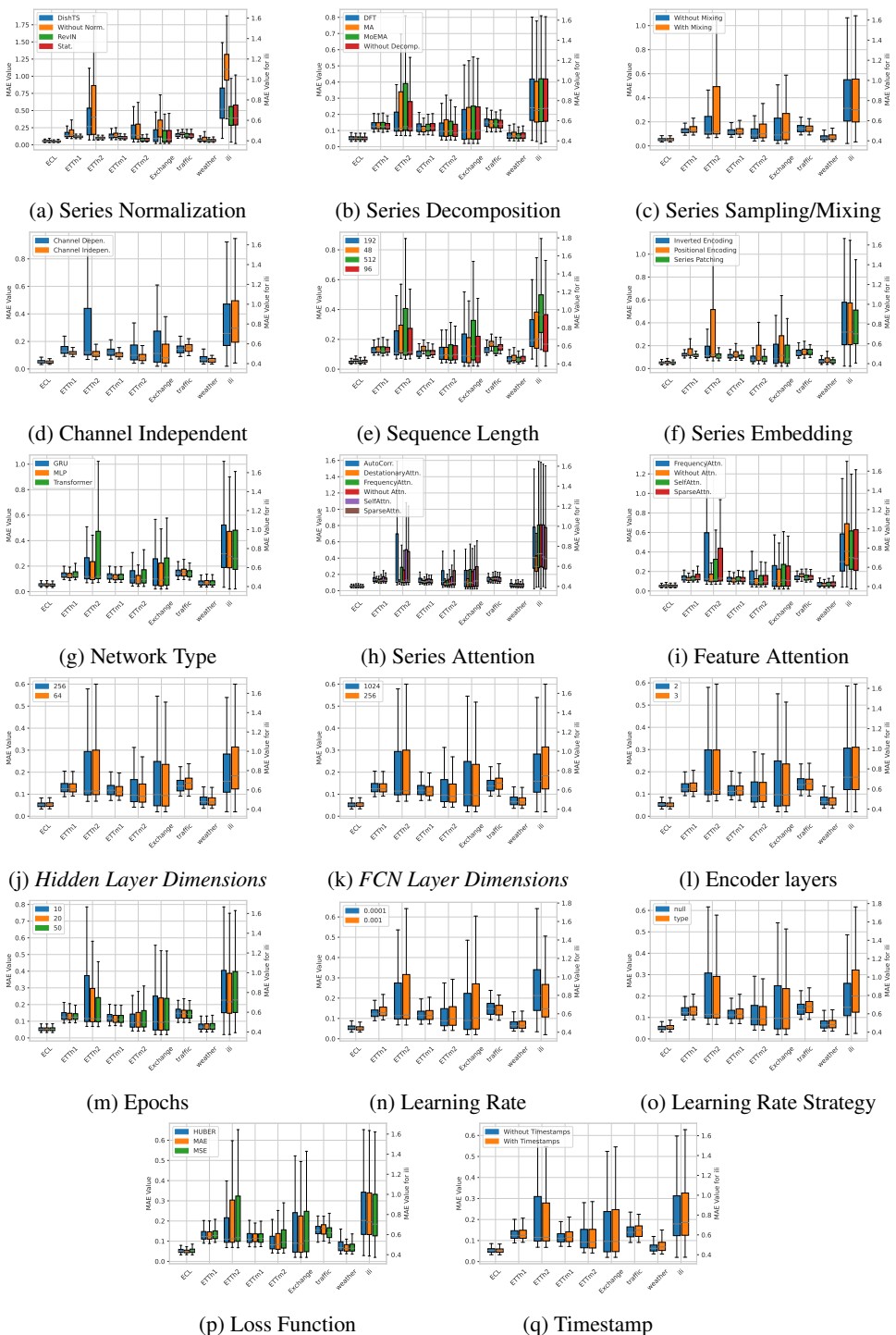

Figure H8: Overall performance across all design dimensions in long-term forecasting. The results (**MAE**) are averaged across all forecasting horizons.

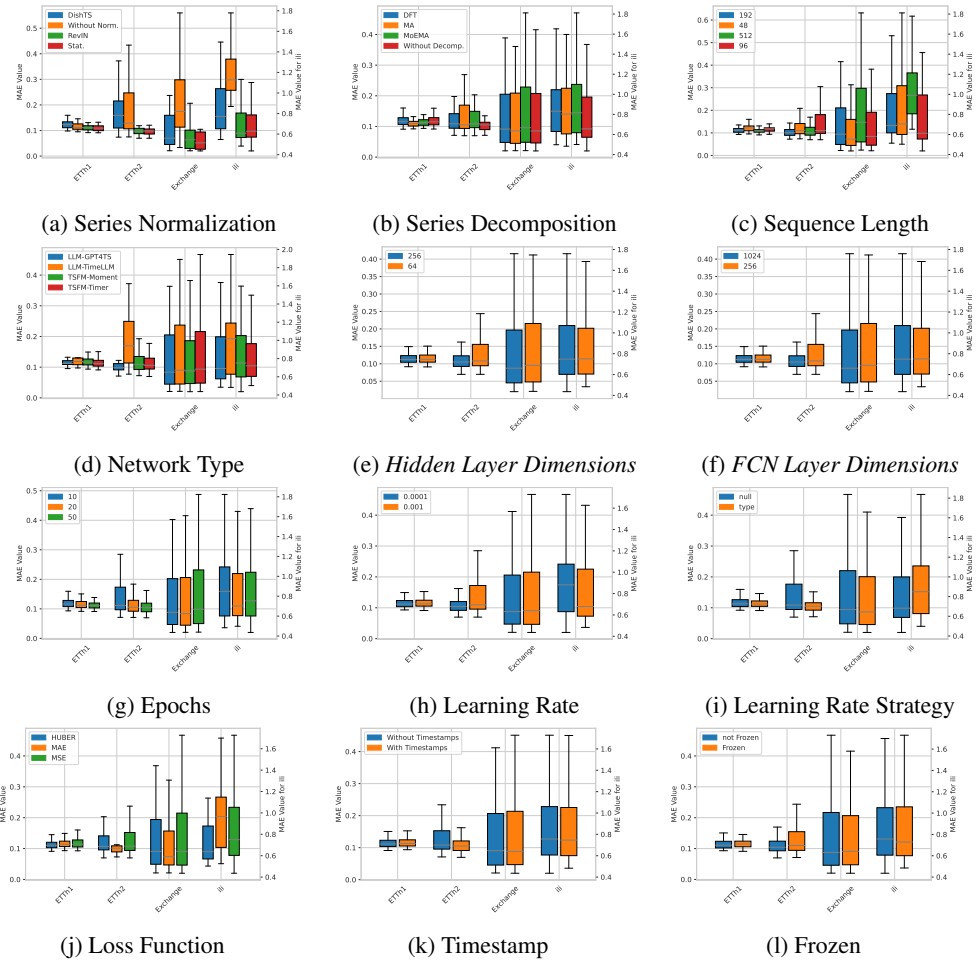

Figure H9: Overall performance across all design dimensions when using LLMs or TSFMs in long-term forecasting. The results (**MAE**) are averaged across all forecasting horizons.

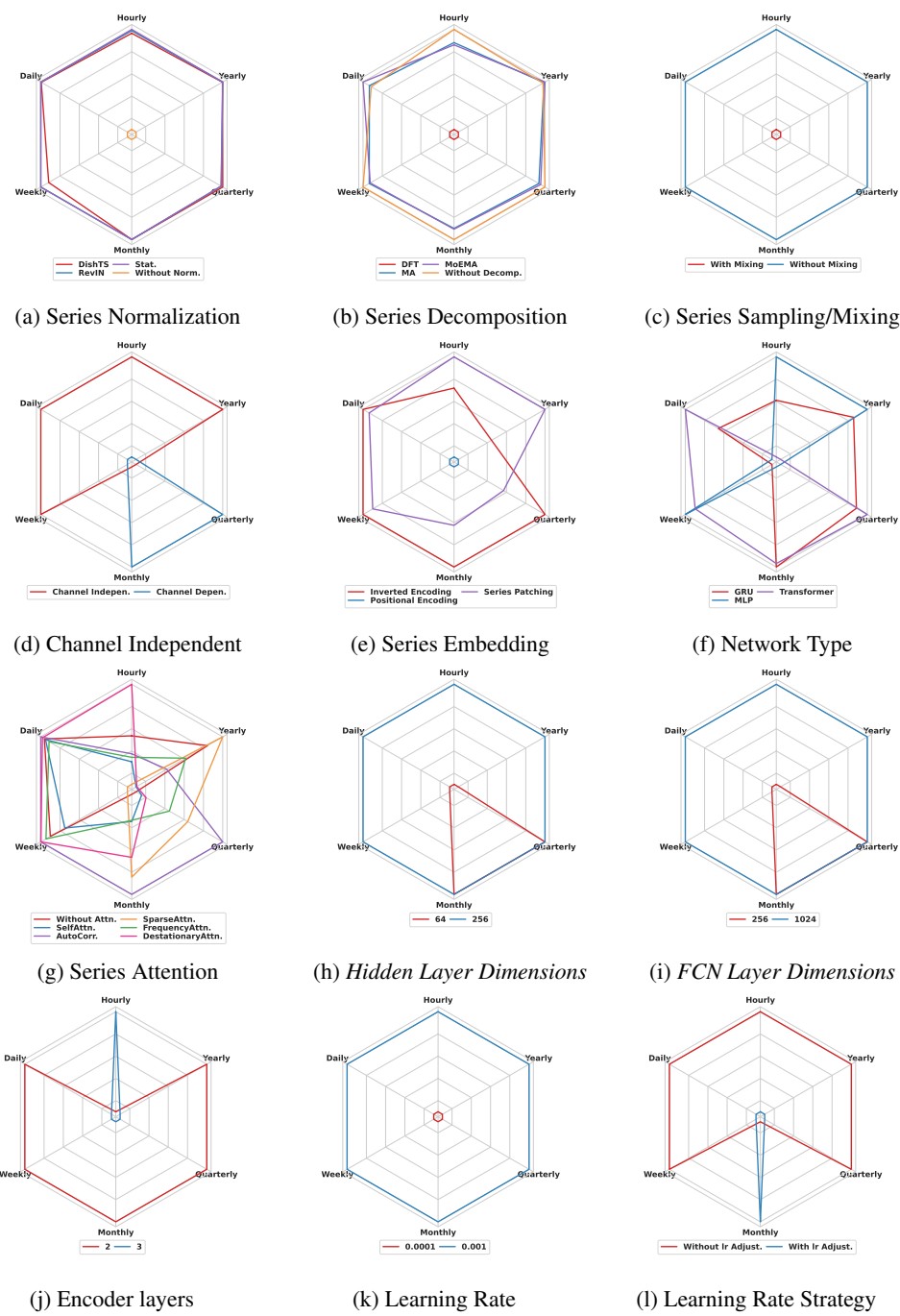

(a) Series Normalization     (b) Series Decomposition     (c) Series Sampling/Mixing

(d) Channel Independent     (e) Series Embedding     (f) Network Type

(g) Series Attention     (h) *Hidden Layer Dimensions*     (i) *FCN Layer Dimensions*

(j) Encoder layers     (k) Learning Rate     (l) Learning Rate Strategy

Figure H10: Overall performance across all design dimensions in short-term forecasting. The results (**MASE**) are based on the 75th percentile across all forecasting horizons.

### H.3 Complete Evaluation Results of Short-term Forecasting Using MASE, OWA and sMAPE as the Metric

For short-term forecasting, we comprehensively evaluate different design dimensions using both spider charts and box plots. The spider charts—shown in Figure H10, Figure H11, and Figure H12—visualize performance across datasets, with each vertex representing a benchmark dataset. Closer proximity to a vertex indicates stronger performance of a particular design choice in that dataset.

Complementary box plots are provided in Figure H13, Figure H14, and Figure H15, offering a statistical perspective on the distribution and robustness of performance across evaluation metrics.

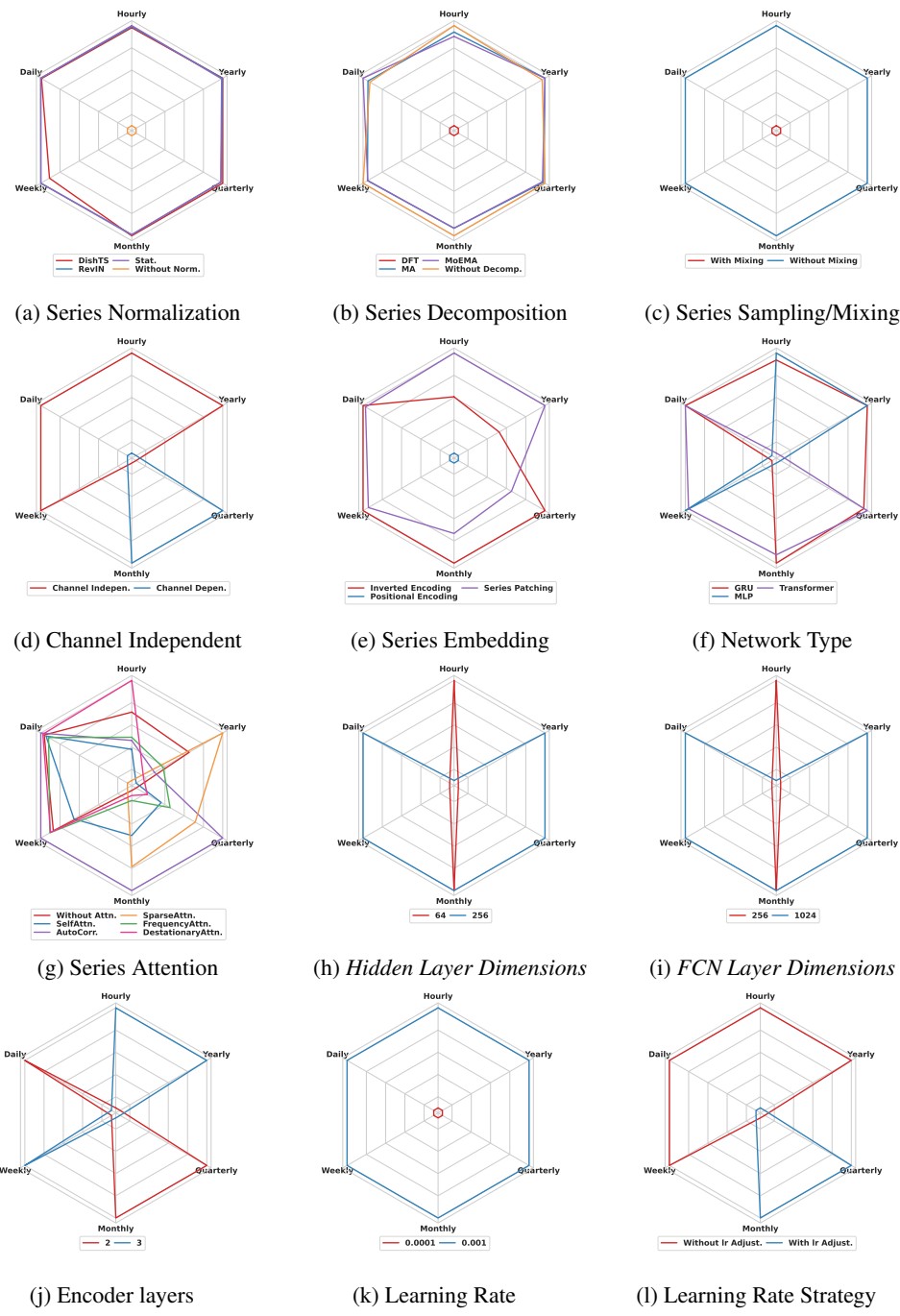

Figure H11: Overall performance across all design dimensions in short-term forecasting. The results (**OWA**) are based on the 75th percentile across all forecasting horizons.

Overall, the relative performance trends observed under MASE, OWA, and sMAPE metrics are consistent with those found in long-term forecasting tasks, reinforcing the generalizability and stability of our architectural choices.

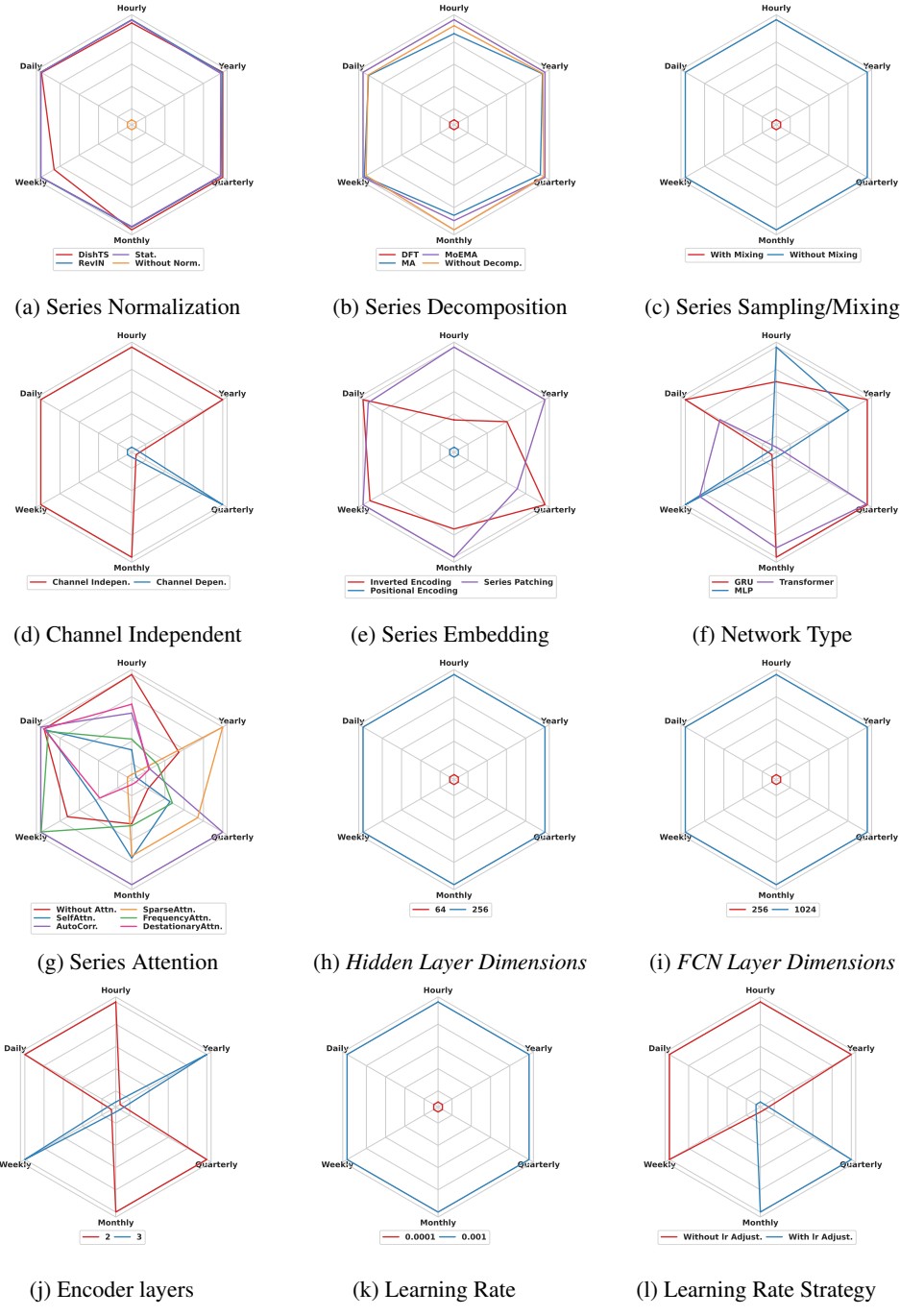

Figure H12: Overall performance across all design dimensions in short-term forecasting. The results (**SMAPE**) are based on the 75th percentile across all forecasting horizons.

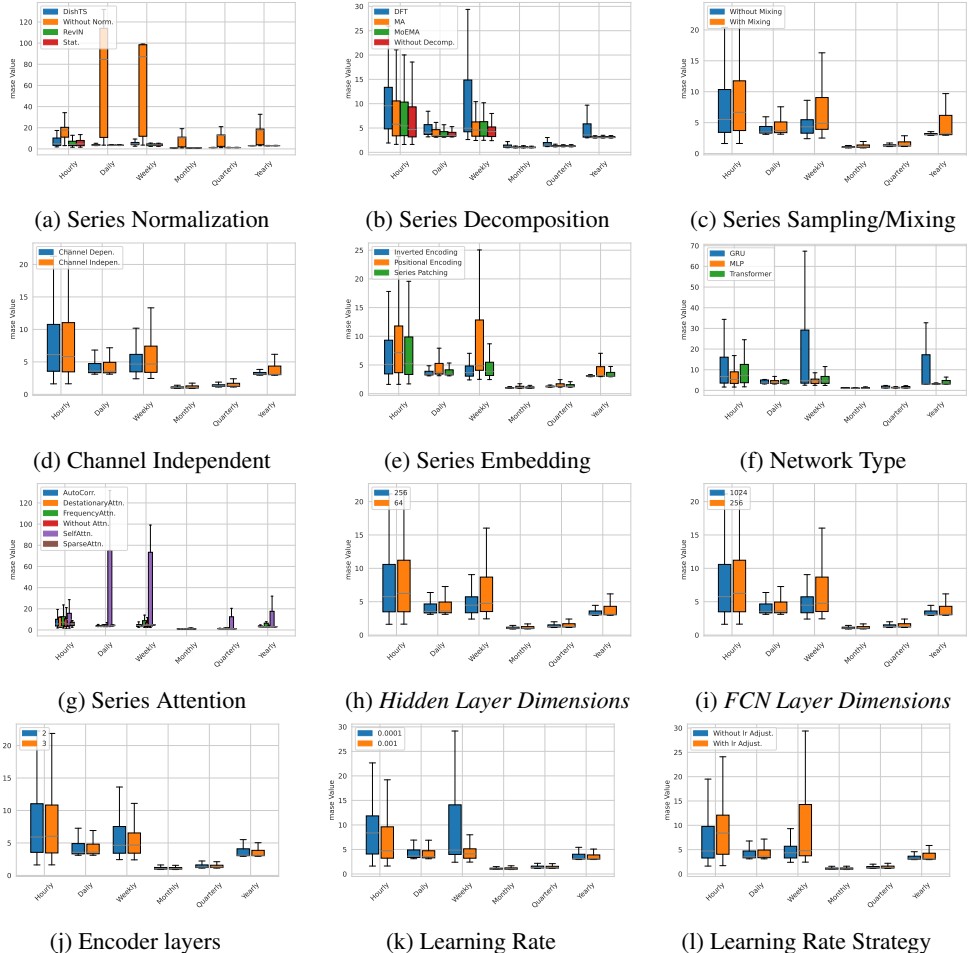

Figure H13: Overall performance across all design dimensions in short-term forecasting. The results are based on **MASE**.

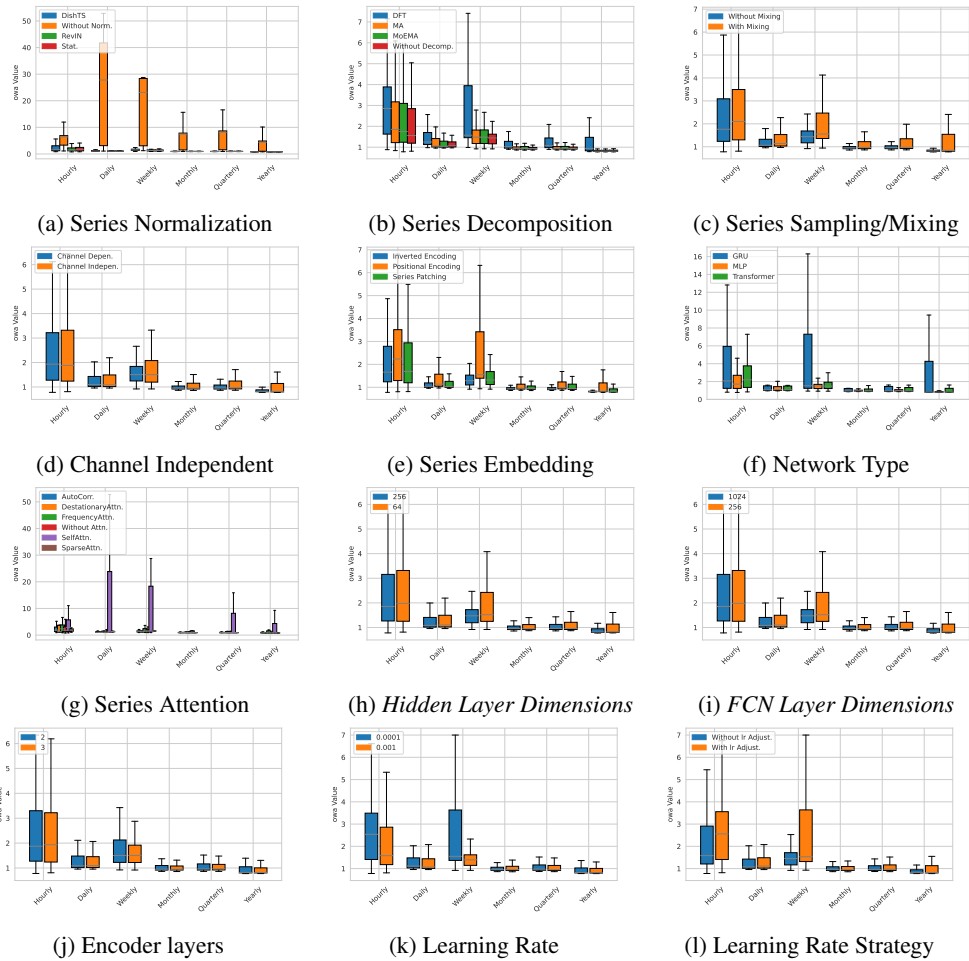

Figure H14: Overall performance across all design dimensions in short-term forecasting. The results are based on **OWA**.

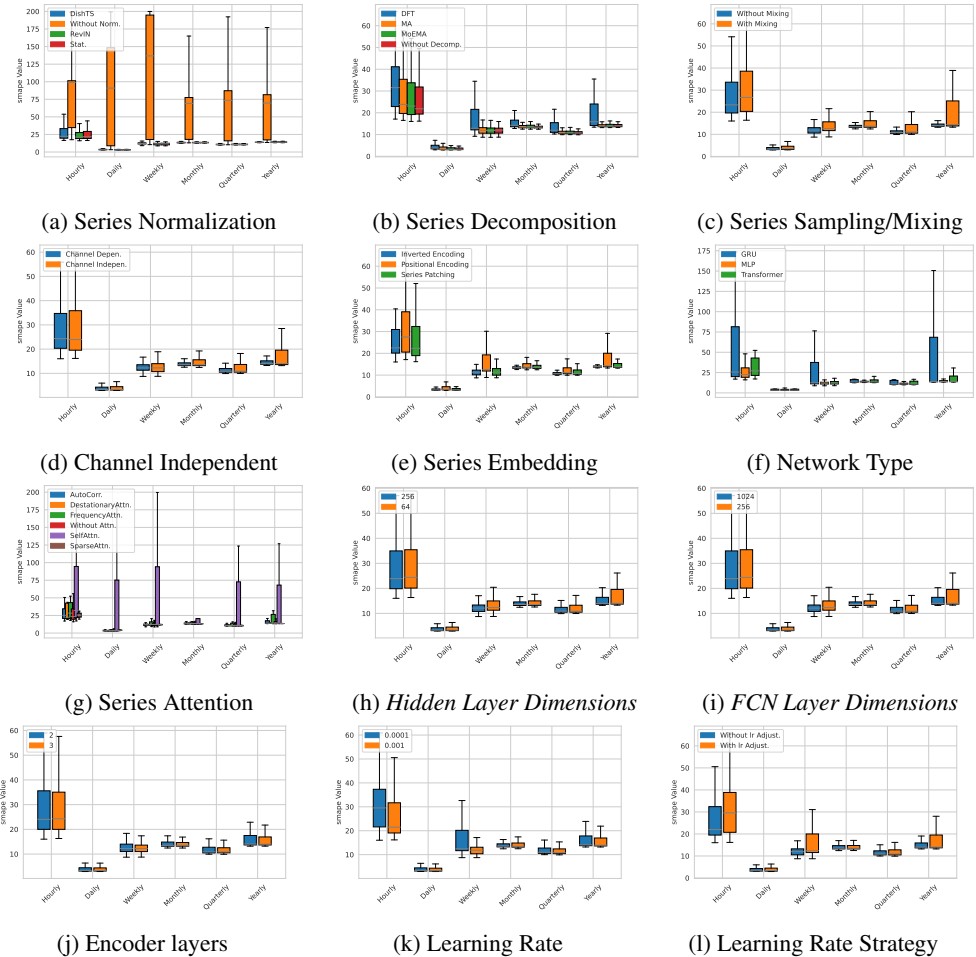

Figure H15: Overall performance across all design dimensions in short-term forecasting. The results are based on **SMAPE**.