# OpenReview forum: "TSGym: Design Choices for Deep Multivariate Time-Series Forecasting"
_NeurIPS.cc/2025/Datasets_and_Benchmarks_Track — Submitted to NeurIPS 2025 Datasets and Benchmarks Track_

### Official Review · Reviewer_jDWu · 2025-06-09

**Rating:** 4
**Confidence:** 3

**Summary:**

This paper introduces TSGym to decompose, evaluate, and automate the design of multivariate time series forecasting (MTSF) models. It first decomposes MTSF models into different components and evaluates each of them separately. Then, based on this analysis, it proposes an automated model component selection mechanism to facilitate automated learning.

**Additional Feedback:**

This paper addresses a timely and important problem in MTSF and proposes a tool that could be very useful to both researchers and practitioners. However, the current version falls short in terms of clarity, analytical depth, and methodological transparency. While analytical depth may be an overreach, I recommend that authors improve the clarity of the method section, better contextualize the empirical findings, and discuss computational costs more explicitly. With such revisions, the paper could become a strong contribution.

**Dataset Code Accessibility:**

Yes

**Ethical Considerations:**

No, there are no or only very minor ethics concerns

**Final Justification:**

I am satisfied with the rebuttal and happy to see the additional results. However, due to the character limit, some interesting claims outlined in the rebuttal could not be presented, and I was unable to further evaluate them. Based on that, I raised my score to 4. I believe that, provided the authors can clearly present the discussions and details in the manuscript, the paper makes a good contribution.

**Limitations Weaknesses:**

* **Computational Cost and Efficiency Not Fully Addressed:**
While the proposed pipeline is automated, it still relies on training and evaluating a large number of model variants. The authors mention meta-learning and zero-shot transfer for test datasets, but the cost of preparing the meta-training data (i.e., evaluating many models on many datasets) is non-trivial. It would be good to see either a more explicit discussion of this overhead or strategies to mitigate it (e.g., smarter sampling or progressive search).

* **Lack of Principled Design Guidance:**
TSGym’s model construction process is purely empirical. It predicts performance via a meta-learner trained on past data, but lacks theoretical justification or interpretability guarantees. As such, the automated selection acts more like a black box than a design assistant grounded in understanding the underlying principles of MTSF.

* **Limited Discussion and Interpretation of Results:**
While the authors conduct many experiments and provide performance tables, the paper would benefit from deeper analysis. For instance, why do certain design choices (e.g., channel-independence, series-patching) work better on some datasets but not others? Currently, the insights are mostly descriptive rather than analytical, which weakens the impact of the benchmarking contribution.

* **Dense Presentation Hinders Accessibility:**
The paper attempts to cover a large number of design choices and evaluations, but the presentation is often dense and hard to parse. Important technical details are embedded in long paragraphs without supporting figures or diagrams. The clarity of the methodology and pipeline would be improved by the use of visual schematics or tabular summaries (especially for the design dimensions).

**Strengths Contributions:**

* **Component-Level Evaluation is a Timely Contribution:**
The paper tackles an important gap in the multivariate time-series forecasting (MTSF) literature—namely, the lack of fine-grained analysis of model components. By systematically decoupling forecasting pipelines into design dimensions (e.g., preprocessing, encoding, architecture), the authors provide an insightful framework for understanding which elements contribute most to performance across datasets. This is a valuable step forward from the prevailing practice of evaluating models as monolithic entities.

* **TSGym Provides a Modular and Scalable Benchmarking Tool:**
The authors introduce TSGym, which serves both as a large-scale benchmark and an automated model construction framework. The systematic evaluation of 16 design dimensions across 10 datasets results in a rich diagnostic resource for the community, and the modularity of the approach enhances its future extensibility.

* **Promising Empirical Results with Clear Potential for Practitioners:**
The experimental results are promising. TSGym outperforms a wide range of state-of-the-art (SOTA) models in both long- and short-term forecasting tasks. Particularly notable is the fact that it performs well even when constrained to light-weight models like MLPs and RNNs, suggesting practical utility under resource constraints.

---

> ### Author Rebuttal · Authors · 2025-07-31
>
> **Q1**. Computational Cost and Efficiency Not Fully Addressed. A more explicit discussion of this overhead or strategies to mitigate it (e.g., smarter sampling or progressive search) would be better.
>
> **R1**.
> Thank you for raising this important point. We ran experiments using 4×NVIDIA A100 (80GB) and 8×NVIDIA A100 (40GB) GPUs with parallelization to speed up runtime. Building the meta-learner's training pool took 2–3 days in total, with costs varying by dataset size and dimensions. Moreover, to address concerns regarding computational cost and efficiency, we explored
>
> 1.**Efficiency of Random Sampling.**  We trained the meta-learner on progressively larger subsets of our model combination pool (from 5% to 100%).  As the results in the table below show, its performance is highly robust to a reduction in training data/combinations. Near-optimal performance is achieved with just 25% of the model combinations, which shows that a significantly lower computational cost is feasible without a major degradation in performance.
>
> |$M_s$|MSE(avg. Rank)|MAE(avg. Rank)|
> |:-:|:-:|:-:|
> |5%|3.67|3.44|
> |10%|2.67|3.00|
> |25%|1.67|1.78|
> |50%|1.89|2.67|
> |75%|1.89|2.22|
> |100%|1.67|2.00|
>
> 2.**Optimization via Bayesian Search with Optuna [1].** Following your suggestion, we adopted Optuna, which is based on Bayesian optimization, as a more efficient sampling strategy than random search. After an initial 50 random trials, Optuna intelligently sampled an additional 50 configurations. The experimental results confirm that **Optuna significantly improves search efficiency, discovering superior model combinations with far fewer evaluations**.
>
>   The table below compares the sampled combination distributions, where **Trials to Exceed Best Random** indicates the number of iterations Optuna required to outperform the best result found by a much larger random search. Optuna consistently discovers **better-performing** models with **significantly fewer evaluations** (less than 100 trials vs. 1.5-5 times more for random search). This confirms that a meta-learner trained on this intelligently curated set is more effective, and we will add this analysis to our paper.
>
> |**Dataset**|**Method**|**Trials to Exceed Best Random**|**Max mse**|**Min mse**|**Q1 mse**|**Median mse**|**Q3_mse**|**Total Count**|
> |:-:|:-:|:-:|:-:|:-:|:-:|:-:|:-:|:-:|
> |**ECL**|Optuna|**56**|**0.414**|**0.131**|**0.158**|**0.182**|**0.213**|**400**|
> ||Random||0.862|0.134|0.179|0.212|0.247|1493|
> |**ETTh1**|Optuna|**70**|**1.210**|**0.355**|**0.416**|**0.450**|**0.519**|**400**|
> ||Random||1.860|0.376|0.442|0.494|0.581|1129|
> |**ETTh2**|Optuna|**91.5**|**9.060**|**0.268**|**0.342**|**0.400**|**0.539**|**400**|
> ||Random||30.771|0.270|0.391|0.464|1.017|1147|
> |**ETTm1**|Optuna|**59.5**|**4317.839**|**0.293**|**0.341**|**0.405**|**0.472**|**400**|
> ||Random||304538.500|0.295|0.378|0.447|0.547|623|
> |**ETTm2**|Optuna|**56.5**|**9.344**|**0.159**|**0.221**|**0.279**|**0.385**|**400**|
> ||Random||198.023|0.167|0.259|0.371|0.584|762|
> |**Exchange**|Optuna|**90.25**|**15.054**|**0.081**|**0.169**|**0.280**|**0.681**|**400**|
> ||Random||12.106|0.080|0.189|0.391|0.959|2033|
> |**Ili**|Optuna|**54**|**7.503**|**1.506**|**1.891**|**2.302**|**3.080**|**400**|
> ||Random||7.532|1.495|2.363|2.842|4.027|3976|
> |**Traffic**|Optuna|**93**|**1.051**|**0.387**|**0.438**|**0.491**|**0.612**|**400**|
> ||Random||1.473|0.379|0.487|0.589|0.686|1018|
> |**Weather**|Optuna|**74**|**29.895**|**0.144**|**0.193**|**0.245**|**0.312**|**400**|
> ||Random||109.467|0.143|0.207|0.263|0.344|656|
>
>   Consequently, the following experimental results indicate that meta-learner trained on the high-quality samples from **Optuna achieved better performance than one trained with random search in most cases**. Optuna performed slightly worse on the ILI dataset due to testing 9 times fewer trials than random search, but increasing its trial count would likely improve results.
>
> |**Models**|**TSGym(OptunaBased)**||**TSGym**||
> |:-:|:-:|:-:|:-:|:-:|
> |**Metric**|**MSE**|**MAE**|**MSE**|**MAE**|
> |**ETTm1**|**0.350**|**0.373**|0.357|0.383|
> |**ETTm2**|**0.252**|**0.310**|0.261|0.319|
> |**ETTh1**|0.429|**0.440**|**0.426**|**0.440**|
> |**ETTh2**|**0.354**|**0.392**|0.358|0.400|
> |**ECL**|**0.166**|**0.262**|0.170|0.265|
> |**Traffic**|0.439|**0.295**|**0.435**|0.313|
> |**Weather**|**0.229**|**0.258**|**0.229**|0.268|
> |**Exchange**|0.457|0.446|**0.410**|**0.431**|
> |**ILI**|3.194|1.234|**2.233**|**1.015**|
>
>   We also note that Optuna can provide interpretability. Its built-in fANOVA importance analysis (as shown below) quantifies each component's contribution to model performance through regression modeling. This analysis revealed that Sequence Length and Series Normalization are the most critical factors, accounting for the majority of performance variance.
>
> ||Design Dimensions | Importance |
> |:-:|:-:|:-:|
> |1 | Sequence Length|0.270 |
> |2 | Series Normalization |0.255 |
> |3 |Series Embedding|0.134 |
> |4 |Feature Attention |0.077 |
> |5 |Series Decomposition|0.053 |
> |6 | Channel Independent|0.050 |
> |7 | Series Sampling/Mixing |0.029 |
> |8 | Epochs |0.025 |
> |9 |d_model, d_ff|0.020 |
> |10|Learning Rate |0.020 |
>
> [1] Takuya Akiba, Shotaro Sano, Toshihiko Yanase, Takeru Ohta, and Masanori Koyama. 2019. Optuna: A Next-generation Hyperparameter Optimization Framework. In KDD.
>
> **Q2**. Since the model selection process is purely empirical, does it lack of Principled Design Guidance? How can the system offer interpretable justifications for its automated choices, moving beyond its "black box" nature?
>
> **A2**.
> Thank you for your question. We would like to clarify that TSGym is designed not only as an automated model construction tool but also as a systematic framework to discover and validate design principles empirically. Our goal is to transform the art of MTSF model design into a more data-driven science, and our paper provides guidance in two key ways.
>
> First, **a significant portion of our paper is dedicated to a component-level analysis that moves beyond holistic model evaluation to generate transparent design guidance.** Our large-scale evaluation in Section 4.2, for instance, provides several clear takeaways, such as the universal effectiveness of normalization methods like RevIN and the dataset-specific utility of decomposition. This systematic decoupling is a primary contribution that offers the kind of design insights the reviewer is calling for.
>
> Second, inspired by your valuable feedback, we conducted a new analysis using an XGBoost model as our meta-learner (which also achieves highly robust performance) to explicitly quantify feature importance. The following table presents the top five most important features for each dataset, **revealing the most critical data characteristics and design dimensions** for effective model construction.
>
> ||||||||
> |:-:|:-:|:-:|:-:|:-:|:-:|:-:|
> |**ETTm1**|mean_Negativeturningpoints|series norm|mean_Centroid|mean_MFCC_0|min_Positiveturningpoints|
> ||0.08|0.06|0.05|0.05|0.05|
> |**ETTm2**|mean_Negativeturningpoints|series norm|mean_MFCC|mean_Centroid|q25_Kurtosis|
> ||0.10|0.05|0.05|0.05|0.04|
> |**ETTh1**|mean_MFCC_10|series norm|mean_Spectralroll-on|mean_Spectraldistance|mean_MFCC|
> ||0.08|0.06|0.05|0.04|0.04|
> |**ETTh2**|mean_MFCC_0|series norm|mean_Medianfrequency|mean_Centroid|min_Meanabsolutediff|
> ||0.07|0.05|0.05|0.04|0.04|
> |**ECL**|mean_MFCC|std_MFCC|series norm|mean_Centroid|mean_Maxpowerspectrum|
> ||0.08|0.08|0.07|0.06|0.04|
> |**Traffic**|q25_Kurtosis|series norm|mean_Centroid|mean_Maxpowerspectrum|mean_MFCC|
> ||0.08|0.07|0.06|0.05|0.05|
> |**Weather**|mean_MFCC|mean_Negativeturningpoints|series norm|min_Negativeturningpoints|mean_Centroid|
> ||0.14|0.11|0.06|0.05|0.04|
> |**Exchange**|std_MFCC|mean_Negativeturningpoints|mean_MFCC|series norm|mean_Medianfrequency|
> ||0.11|0.1|0.07|0.06|0.03|
> |**ILI**|mean_MFCC|mean_Maximumfrequency|channel independent|series norm|mean_LPCC|
> ||0.09|0.06|0.05|0.05|0.05|
>
> **Q3**.  Can you provide a deeper analysis of why certain design choices (e.g., channel-independence, series-patching) perform better on some datasets but not others? Moving from a descriptive summary of performance to an analytical interpretation can strengthen this benchmark.
>
> **A3**.
> Thank you for your suggestions. To provide deeper insights, we formulated and tested **7 most contentious claims** from the research community across various datasets. Due to the character limit of this rebuttal, we can only present the claims below. The detailed experimental evidence and conclusions for each will be integrated into our revised paper, and we welcome further discussion on these findings.
>
> Claim 1: Model scale should match data scale. (Yes)
>
> Claim 2: Transformers are less robust than MLPs. (Yes)
>
> Claim 3: Transformers exhibit a higher upper bound than MLPs. (No)
>
> Claim 4: The superiority of Channel-Independent (CI) vs. Channel-Dependent (CD) correlates with inter-channel correlation. (No)
>
> Claim 5: Series normalization mitigates distribution shift. (Yes)
>
> Claim 6: Novel sequence encodings outperform the traditional single-timestep encoding.  (Yes)
>
> Claim 7: Novel attention mechanisms outperform vanilla self-attention. (Yes)
>
> **Q4**. Can the presentation be made less dense and more accessible, particularly for the complex technical details? The clarity of the methodology and pipeline would be improved by the use of visual schematics or tabular summaries (especially for the design dimensions).
>
> **A4**.
> We agree that clarity is essential. Our paper currently uses Fig. 1 to provide a high-level overview of the TSGym pipeline, while Table 1 and Appendix G.1 offer detailed descriptions of every component in our design space. To further enhance transparency, we have also created a new, detailed diagram that illustrates the input tensor's flow step-by-step through the pipeline. While space constraints prevent us from including it in this rebuttal, we will add this diagram to our revised paper.

---

> > ### Comment · Reviewer_jDWu · 2025-08-05
> >
> > Dear authors,
> >
> > Thank you very much for the detailed rebuttal. I am satisfied with it and will increase my score, and I would be happy to recommend it for acceptance. It's a shame that the authors could not present the 7 claims outlined in the rebuttal due to the character limit. I want to alert the authors, however, that many of these claims are very bold and would require strong evidence to support them. If the authors plan to include them in the manuscript, please make sure they are justified.
> >
> > Thanks!
> >
> > Reviewer

---

> > > ### Author Response · Authors · 2025-08-05
> > > **Thanks for your insightful and effective suggestions.**
> > >
> > > We are grateful for the suggestions provided by you and other reviewers, which have enabled TSGym to become a more profound and comprehensive benchmarking work. It not only includes **a large number of experiments** and **effective automated selection** schemes, but also offers **insightful observations**. In the future, we will present the complete experimental results in both the paper and on GitHub to support these conclusions. Additionally, we believe that our extensive experimental results will bring **more interesting findings** to the TSF research community. **Anyone** will be able to access and independently analyze the insights derived from these results on our GitHub repository at a later date.

---

> ### Author Response · Authors · 2025-08-04
> **Follow-up On Our Responses**
>
> We want to send this friendly reminder and are happy to address any further comments reviewers may have. Also, If our responses address your questions, we want to kindly ask for your reconsideration of the score.
>
> Thank you so much for devoting time to making TSGym a better work. We are looking forward to any further discussions :)

---

### Official Review · Reviewer_ni7G · 2025-07-01

**Rating:** 5
**Confidence:** 3

**Summary:**

This paper introduces a framework for MTSF, by decomposing forecasting pipelines into 16 core design dimensions—including preprocessing techniques, tokenization strategies (patch-based vs. inverted encoding), and network architectures (Transformers, LLMs, TSFMs)—it enables granular benchmarking across 10 diverse datasets. The authors report findings include simple components like channel-independent processing and MLP backbones often outperform complex attention mechanisms. additionally, the proposed framework leverages meta-learning to automate optimal component selection, constructing models that outperform some competing methods in long-term forecasting tasks.

**Dataset Code Accessibility:**

Partly

**Dataset Code Comments:**

There's a GitHub, and the authors claim all the code scripts will be made public.

**Ethical Considerations:**

No, there are no or only very minor ethics concerns

**Final Justification:**

The responses from the authors primarily addressed my concerns.

**Limitations Weaknesses:**

1. AutoML and Neural Architecture Search (NAS) (e.g., TPOT, h2o-3, and previously Microsoft nni (unfortunately archived now)) ​​have​​ been developed for many years. Although the authors claim their auto model construction component ​​is specifically​​ designed for MTSF tasks, ​​general​​ AutoML techniques are applicable to diverse scenarios. Therefore, ​​a comparison between them is relevant​​. In my view, the proposed auto model construction component and other AutoML or NAS methods are broadly the ​​same category​​ of technique, ​​making​​ a comparison between them is fair and helpful.

2. It is unclear​​ how the ​​16 distinct dimension evaluations​​ actually ​​contribute​​ to the model design. In deep learning training, ​​it is common for​​ these different dimensions ​​to be​​ highly correlated, and it is not guaranteed​​ that a component-level ​​improvement​​ ​​will​​ ultimately ​​lead to​​ improved overall performance.

3. ​​It is difficult to classify​​ the ​​third purported​​ contribution, discussing emerging models, as ​​genuinely​​ a valid contribution point.

**Strengths Contributions:**

1. The work introduce a component-level decomposition framework​​ for multivariate time series forecasting tasks, isolating 16 design dimensions, including pre-processing, encoding, architecture, and optimization etc, which indeed helps one to visually see the properties of specific datasets from different component level insights.

2. Although autoML and NAS are not new, the proposed model construction component is designed for MTSF​, it is results show that using dataset meta-features to assemble models that outperform SOTA methods (e.g., PatchTST, TimeMixer) in 66.7% of long-term tasks while accommodating emerging paradigms like LLMs/TSFMs.

---

> ### Author Rebuttal · Authors · 2025-07-31
>
> **Q1.** AutoML and Neural Architecture Search (NAS) (e.g., TPOT, h2o-3, and previously Microsoft nni (unfortunately archived now)) have been developed for many years. Although the authors claim their auto model construction component is specifically designed for MTSF tasks, general AutoML techniques are applicable to diverse scenarios. Therefore, a comparison between them is relevant. In my view, the proposed auto model construction component and other AutoML or NAS methods are broadly the same category of technique, making a comparison between them is fair and helpful.
>
> **A1.** We sincerely thank you for your insightful feedback and valuable suggestions. To address this point, we first investigated a number of well-known AutoML and NAS libraries, including TPOT[1], H2O-3[2], Microsoft NNI[3], Auto-Keras[4], Auto-Sklearn[5], NASLib[6]. We noted, however, that **most of these frameworks are not specifically designed for time-series forecasting**, which would make adapting them for MTSF tasks both burdensome and likely result in unfair comparison.
>
> For this reason, we **selected two leading AutoML libraries that explicitly support MTSF** for a direct and meaningful comparison: **AutoGluon-TimeSeries[7]** and **AutoTS[8]**. We then benchmarked TSGym against these two methods on both short-term and long-term forecasting tasks, with the results presented below.
>
> **Short-term Forecasting Comparison**
>
> | Models | TSGym (ours) | AutoGluon | AutoTS |
> | :--- | :--- | :--- | :--- |
> | **OWA** | **0.872** | 0.950 | 2.002 |
> | **SMAPE** | **12.013** | 13.178 | 18.977 |
> | **MASE** | **1.575** | 1.775 | 4.981 |
>
> **Long-term Forecasting Comparison**
>
> |  | TSGym (ours) |  | AutoGluon |  | AutoTS |  |
> |:---:|:---:|:---:|:---:|:---:|:---:|:---:|
> | Metric | MSE | MAE | MSE | MAE | MSE | MAE |
> | ETTm1 | **0.357** | **0.383** | 0.482 | 0.408 | 0.744 | 0.546 |
> | ETTm2 | **0.261** | **0.319** | 0.273 | 0.337 | 0.392 | 0.389 |
> | ETTh1 | **0.426** | **0.440** | 0.503 | 0.473 | 0.981 | 0.610 |
> | ETTh2 | **0.358** | **0.400** | 0.419 | 0.430 | 0.589 | 0.488 |
> | ECL | **0.170** | **0.265** | 0.265 | 0.328 | 0.327 | 0.355 |
> | Traffic | **0.435** | 0.313 | 0.555 | 0.325 | 0.739 | **0.311** |
> | Weather | **0.229** | **0.268** | 0.236 | 0.270 | 0.519 | 0.372 |
> | Exchange | 0.410 | 0.431 | **0.330** | **0.393** | 0.588 | 0.494 |
> | ILI | **2.233** | 1.015 | 2.271 | **0.979** | 2.533 | 1.049 |
>
> As the results demonstrate, TSGym **consistently and significantly outperforms both AutoGluon and AutoTS** across the vast majority of datasets for both short-term and long-term forecasting tasks.
>
> In summary, while TSGym shares the goals of AutoML, its distinct methodology——automated model construction via fine-grained component decomposition and meta-learning——offers a clear advantage over existing AutoML methods for MTSF. This specialized design, now validated by both state-of-the-art deep-learning MTSF methods and the newly included AutoML baselines, represents a significant and valuable contribution to the MTSF community. We will add this comparison and discussion to our revised manuscript.
>
> [1] Ribeiro, P. et al. (2024). TPOT2: A New Graph-Based Implementation of the Tree-Based Pipeline Optimization Tool for Automated Machine Learning. In: Winkler, S., Trujillo, L., Ofria, C., Hu, T. (eds) Genetic Programming Theory and Practice XX. Genetic and Evolutionary Computation. Springer, Singapore.
>
> [2] Arora, A., Candel, A., Lanford, J., LeDell, E., and Parmar, V. (Oct. 2016). Deep Learning with H2O.
>
> [3] Microsoft. Neural Network Intelligence (version v3.0pt1).
>
> [4] Haifeng Jin, François Chollet, Qingquan Song, and Xia Hu. "AutoKeras: An AutoML Library for Deep Learning." the Journal of machine Learning research 6 (2023): 1-6.
>
> [5] Efficient and Robust Automated Machine Learning Matthias Feurer, Aaron Klein, Katharina Eggensperger, Jost Springenberg, Manuel Blum and Frank Hutter Advances in Neural Information Processing Systems 28 (2015)
>
> [6] Ruchte, Michael and Zela, Arber and Siems, Julien and Grabocka, Josif and Hutter, Frank. NASLib: A Modular and Flexible Neural Architecture Search Library. (2020)
>
> [7] Erickson, Nick, et al. "AutoGluon-Tabular: Robust and Accurate AutoML for Structured Data."
>
> [8] winedarksea. (2024). AutoTS. Github.
>
> **Q2.** It is unclear how the 16 distinct dimension evaluations actually contribute to the model design. In deep learning training, it is common for these different dimensions to be highly correlated, and it is not guaranteed that a component-level improvement will ultimately lead to improved overall performance.
>
> **A2.** Thank you for this insightful question. We agree that **analyzing the interactions between design dimensions is the central challenge** of MTSF model design. However, a principled, theoretical analysis of all pairwise and higher-order interactions between our **16 design dimensions may be analytically intractable**, mainly because of the immense complexity of the interaction space.
>
> The core of TSGym is that we rely on the meta-learner to discover the complex "chemical reactions" among these design dimensions empirically. By training on the performance of diverse combinations, it effectively learns which components work well together for a given dataset. Moreover, **inspired by your suggestion, we advanced beyond our initial random sampling of model combinations by experimenting with Bayesian Optimization**, an optimization strategy designed to **actively consider component correlations during the sampling process**. The meta-learner trained on these curated combinations proved to be a more effective predictor (please see the Table below), validating that our framework can be enhanced to explicitly leverage these complex component interactions. We thank you again for indicating this valuable direction and will add this new analysis to our revised manuscript.
>
> | **Models**     | **TSGym**      |    **(Optuna)**                | **TSGym**          |                    | **DUET**            |                    |
> |:--------------:|:---------------------:|:------------------:|:-----------------:|:------------------:|:------------------:|:------------------:|
> | **Metric**     | **MSE**        | **MAE**            | **MSE**           | **MAE**            | **MSE**            | **MAE**            |
> | **ETTm1**      | **0.350**             | **0.373**          | 0.357             | 0.383              | 0.407              | 0.409              |
> | **ETTm2**      | **0.252**             | **0.310**          | 0.261             | 0.319              | 0.296              | 0.338              |
> | **ETTh1**      | 0.429                 | 0.440              | **0.426**         | 0.440              | 0.433              | **0.437**          |
> | **ETTh2**      | **0.354**             | **0.392**          | 0.358             | 0.400              | 0.380              | 0.403              |
> | **ECL**        | **0.166**             | **0.262**          | 0.170             | 0.265              | 0.179              | **0.262**              |
> | **Traffic**    | 0.439                 | **0.295**          | **0.435**         | 0.313              | 0.797              | 0.427              |
> | **Weather**    | **0.229**             | **0.258**          | **0.229**             | 0.268              | 0.252              | 0.277              |
> | **Exchange**   | 0.457                 | 0.446              | 0.410             | 0.431              | **0.322**          | **0.384**          |
> | **ILI**        | 3.194                 | 1.234              | **2.233**         | **1.015**          | 2.640              | 1.018              |
> | **1stCount**   | **11**                |                    | 5                 |                    | 4                  |                    |
>
> **Q3.** It is difficult to classify the third purported contribution, discussing emerging models, as genuinely a valid contribution point.
>
> **A3.** We thank you for this comment and would like to clarify the value of this contribution. Our work moves beyond treating large models holistically, as is common practice, and instead provides **the first systematic analysis of their compatibility** with other key design dimensions. This fine-grained analysis yields critical, evidence-based guidance. For instance, it reveals **that series normalization is essential for stable LLM/TSFM** performance and that certain backbones, like Time-LLM, are highly sensitive to other design dimensions.
>
> Crucially, by understanding these component synergies, TSGym's automation can then **construct LLM/TSFM-based models** that are tailored to the **specific characteristics of a given dataset**, creating more effective and robust solutions. Therefore, we believe this work offers a valuable contribution by providing a practical framework and actionable principles for building better, customized large models for time series forecasting.

---

> > ### Comment · Reviewer_ni7G · 2025-08-06
> >
> > Dear authors,
> >
> > Thanks for the detailed responses, and the extra experiments. These primarily addressed my concerns and questions, I am happy to recommend an accept rating for this manuscript.

---

> ### Author Response · Authors · 2025-08-04
> **Follow-up On Our Responses**
>
> We want to send this friendly reminder and are happy to address any further comments reviewers may have. Also, If our responses address your questions, we want to kindly ask for your reconsideration of the score.
>
> Thank you so much for devoting time to making TSGym a better work. We are looking forward to any further discussions :)

---

### Official Review · Reviewer_8DCH · 2025-07-03

**Rating:** 5
**Confidence:** 5

**Summary:**

The current research in MTSF tends to evaluate models holistically, which obscures individual contributions and leaves critical issues unaddressed. This work bridges these gaps by systematically decomposing deep MTSF methods into core, fine - grained components such as series - patching tokenization, channel - independent strategy, attention modules, or even Large Language Models and Time - series Foundation Models. The key contributions are as follows:

1 TSGym, the first large-scale benchmark that systematically decouples deep MTSF methods.

2 Automated MTSF model construction.

3 Discussion on emerging large time-series models.

**Additional Feedback:**

See Limitations Weaknesses

**Dataset Code Accessibility:**

Yes

**Dataset Code Comments:**

The authors have supplied comprehensive code, which offers a novel benchmark for researchers in this field.

**Ethical Considerations:**

No, there are no or only very minor ethics concerns

**Final Justification:**

The author's response has resolved my concern. I maintain my recommendation for acceptance.

**Limitations Weaknesses:**

1 The lines in Figs. 1b and 1f are overly crowded, making it difficult to distinguish differences between them. Further improvements are needed to enhance clarity.

2 Most current deep learning methods are designed with an end-to-end and globally optimal approach. By decomposing MTSF into various sub-stages, does this approach risk undermining global optimality in favor of local optimality?

3 Is this decomposition method overly cumbersome? Which phase has the most significant impact on the task?

**Strengths Contributions:**

The authors introduce TSGym, a novel framework that systematically deconstructs deep multivariate time series forecasting (MTSF) models into independent design dimensions, enabling fine-grained evaluation and automated, data-adaptive model construction. By benchmarking 16 design dimensions across 10 datasets and integrating diverse architectures including MLPs, Transformers, LLMs, and TSFMs, TSGym not only advances the understanding of performance drivers in MTSF but also surpasses existing state-of-the-art methods, significantly broadening the scope and automation potential of time series forecasting research. In addition, this paper has the following strengths:

1 The paper is well-written, with a clear logic and rich experiments.

2 This work can bring a new perspective benchmark to the field.

3 The code for this work has been released, providing researchers with a great baseline.

---

> ### Author Rebuttal · Authors · 2025-07-31
>
> **Q1**. The lines in Figs. 2b and 2f are overly crowded, making it difficult to distinguish differences between them. Further improvements are needed to enhance clarity.
>
> **A1**.
> We agree that the dense lines hinder readability. While we explore alternatives like shaded bands, the visual clutter persisted. For this reason, we present the full results in the appendix with detailed tables and boxplots for readers who wish to dig deeper. Importantly, the very tangle of lines conveys an important message: **no single design choice achieves consistently superior performance across datasets along this dimension**. We will explicitly report this conclusion in the revised paper.
>
> **Q2**. Most current deep learning methods are designed with an end-to-end and globally optimal approach. By decomposing MTSF into various sub-stages, does this approach risk undermining global optimality in favor of local optimality?
>
> **A2**.
> Thank you for this critical question. We **clarify a distinction between optimality in model training versus model design**. While current SOTA deep-learning models are trained for optimal weights, their designs are often limited, exploring only a narrow search space by fixing most components to default settings.
>
> TSGym addresses this limitation at the design stage. By following the standard MTSF modeling pipeline to perform a fine-grained decoupling of existing methods, it explores novel synergies of components missed by fixed SOTA architectures. The results validate this approach: the optimal design found by our search significantly outperforms all existing SOTA models. This demonstrates that our method is a more effective strategy for finding a globally optimal design.
>
> **Q3**. Is this decomposition method overly cumbersome? Which phase has the most significant impact on the task?
>
> **A3**.
> Thanks for your insightful feedback. We agree that our 16-dimension decomposition is analytically complex, and this complexity mirrors the current state of the MTSF research field itself. The landscape has grown incredibly diverse, e.g., a vast body of research has focused on developing novel network architectures, from MLPs, RNNs, CNNs, and Transformers to emerging large time-series models (LLMs/TSFMs). Furthermore, innovation is occurring across numerous dimensions beyond backbones, including normalization, decomposition, and various input encoding strategies. This rapid expansion makes it increasingly difficult to discern which components genuinely drive performance.
>
> It is precisely this challenge that necessitates our fine-grained approach. While we acknowledge the analytical complexity, we believe this level of decomposition is essential for systematically understanding the MTSF design space, and TSGym is designed to manage this complexity for the user. To quantitatively investigate the influence of four phases during MTSF pipeline (shown in Fig. 1)—*Series Preprocessing*, *Series Encoding*, *Network Architecture*, and *Network Optimization*, we design an ablation-based experiment. We find that, compared to complex model structures and training methods, **the series preprocessing and encoding phases are more crucial**. We will present these findings and the exploration process in the revised paper to enhance the utility of TSGym. The experiment details are as follows:
>
> Firstly, we establish the typical design choices as a baseline for 9 original datasets and 3 additional datasets (AQShunyi, AQWan, and CzeLan), which is defined as follows: *{Series Normalization=Null, Series Decomposition=Null, Series Sampling=False, Channel Independent=False, Sequence Length=48, Series Embedding=Positional Encoding, Network Type=RNN, Series Attention=Null, Feature Attention=Null, d_model=64, d_ff=256, Encoder Layers=2, Epochs=10, Loss Function=MSE, Learning Rate=1e-3, Learning Rate Strategy=Null}*. This baseline is applied to 48 experimental settings across 12 datasets and 4 different forecasting lengths.
>
> Next, to evaluate the impact of each phase, we keep the other three phases fixed at their simplest design choices and select a range of combinations for the current phase under evaluation. For example, when evaluating the impact of the Network Architecture phase, we fix the other three phases to the combinations *{Series Normalization=Null, Series Decomposition=Null, Series Sampling=False, Channel Independent=False, Series Embedding=Positional Encoding, Loss Function=MSE, Learning Rate=1e-3, Learning Rate Strategy=Null}* and isolate the results varied during this phase.
>
> To account for differences across datasets, we measured the influence of each phase by comparing the relative improvement of the current phase against the corresponding baseline for each dataset and forecasting length. This is formally defined as:
>
> $$AvgRelErr_{phase} = \frac{1}{|S_{phase}|} \sum_{s \in S_{phase}} ( -\frac{MSE_{s}^{variant} - MSE_{s}^{baseline}}{MSE_{s}^{baseline}} )$$
> where
> - $S_{phase}$ is the set of all (dataset, horizon) settings that satisfy the ablation criteria for the phase;
> - $MSE_{s}^{variant}$ is the MSE of a candidate configuration under setting s;
> - $MSE_{s}^{baseline}$ is the MSE of the minimal baseline under the same setting s.
>
> The table below reports the maximum, mean, and standard deviation of the relative improvement for each phase compared to the corresponding baseline. These values represent the potential, average improvement level, and stability of the current phase, respectively. In summary, compared to complex model structures and training methods, sequence preprocessing and encoding phases are more important.
>
> |**Phase**|**Max Improvement**|**Mean Improvement**|**Std Improvement**|
> |:-:|:-:|:-:|:-:|
> |**Series Preprocessing**|90.87%|19.96%|0.394|
> |**Series Encoding**|83.90%|30.98%|0.248|
> |**Network Architecture**|66.75%|7.12%|0.290|
> |**Network Optimization**|26.54%|10.01%|0.109|

---

### Official Review · Reviewer_2xTN · 2025-07-03

**Rating:** 4
**Confidence:** 4

**Summary:**

The paper describes TSGym -- an evaluation framework for timeseries models and an automated model selection approach utilizing metalearning. TSGym analyzes forecasting approaches by decomposing them into different components, which is motivated by the insufficient scope and granularity of previous studies. The impact of the different components is analyzed, MLP models are shown competitive with transformers across many conditions, foundation models are shown to often provide benefit, and the metalearning approach improves over SOTA in many conditions.

**Additional Feedback:**

- In Fig. 3, how was it decided to focus on the 75th percentile specifically? Do the results look similar at other percentiles?
- Why is TSMixer not in Tab 2/3 but included in the discussion in the main text?
- How was the subset of the datasets used for the experiments reported in Tab. 5 / Fig. 3 selected?
- What was the cardinality of M_s for training the meta-learner? Have experiments been done with different draws of the combinations of the design dimensions? What was the impact if so? How does the meta-learner performance scale with the size of M_s?
- How many distinct combinations of the meta features were seen during training of the meta-learner?
- What was the training data used for the meta learner? Are any of the meta-learner evaluations done on datasets that were not used during training at all?
- Fig. G1 reports a PCA analysis of meta features. What can we conclude about the similarity of the datasets from this? Can this be explained in interpretable terms that might apply to other datasets?

**Dataset Code Accessibility:**

Yes

**Dataset Code Comments:**

The code is available on GitHub. The paper does not contribute new datasets.

**Ethical Considerations:**

No, there are no or only very minor ethics concerns

**Final Justification:**

In their rebuttal, the authors have addressed the main points that I raised regarding the clarity of the new insights provided by TSGym and the details of how the meta-learner was trained and whether it works effectively on new datasets. I also really appreciated their willingness to engage in the discussion with all reviewers and provide additional evidence where needed. The final quality of the paper will depend on the careful inclusion of the new data into the manuscript and adjusting the presentation and discussion of the claims, on which I'm giving the authors the benefit of the doubt and raising my score to 4 accordingly.

**Limitations Weaknesses:**

- The abstract promises "more profound insights" but the text does not clearly back up this statement anywhere. Similarly, the importance of the decoupled analysis is highlighted, but the paper would benefit from being clearer about what novel insights it enabled.
- The paper includes interesting observations regarding the good performance of MLPs and TSFMs. It would be helpful to have them summarized in a clear and concise way somewhere as key takeaways.
- Very few conclusions apply universally across all or most datasets, which in itself is not surprising. Can the observations in e.g. Fig. 2 be explained by the properties of the datasets? For instance, could specific conclusions only hold for low dimensional datasets, etc? Could the metalearner be analyzed to highlight which features are most relevant for any given modeling approach?
- nit: 151: methods proposing -> methods proposed

Regarding presentation, I think the paper does some of the above directionally already, but does not go far enough and in places the text feels like it's just superficially summarizing data already visible in the figures, without drawing any deeper conclusions (which also stands in contrast to the "profound insights" mentioned in the abstract). For instance, "channel-independent methods consistently  outperform channel-dependent ones, with the exception of traffic and ILI". This is a valuable insight. But why are traffic and ILI different? When should one use a channel-independent method with a new dataset? There is *a lot* of figures and tables within the paper overall, especially when one includes the appendices. What are some general high level insights and guidelines for future work that a practitioner should take away from it? This would ideally be explicitly stated in the conclusions section. As is, the text reads too much like "we ran a lot of experiments" (great) and "are presenting the results here" (also great, but needs more analysis to be truly useful).

Based on what I was able to find the paper, it was unclear to me how well would the metalearner be expected to perform on a completely novel dataset. This is important to clarify (see feedback questions below).

**Strengths Contributions:**

- Isolated evaluations of core components of a time-series modeling pipeline (4 stages: preprocessing, encoding, architecture, optimization), broken down into 16 design dimensions.
- Extensive baselines (27 models) covering six model types: RNNs, CNNs, Transformers, MLPs, LLMs, TSFMs.
- Intro section provides extensive citations.
- Improved results from automated model construction with the TSGym metalearner across many datasets.

Overall, the paper is a valuable and extensive exploration of MTSF models, which is its primary strength and contribution. The comprehensiveness of the evaluated combinations goes beyond prior work. The authors also explicitly discuss how the current effort is similar but different to their prior work in ADGym. The paper would be strengthened with some improvement to the presentation of the results (see below).

The metalearner approach also appears to be quite promising, and could be a second significant contribution but I'd like to ask the authors to clarify some questions around it (see "additional feedback").

---

> ### Author Rebuttal · Authors · 2025-07-31
>
> **Q1**. Lacking deeper dataset-grounded analysis and succinct key takeaways.
>
> **A1**.
> Thank you for the suggestions. To delve deeper into how each component behaves, we formulate **7 most contentious claims** and clarify them with our benchmark under 9 origin datasets and 3 new added ones, AQShunyi, AQWan, and CezLan. These in-depth insights substantially raise the benchmark’s quality and will be **highlighted** in the Introduction and Experiment section.
>
> - **Claim 1**: Model scale should match data scale.
>
> - **Conclusion 1: Yes.** We validate this claim from 2 aspects:  1) **Hidden-dimension alignment**: On the two high-dimensional datasets (ECL and Traffic), the 256 d_model surpasses 64 across all four prediction horizons, while for the remaining datasets, 64 outperforms 256 in 32 of 40 settings. 2) **Model Backbone**: Transformer dominates on the three largest datasets (ECL, Traffic, Weather) across all 12 settings, whereas MLP prevails in 29 of 36 smaller-scale settings.
>
> - **Claim 2**: Transformers are less robust than MLPs.
>
> - **Conclusion 2: Yes.** Using the inter-quartile range (IQR) as the robustness metric, Transformers fare worse than MLPs in **34 out of 48 settings**. The average relative IQR gap, $\frac{1}{N} \sum_{i=1}^{N} \frac{IQR_{Transformer,i} - IQR_{MLP,i}}{IQR_{MLP,i}}$, reaches as high as **54.6%**. This conclusion is consistent with the findings reported in [2].
>
> - **Claim 3**: Transformers exhibit a higher upper bound than MLPs.
>
> - **Conclusion 3: No.** Even when incorporating multiple attention variants, **Transformers do not guarantee a performance advantage over MLPs.** We compared the upper bound defined as the best result achieved across all of their respective settings. Transformers outperform in 27 of the 48 settings, while underperforming in 21, with an average margin of only +0.038%. This conclusion does not fully align with the findings in [2].
>
> - **Claim 4**: The superiority of Channel-Independent (CI) vs. Channel-Dependent (CD) correlates with inter-channel correlation.
>
> - **Conclusion 4: No.** In the 13 settings where CD models outperform CI models, the average inter-channel correlation is 0.696, versus 0.624 in the 35 settings where CI is preferred. Yet ECL, with the second-highest correlation 0.802, still favors CI on every horizon. This finding suggests the CI/CD decision must consider the data's semantic context, not just statistical correlation, which challenges the purely correlation-based conclusions from prior work [1].
>
> **Due to space constraints**, we highlight 3 additional claims that were also validated by our extensive experiments. We would be happy to discuss these and other findings in more detail.
>
> - **Claim 5**: Series normalization mitigates distribution shift. (**Yes**)
>
> - **Claim 6**: Novel sequence encodings outperform the traditional single-timestep encoding. (**Yes**)
>
> - **Claim 7**: Novel attention mechanisms outperform vanilla self-attention. (**Yes**)
>
> [1] Shao Z, et al. Exploring progress in multivariate time series forecasting: Comprehensive benchmarking and heterogeneity analysis[J]. TKDE, 2024.
>
> [2] Qiu X, et al. TFB: Towards Comprehensive and Fair Benchmarking of Time Series Forecasting Methods[J]. VLDBJ, 2024.
>
> **Q2**. Can the observations in e.g. Fig. 2 be explained by the properties of the datasets?  Could the metalearner be analyzed to highlight which features are most relevant for any given modeling approach?
>
> **A2**.
> To directly investigate feature importance, we conducted a new analysis using an interpretable XGBoost meta-learner.
> While this ML-based version is slightly outperformed by our original DL-based model (0.447 vs 0.426 at average MAE, as it cannot learn rich, high-dimensional embeddings for the model combinations), its competitive performance **allows us to analyze feature importance and provide deeper insights you called for.** The results shown below reveal that specific meta-features (e.g., spectral MFCCs and temporal Negative Turning Points) and design dimensions (most notably, series normalization) are consistently the most critical factors in model selection across diverse datasets.
>
> |||||
> |:-:|:-:|:-:|:-:|
> |**ETTm1**|mean_Negativeturningpoints|series norm|mean_Centroid|
> ||0.08|0.06|0.05|
> |**ETTh1**|mean_MFCC|series norm|mean_Spectralroll-on|
> ||0.08|0.06|0.05|
> |**ECL**|mean_MFCC|std_MFCC|series norm |
> ||0.08|0.08|0.07|
> |**Traffic**|q25_Kurtosis|series norm|mean_Centroid|
> ||0.08|0.07|0.06|
> |**Weather**|mean_MFCC|mean_Negativeturningpoints|series norm |
> ||0.14|0.11|0.06|
> |**Exchange**|std_MFCC|mean_Negativeturningpoints|mean_MFCC|
> ||0.11|0.1|0.07|
> |**ILI**|mean_MFCC|mean_Maximumfrequency|channel independent|
> ||0.09|0.06|0.05|
>
> **Q3**. Typographical errors regarding “methods proposing” and “TSMixer”.
>
> **A3**.
> We corrected the mistakes. Additionally, “TSMixer” is our typographical error; the correct model name is "TimeMixer".
>
> **Q4**. In Fig. 3, how was it decided to focus on the 75th percentile specifically? Do the results look similar at other percentiles?
>
> **A4**.
> We apologize for the lack of clarity. To be precise, the metric we report is the top **25th percentile** of the error metric. We chose this because it reflects a realistic yet strong model performance, avoiding the unrealistic best-case outlier. Furthermore, while a full analysis of the performance distributions—including best, median, and IQR—is available in the appendix, our key findings, such as the consistent effectiveness of Series Normalization, remain robust across these different statistical views. We will clarify this in our revised manuscript.
>
> **Q5**. How was the subset of the datasets used for the experiments reported in Tab. 5 / Fig. 3 selected?
>
> **A5**.
> We use ETTh1, ETTh2, Exchange, and ILI for their modest cost; large datasets such as ECL and Traffic preclude exhaustive runs. To curb randomness, we conduct ample experiments on these four sets, ensuring reliable conclusions.
>
> **Q6**. Report the training cardinality of M_s, the sensitivity to alternative design-space samples, and the scaling of meta-learner performance with |M_s|.
>
> **A6**.
> In our experiments, the $M_s$ contained approximately 500 model combinations on average per dataset, with the exact number varying based on dataset scale. To analyze how meta-learner performance scales with the size of this pool, we trained our meta-learner on progressively larger subsets of the pool (from 5% to 100%) and evaluated the average rank of the model selected by TSGym. The results are presented in the table below.
>
> |$M_s$|MSE(avg. Rank)|MAE(avg. Rank)|
> |:-:|:-:|:-:|
> |5%|3.67|3.44|
> |10%|2.67|3.00|
> |25%|1.67|1.78|
> |50%|1.89|2.67|
> |75%|1.89|2.22|
> |100%|1.67|2.00|
>
> We found that the meta-learner's accuracy rises until pool size reaches 25%, then plateaus. Notably, we show that TSGym is highly sample-efficient: even with 10% of combinations, it already outperforms strong baselines like DUET (avg. Rank 4.11 for mse and 3.67 for mae). This confirms that a moderately-sized pool is sufficient for near-optimal performance and inspires future work into smarter sampling strategies, such as Bayesian Optimization, which could further enhance this efficiency. We will add this analysis to our paper.
>
> **Q7**. How many distinct combinations of the meta features were seen during training of the meta-learner?
>
> **A7**.
> The meta-features reported in Section 4 are derived from TSFEL [1], yielding a 156-dimensional embedding per channel. For multivariate series, we aggregate across channels via nine statistics (mean, median, etc.), producing a 1,404-dimensional embedding.
>
> [1] Barandas M, et al. TSFEL: Time series feature extraction library[J]. SoftwareX, 2020, 11: 100456.
>
> **Q8**. Define the meta-learner’s training data, confirm zero-shot evaluation framework, and estimate performance on novel data.
>
> **A8**.  The meta-learner’s goal is to predict a decomposition's relative ranking on a dataset. Each sample pairs (i) learnable embeddings of its design choices and (ii) meta-features describing the dataset.
>
> **No target test datasets are used during training**, making our meta-learner a zero-shot AutoML model. For example, the performance we report on ETTh1 is produced by a meta-learner trained solely on the other 8 datasets;
>
> Following your suggestions, we fit the meta-learner on the origin 9 datasets and evaluated on 3 unseen datasets (AQShunyi, AQWan, and CzeLan). TSGym beats 9 other baselines on the 3 new datasets in most settings, ranking **first 15 times** and **second 9 times** in all 24 settings (4 prediction horizons, 2 metrics). Owing to text length limit, complete results are not listed here but will be added in updated script.
>
> **Q9**. Fig. G1 reports a PCA analysis of meta features. What can we conclude about the similarity of the datasets from this? Can this be explained in interpretable terms that might apply to other datasets?
>
> **A9**.
>
> 1. Our PCA analysis of meta-features (Fig. G1) confirms that datasets cluster by inherent properties like domain (e.g., ETT) and temporal frequency (e.g., M4-Hourly vs. M4-Yearly). This demonstrates that **the similarity between datasets in the meta-feature space indicates the potential for knowledge transfer.**
>
> 2. We conducted a case study on the testing ILI dataset by enriching the meta-learner's training pool with two new, more similar datasets (COVID-19 and FRED-MD), where ILI's performance (see the table below) **significantly improved without degrading the other datasets.**
>
> ||TSGym||TSGym(Add covid-19&fred-md)||
> |:-|:-|:-|:-|:-|
> |**Metric**|**MSE**|**MAE**|**MSE**|**MAE**|
> |**ETTm1**|**0.357**|**0.383**|0.360|0.386|
> |**ETTm2**|0.261|**0.319**|**0.258**|0.320|
> |**ETTh1**|**0.426**|0.440|0.429|**0.436**|
> |**ETTh2**|**0.358**|**0.400**|0.368|0.406|
> |**ECL**|0.170|**0.265**|0.170|0.269|
> |**Traffic**|0.435|0.313|**0.428**|**0.306**|
> |**Weather**|0.229|0.268|**0.228**|**0.266**|
> |**Exchange**|0.410|0.431|**0.373**|**0.408**|
> |**ILI**|2.233|1.015|**2.098**|**0.941**|

---

> > ### Comment · Reviewer_2xTN · 2025-08-05
> >
> > Thank you for the extensive response and for addressing my questions! The clarifications around the details of the metalearner and the explicit discussion of the "contentious claims" in particular should significantly strengthen the paper, so I will increase my score accordingly. I'm looking forward to the revised manuscript!

---

> > > ### Author Response · Authors · 2025-08-05
> > > **Thanks for your constructive feedback and endorsement**
> > >
> > > We sincerely appreciate your constructive feedback and endorsement; your insights have substantially enriched TSGym into a more incisive benchmark work. In a forthcoming release, we will report the conclusions derived from these findings and, simultaneously, make the complete set of experimental results publicly available in our GitHub repository, enabling researchers in the field to conduct analyses. Thank you!

---

> ### Author Response · Authors · 2025-08-04
> **Follow-up On Our Responses**
>
> We want to send this friendly reminder and are happy to address any further comments reviewers may have. Also, If our responses address your questions, we want to kindly ask for your reconsideration of the score.
>
> Thank you so much for devoting time to making TSGym a better work. We are looking forward to any further discussions :)

---

### Note · Authors · 2025-08-12

We sincerely thank all the reviewers for their encouraging and insightful comments. The primary changes are summarized below:

**Deeper Insights:**

- Based on the extensive experimental results, we discuss seven proposed claims and offer constructive conclusions (2xTN, jDWu).
- Through a more transparent tree-based meta-learner structure, we provide an importance analysis of both meta-features and design dimensions. (2xTN, jDWu)
- We explore the relationship between the similarity and transferability of meta-features across different datasets, and investigate the benefit of incorporating more similar datasets for target dataset enhancement. (2xTN)
- We design a new ablation experiment to systematically evaluate the importance of the four key phases in MTSF. (8DCH)
- We examine how the size of model combinations impacts downstream performance of meta-leaner, revealing the marginal effect of experimental scale and the value of smarter sampling methods. (2xTN, jDWu)

**Smarter Sampling Method:**
- Implementing Bayesian optimization via Optuna, we explore the impact of smarter sampling on preparing the meta-training data (i.e., model combinations). Our experiments demonstrate that Optuna effectively reduce experimental costs and improve automated selection performance. (jDWu)

**New Datasets and Baselines:**
- We add three new datasets, AQShunyi, AQWan, CzeLan, to investigate performance differences under specific data characteristics such as distribution shift. (2xTN)
- We add two AutoML-based methods, AutoGluon and AutoTS, as baselines and validate TSGym's effectiveness compared to these AutoML methods. (ni7G)

**Paper Writing:**
- We correct typos in the paper and clarify the metrics used to analyze design choices. (2xTN)
- We emphasize the key conclusions from the experiments in the Introduction and Experiment sections.(2xTN)
- We create a new, detailed diagram illustrating the step-by-step flow from the input tensor to the output series in the TSGym pipeline. (jDWu)

**Long-Term Plan:**
We will maintain and foster the long-term development of TSGym, building it into an active, open platform to provide automated solution for MTSF tasks, and also evaluating the effectiveness of proposed components in the latest MTSF advancements. Furthermore, the framework is designed to progressively integrate these validated components, allowing TSGym to continuously evolve and improve.

---

### Decision · Program_Chairs · 2025-09-18

**Decision:**

Reject

**Comment:**

Several reviewers increased their score after the discussion with the authors, and now all reviewers recommend acceptance. The topic of the paper is central for NeurIPS. Poster presentation is appropriate, because the results are not especially surprising.

===== FINAL UPDATE FROM DB Track PCs ====

The final decision for this paper has been taken by the program chairs after consultation with the SACs. All Senior Area Chairs have ranked papers according to the feedback from the AC during the review process. We decided to leave the original meta-review to reflect the opinion of the AC in light of the initial discussions with reviewers and SAC.